# Is the sparsity of high dimensional spaces the reason why VAEs are poor generative models?

## Abstract

Variational autoencoders (VAE) encode data into lower-dimensional latent vectors before decoding those vectors back to data. Once trained, decoding a random latent vector from the prior usually does not produce meaningful data, at least when the latent space has more than a dozen dimensions. In this paper, we investigate this issue drawing insight from high dimensional physical systems such as spin-glasses, which exhibit a phase transition from a high entropy random configuration to a lower energy and more organised state when cooled quickly in the presence of a magnetic field. The latent vectors of a standard VAE are by construction distributed uniformly on a hypersphere, and thus similar to the high entropy spin-glass state. We propose to formulate the latent variables of a VAE using hyperspherical coordinates, which allows compressing the latent vectors towards an island on the hypersphere, thereby reducing the latent sparsity, analogous to a quenched spin-glass. We propose a new parametrization of the latent space with limited computational footprint that improves the generation ability of the VAE.

## 1 Introduction

In today's machine learning landscape, and deep learning in particular, one of the main mathematical tools to represent data (e.g. images) are high dimensional (HD) Euclidean spaces.

However, our intuition about Euclidean geometry stems from the physical world and everyday life, which are low dimensional spaces (mostly two and three dimensions). This presents a challenge because HD spaces behave, mathematically speaking, in very different ways than their low dimensional counterparts, often in ways that seem counterintuitive or even paradoxical if interpreted through low dimensional intuition.

HD spaces have been used in physics, for example to model the state space of systems such as magnetic materials. We will argue in this paper that the state space of some physical systems studied in statistical physics has a remarkable similarity with the HD spaces created by generative models, in particular Variational Autoencoders (VAE) (Diederik P Kingma, 2014). We will discuss how our handling of HD spaces in machine learning can benefit from the intuition about those physical systems.

We will highlight how issues associated with VAE are related to volume and entropy, that create voids and sparsity in latent spaces, hampering their performance to generate meaningful new samples, even when reconstruction metrics can be optimized very well.

Finally, we will propose a method that parametrize latent data on the hypersphere with hyperspherical coordinates. This allows manipulating the data distribution on the latent manifold more effectively. In particular, we use it to compress the latent manifold volume and reduce the sparsity. This is made possible thanks to an efficient transformation between Cartesian and hyperspherical coordinates, which can be implemented with minimal computational overhead using a fully vectorized algorithm, for high enough dimensions (it becomes costly in the very large case; but, as our results show, those cases are of no practical interest from the point of view of the metrics we are checking, see Fig.2).

## 1.1 VARIATIONAL AUTOENCODER

An autoencoder (AE) is a common self-supervised method to encode the data into a latent space of lower dimension $n$. Variational autoencoder (VAE) (Diederik P Kingma, 2014; Kingma & Welling, 2019) introduces a prior to control the distribution of the latent, allowing to sample the latent space and untangle the dimensions (Rolínek et al., 2019; Bhowal et al., 2024). In its simplest implementation, a VAE consists in a probabilistic encoder which, for each input point $x \in X$ from a dataset $X$, produces a latent distribution $q_x = \mathcal{N}(\mu_x, \sigma_x)$. During training, the reparameterization trick is used: the encoder estimates $\mu$ and $\sigma$, and a sample $z$ from $q_x$ is computed as $z = \mu + \epsilon \odot \sigma$, where $\odot$ denotes element-wise multiplication. Then, the decoder is applied to this sample to obtain the reconstructed datapoint, $\hat{x}_z$. The VAE's loss can then be interpreted as an AE (with its Mean Square Error, MSE, loss) with an additional term, KLD $(\mathcal{N}(\mu_x, \sigma_x) \| \mathcal{N}(0, I))$, that regularizes the latent space by forcing each of the encoded distributions to become similar to a prior one ($\mathcal{N}(0, I)$ in this implementation), where *KLD* refers to the Kullback-Leibler divergence. The two mentioned terms are computed over a mini batch of size $N_b$:

$$\text{MSE}(x, \hat{x}_z) = \frac{1}{N_b} \sum_{l=1}^{N_b} \| x_l - \hat{x}_z^l \|^2, \ \text{KLD}(z, \epsilon) = -\frac{1}{2} \sum_{l=1}^{N_b} \sum_{k=1}^{n} \left(1 + \log\left(\sigma_{k,l}^2\right) - \mu_{k,l}^2 - \sigma_{k,l}^2\right) \quad (1)$$

The final cost to be optimized weights the two terms with the gain $\beta$ (Higgins et al., 2017):

$$\mathcal{L} = \text{MSE}(x, \hat{x}_z) + \beta \text{KLD}(z, \epsilon) \quad (2)$$

The prior, and thus the latent, is a high dimensional independent multivariate Gaussian, which has specific properties that we briefly recall in the next section, by way of background for the following sections.

## 1.2 HIGH DIMENSIONAL SPACES IN MATHEMATICS

A multivariate Gaussian sampling in a HD Euclidean space of dimension $n$ is such the probability to find samples close to the origin is close to zero (despite having maximum probability density) and most of the samples lie close to a $(n-1)-$hypersphere, $\mathbb{S}_{\sqrt{n}}^{n-1}$, of radius $\sqrt{n}$. The distribution of the norm of those samples follows a $\chi(n)$ distribution. Therefore, the samples are located within a region very close to the hypersphere, region which becomes very thin in high dimensions, relative to the radius $\sqrt{n}$. These effects are called 'concentration of measure', in the mathematical literature.

As $n$ increases, a multivariate Gaussian tends towards the uniform distribution on that hypersphere. In addition, any two samples from $\mathcal{N}(0, I_n)$ are always almost orthogonal to each other (this is called almost-orthogonality). A formal description of these phenomena can be found elsewhere (Vershynin, 2018). See also Appendix A.4.

These facts are closely related to how *(hyper-)volume* behaves in HD spaces: if we consider the standard uniform measure on the hypersphere, then most of its volume or mass is *concentrated* in very thin 'equatorial' bands for *any* randomly chosen north pole (this is, of course, just an intuitive statement, for a formal description see Wainwright (2019)). The contrast with our intuition coming from two dimensional spheres is remarkable. In the next section, we will review that similar HD spaces exist for spin-glass systems.

## 1.3 HIGH DIMENSIONAL SPACES IN (STATISTICAL) PHYSICS

Consider a system consisting of $n$ persons each simultaneously tossing a coin. After the tossing, we can record the result with a vector $x \in \{H, T\}^n$ (H for heads, T for tails). For example, it could be $x_0 = (H, H, T, H, ..., T)$. Each of these vectors is called a possible *microstate* for the system.

Given a microstate, we could define a function $F_H(x)$ on microstates which for example, counts how many heads are in that microstate: the value obtained is called a *macrostate*. Different microstates can give rise to the same macrostate. In statistical mechanics, the (Boltzmann) entropy of a system is proportional to the natural logarithm of the number of different microstates giving rise to a same

macrostate. The configurations that maximize this number (and, thus, also entropy) are the thermal equilibrium ones.

For a general continuum microstate space, the thermal equilibrium configurations are characterized by the Gibbs probability measure: $\mu(\mathrm{d}x) \propto e^{-\beta\mathcal{H}(x)}\nu_0(\mathrm{d}x)$, where $\beta$ is the inverse temperature, $\nu_0(\mathrm{d}x)$ is a fixed reference measure on the submanifold of $\mathbb{R}^n$ formed by all the possible microstates, and $\mathcal{H}(x)$ is the energy function of the system (Bricmont, 2022).

there is a class of systems that have been extensively studied called *spin glasses* (Parisi, 2002). In these systems, the submanifold of microstates is given precisely by the $(n-1)$-hypersphere, $\mathbb{S}^{n-1}_{\sqrt{n}}$, and $\nu_0(\mathrm{d}x)$ by the standard uniform measure on it. The dimension $n$ is taken to be very large. The uniform measure is one of maximal entropy, even among all the measures of thermal equilibrium.

At very high temperatures, $\mu(\mathrm{d}x) \approx \nu_0(\mathrm{d}x)$, two 'replicas' of the system are two *i.i.d.* samples from the Gibbs measure. Their overlap, measured by their inner product, will be almost zero, because of the almost-orthogonality effect of HD spaces mentioned before. This means that these samples are in the 'equatorial' region, where most of the volume is concentrated. The system is said to be in a 'replica symmetric phase'.

Until a critical temperature value, all solutions have this qualitative behaviour. This is to be expected since the function whose stationary points define a general solution for these systems is indifferent to a permutation of replicas. What is surprising in these systems is the emergence at low enough temperatures of a different phase, which is *not* replica symmetric; that is, the inner product or correlation between them is appreciably different from zero. This is called the 'replica symmetry breaking phase' (Montanari & Sen, 2024). Two replicas, in this case, lie in a very thin band or ring centered at a deep minimum of the energy function (Subag, 2017), center which is not in the initial high volume equatorial region. This ring is a $(n-2)$-hypersphere, submanifold of the initial $(n-1)$-hypersphere.

We discussed so far the 'static' structure of the Gibbs measure across the temperature range. A separate and complicated question is how, starting from a high temperature state, one can reach *dynamically*, by some physical process[1], the replica symmetry breaking phase; that is, to dynamically achieve this *phase transition*.

This can be very tricky, since the local minima landscape of the energy in the high entropy region is very rugged (with exponentially many local minima), and one would need to overcome it first in order to reach the less entropic regions of the hypersphere (Arous & Jagannath, 2024). In practice, it is usually done by 'quenching' processes, where the system is suddenly cooled down in the presence of externally applied magnetic fields. The energy function for a $p-$spin glass in the presence of an external magnetic field $\mathbf{h}$ is given by $\mathcal{H}(x) = \mathcal{P}_p(x) - \sum_j h_j x_j$, where $\mathcal{P}_p(x)$ is a homogeneous random $p-$polynomial (we omit the normalization constants).

The high entropy 'replica symmetric phase' of the spin-glass is analogous to the training of a VAE where the KLD term forces the latent samples to be uniformly distributed on the hypersphere (the dynamics of spin glasses and the training process of some deep learning models has already received some attention in the literature, for example in Baity-Jesi et al. (2018), and also Arous et al. (2022); in this work, though, we will focus our attention on some other issues, more geometrical in nature). In the next section, we argue that this brings an issue related to the sparsity of the resulting latent space.

## 1.4 HIGH DIMENSIONAL SPACES IN GENERATIVE MODELS (VAE)

One use case of the VAE is to generate data. Since the latent distribution is known (high dimensional multivariate Gaussian), one could sample from that distribution and decode the latent vector to generate a novel data sample. This works very well for simple data and low dimension latent spaces (e.g., MNIST with $n = 2$, as done in Diederik P Kingma (2014)), but not so well for more complicated data or high dimensional latent (e.g., CIFAR10, or MNIST with $n = 32$ as in Cinelli et al. (2021)). Indeed, variations of the VAE are commonly used in generative models of images and

---

[1]Typically, a Langevin dynamics of the form $\mathrm{d}x_t = \mathrm{d}B_t - \beta\nabla\mathcal{H}(x_t)\mathrm{d}t$, for which the Gibbs measure is stationary.

videos (Jonathan Ho Ajay Jain, 2020), but the latent space is not sampled directly: instead, a reverse diffusion process dynamically transforms a random sample into a valid latent location.

One of the reasons why VAE perform poorly when sampling directly the high dimensional latent is because of conflicting constraints. On the one hand, high resolution images need many latent dimensions to capture all the information they convey. Then, the VAE KLD divergence term in the loss encourages the system to distribute this HD latent uniformly on the hypersphere, that is, to maximize entropy. By doing so, the latent becomes extremely sparse given the number of training samples and the immensity of the latent hyperspherical manifold (the volume growing exponentially with the number of dimensions; in these regimes, trying to find a *specific* microstate is akin to a '*needle in the haystack*' kind of situation). The sparsity hampers any attempt to model the latent as a continuous manifold (Peng et al., 2023). But, on the other hand, this usually leads to meaningless decoding of random samples from the prior. These two opposing forces (sparsity due to HD spaces needed to model high resolution images vs need for a continuous manifold for meaningful generation from all of the latent space) conspire to severely limit the capacity of VAEs to function as generative models.

From the previous sections, the regions of high entropy are such that, for a given macrostate, there is an enormous number of different microstates that can realize it. For tasks such as clustering, generation, interpolation, etc., one is interested in specific macrostates of the system. The disordered high multiplicity of possible microstates that can give rise to these macrostates in the high entropy regime may hinder the ability of the model to perform these tasks, as we will explicitly show in the experimental section.

Since the samples in latent space live on the hypersphere, it comes naturally to consider using hyperspherical coordinates to describe latent variables.

## 2 RELATED WORKS

'Hyperspherical Variational Auto-Encoder' Davidson et al. (2018) proposed replacing the standard Euclidean KLD divergence with a KLD divergence between a uniform distribution on the hypersphere as a prior, and a von Mises-Fisher distribution as an approximate posterior. A different way of building a Hyperspherical Variational Auto-Encoder is proposed in Bonet et al. (2022), based on a spherical Sliced-Wasserstein discrepancy, and as an extension of the well-known Euclidean models (Soheil Kolouri Phillip E. Pope, 2020). Hyperspherical aspects of data are studied in Löwe et al. (2023), where high dimensional Rotating Features are introduced. Of particular interest for our project, the authors remarked:

'*We represent Rotating Features in Cartesian coordinates rather than (hyper) spherical coordinates, as the latter may contain singularities that hinder the model's ability to learn good representations. [...]. This representation can lead to singularities due to dependencies between the coordinates. For example, when a vector's radial component (i.e. magnitude) is zero, the angular coordinates can take any value without changing the underlying vector. As our network applies ReLU activation on the magnitudes, this singularity may occur regularly, hindering the network from training effectively*'.

As we reviewed in Section 1.2, in high dimensions, the random samples of an independent multivariate Gaussian distribution fall in the equator of a hypersphere, and thus none of them is near the singularities of the hyperspherical coordinates (the poles and the center of the hypersphere).

While the problem of formulating latent spaces given by non-Euclidean Riemannian manifolds, and hyperspheres in particular has been studied, explicit use of hyperspherical coordinates is avoided. At first look, the conversion from Cartesian to hyperspherical coordinates seems to require computationally expensive recurrent trigonometric formulas (see Appendix A.1). Instead, formulations of Riemannian geometry that rely on Cartesian coordinates are used. Riemannian geometry is not just about a curved metric, but also being able to express it in a convenient coordinate system *adapted* to it (of central importance in physics).

The possible use of hyperspherical coordinates has thus been discarded by all the works that we reviewed. Notwithstanding the previous arguments, we believe that the use of hyperspherical coordinates is feasible and can be beneficial, as our results show. In Appendix A.2 we provide a vec-

torized implementation for transforming between hyperspherical and Cartesian coordinates, which adds only a small computational overhead for training a VAE.

Even if one could assemble a sufficiently large training dataset to densely populate the latent hypersphere (an impossible task in practice), the data in high dimensional latent spaces tend to form manifolds with complicated topologies that include holes and cracks. If a random sample falls into one of these holes, it will be decoded into something meaningless. This also affects interpolations, since locations sampled on a trajectory between two valid latent locations are likely to fall between clusters or classes (in the holes and cracks). This problem has been noted for VAE and called the 'prior hole problem' (Tomczak; Lin & Clark, 2020; Cinelli et al., 2021; Asperti et al., 2021; Singh & Ogunfunmi, 2022; Hao & Shafto, 2023; Aneja et al., 2021). It highlights that the approximate posterior of the data often fails to perfectly match the prior: in the case of a Gaussian distribution in high dimensions the uniform distribution on the hypersphere has no hole.

Common approaches for tackling this problem include a learnable prior (rather than static, as in the standard VAE), or using a mixture of Gaussians as a prior (rather than a single one, as in the standard VAE); see Tomczak for discussion of both cases. A different approach is to dispose of the continuum completely and work with discrete latent representations, as in Aaron van den Oord (2018) where the latent vectors are quantized (VQ-VAE). These discrete representations are useful for tasks dealing with a discrete sequence of symbols, otherwise a joint distribution on the dictionary needs to be estimated after the VAE is trained (Salimans et al., 2017).

There are applications that necessitate and use VAEs with a latent continuum. For example, generative models for drug discovery often deal with chemical properties that span a continuous spectrum (e.g. measure of synthesizability by living entities Ochiai et al. (2023)).

Despite the fact that some of the models in the mentioned references also work on the hypersphere in latent space, the similitude ends there. We do not really consider them suitable for comparison because both the goals and methods of these works are very different from ours. In particular, those approaches are not concerned with the sparsity issue of HD spaces as well as volume compression as a possible solution to it. Davidson et al. (2018), for example (and related elaborations), build a VAE with a KLD term between a uniform distribution on the hypersphere and a von Mises-Fisher approximate posterior. Thus, by construction, there cannot be any compression of the type we discuss in our work, it is still highly similar to the standard VAE in the sense in which the posterior distribution ends approaching a uniform distribution on the hypersphere, where, we claim, the sparsity issue arises. The references using flexible priors are focused on the so-called 'prior hole problem' (mismatch between prior and the posterior). This can cause problems for the generation but is not related to our hypothesis: even with perfect match, the sparsity problem is present, since it belongs to the priors that are often used.

In the next section we formulate a VAE with the latent variables described using hyperspherical coordinates.

## 3 METHOD: VAE WITH HYPERSPHERICAL COORDINATES

Our approach is based on formulating the initial KLD divergence term with a prior from the original VAE, which is in Cartesian coordinates, to one in hyperspherical coordinates. See Appendix A.1 for the standard conversion formulas between Cartesian and hyperspherical in high dimension.

In Cartesian coordinates, the KLD divergence between the estimated posterior defined by $\mu_k$ and $\sigma_k$ and the prior defined by $\mu_k^p$ and $\sigma_k^p$ can be written as (see A.13):

$$\text{KLD}_{\text{CartCoords}}^{w/Prior} \approx \sum_{k=1}^{n} \left( (\mathbb{E}_b[\sigma_k] - \sigma_k^p)^2 + \sigma_b[\sigma_k]^2 + (\mathbb{E}_b[\mu_k] - \mu_k^p)^2 + \sigma_b[\mu_k]^2 \right) \quad (3)$$

where $\mathbb{E}_b$ and $\sigma_b$ denote the batch statistics over mini batches of data of size $N_b$.

So far, not much has been gained other than rewriting the KLD function (in Cartesian coordinates) in terms of the batch statistics. This rewriting was partly inspired by the construction in Bardes et al. (2021), and will be useful for our next step.

The similarity with the spin glass we described in section 1.3 is emerging from this expression. The terms of the form $(x - x_0)^2 = x^2 - 2xx_0 + x_0^2$ can be interpreted in the following manner: $x^2$ contributes to the homogeneous polynomial part of the energy function, $-2xx_0$ corresponds to an external magnetic field $2x_0$, while $x_0^2$ is a just an inconsequential constant term. The VAE is not exactly a spin glass though, since the reconstruction part of the loss, given by the standard MSE, is not a homogeneous polynomial. However, the analogy provides insight into the replica symmetry breaking as a way to move away from the equator, and how the application of external magnetic fields in the adequate manner can help achieve this, as we will see below, by applying these 'magnetic fields' in all the *angular directions* provided by hyperspherical coordinates.

We now introduce hyperspherical coordinates in the KLD formulation. We start with the Cartesian coordinates $(\mu_i, \sigma_i)$, given by the encoder, and transform these to their hyperspherical counterparts $(\overset{\mu}{r}, \overset{\mu}{\varphi}_k; \overset{\sigma}{r}, \overset{\sigma}{\varphi}_k)$ with $r$ a scalar and $k$ the index of the $n - 1$ spherical angles.

The KLD-like objective becomes for the angles $\varphi_k$:

$$
\begin{aligned}
\text{KLD}_{\text{HSphCoords}}^{w/Prior}(\varphi_k) = \sum_{k=1}^{n-1} &\left( \alpha_{\sigma,k} \left( \mathbb{E}_b[\cos \overset{\sigma}{\varphi}_k] - a_{\sigma,k} \right)^2 + \beta_{\sigma,k} \left( \sigma_b[\cos \overset{\sigma}{\varphi}_k] - b_{\sigma,k} \right)^2 \right. \\
&\left. + \alpha_{\mu,k} \left( \mathbb{E}_b[\cos \overset{\mu}{\varphi}_k] - a_{\mu,k} \right)^2 + \beta_{\mu,k} \left( \sigma_b[\cos \overset{\mu}{\varphi}_k] - b_{\mu,k} \right)^2 \right)
\end{aligned}
\tag{4}
$$

and for the norm $r$:

$$
\begin{aligned}
\text{KLD}_{\text{HSphCoords}}^{w/Prior}(r) = \alpha_{\sigma,r} \left( \mathbb{E}_b[\overset{\sigma}{r}] - a_{\sigma,r} \right)^2 + \beta_{\sigma,r} \left( \sigma_b[\overset{\sigma}{r}] - b_{\sigma,r} \right)^2 \\
+ \alpha_{\mu,r} \left( \mathbb{E}_b[\overset{\mu}{r}] - a_{\mu,r} \right)^2 + \beta_{\mu,r} \left( \sigma_b[\overset{\mu}{r}] - b_{\mu,r} \right)^2
\end{aligned}
\tag{5}
$$

with the priors for the mean over the batch $a_{i,j}$, the standard deviation over the batch $b_{i,j}$, and the gains for each term $\alpha_{i,j}$, $\beta_{i,j}$, for $i \in \{\sigma, \mu\}$ and $j \in \{1, ..., n-1, r\}$

We use the cosines rather than the angles to avoid costly extra computations of the corresponding arccosines (Appendix A.1). The reparameterization trick is still done in the Cartesian coordinates representation. The coordinate transformation is done using a vectorized implementation (code provided in Appendix A.2). The coordinate transformation and the extra KLD terms add about 32% computation time during training *per epoch* (measured at: 200 samples per batch, $n = 200$). For more dimensions the increase is higher (cf. A.16). The final cost to be optimized weights the reconstruction term and KLD terms with an overall gain $\beta$ for similarity with the standard $\beta$VAE (2):

$$
\mathcal{L} = \text{MSE}(x, \hat{x}_z) + \beta \left( \text{KLD}_{\text{HSphCoords}}^{w/Prior}(\varphi_k) + \text{KLD}_{\text{HSphCoords}}^{w/Prior}(r) \right)
\tag{6}
$$

### 3.1 Volume Compression of the latent manifold

We discussed previously that the standard VAE forces the latent samples to be uniformly distributed on the hypersphere, maximising the entropy, which results, in high dimensions, in the data being located within equators of the hypersphere where the volume is the greatest. A benefit of using hyperspherical coordinates is the possibility to set a prior for the $\varphi_k$ that forces the latent samples away from the equator, thereby escaping these highly entropic regions. This can be done for each angular coordinate, which are all uncorrelated with each other, by simply setting

$$
a_{\mu,k} \neq 0, \forall k.
\tag{7}
$$

By doing so, the samples can be moved to a zone with much less volume, thereby increasing the density of the latent, with the hope that random samples from that denser region will have better quality decoding because of the reduced sparsity. This can be seen more directly by analyzing the hypervolume element of the hypersphere in hyperspherical coordinates. The volume can be reduced

much faster and effectively by reducing the angular coordinates (away from the equators), than by either reducing just the radius of the hypersphere or, equivalently, all of the Cartesian coordinates.

The higher the dimension, the more pronounced this difference becomes because each added dimension $k$ adds extra powers of $\sin \varphi_k$ in the hypervolume element (Appendix A.3). Then, the further the angles from $\pi/2$, the smaller the infinitesimal hypervolume element becomes as it is multiplied by an increasingly smaller quantity lower than 1. This is a purely geometric effect. It can already be easily seen in the two-dimensional sphere, where a spherical coordinate rectangle of unvarying angular coordinates size has smaller area when moved away from the equator towards any of the poles. Thus, high dimensions bring the problem of high entropy in the equators, but also a non-Euclideanity to the manifold; we explored to which extent one can take advantage of the latter to mitigate the former.

Finally, by setting

$$a_{\mu,r} = \sqrt{n} \tag{8}$$

(and normalizing $z$, after sampling via the reparameterization trick, to the same radius $\sqrt{n}$) we can force the latent samples to be on the hyperspherical surface of that radius.

## 4 EXPERIMENTAL RESULTS

### 4.1 MODEL AND IMPLEMENTATION

For all our experiments, we use a ResNet-type architecture (He et al., 2015) for both encoder and decoder. When using the loss in hyperspherical coordinates (6), we use an annealing schedule (Fu et al., 2019) for the gain $\beta$ of the KLD-like loss, consisting of an initial stage which increases proportionally with $\sqrt{\text{epoch}}$ for the first 100 epochs, and is constant afterwards. This was necessary because we observed that too much compression of the volume was detrimental to the performance, while a strong compression was still necessary at the initial stage. The total training was 300 epochs in all cases.

### 4.2 CHOOSING THE GAIN FOR EACH LOSS TERMS

The constants $\alpha_{i,j}$, $\beta_{i,j}$ multiplying the elements of the hyperspherical loss are proportional to $1/\sqrt{k+1}$, where $k$ is the coordinate index. This was necessary because, unlike the Cartesian coordinates, the hyperspherical coordinates are asymmetric and vary with $k$. This can be seen in the transformation formulas (Appendix A.1), where a product of an increasing amount of sine functions is necessary as the coordinate index increases. We chose $1/\sqrt{k+1}$, guided by the fact that the vector whose Cartesian coordinates are $(1, 1, ..., 1)$ has a cosine of its spherical angles equal to $1/\sqrt{k+1}$ as the coordinate index, and because it gave the best results experimentally.

In this way, we were able to avoid lengthy calculations to obtain the mathematically exact formulas for both these constants and the KLD in hyperspherical coordinates, which we do not believe, anyway, to be of the most importance for the particular goals we had in this work.

### 4.3 VISUALISATION OF THE HIGH ENTROPY LATENT IN STANDARD VAE

A standard VAE with 128 latent dimensions was trained using the MNIST dataset (Y.LeCun et al, 1998). Generating and decoding random samples from the prior latent resulted in meaningless decoded/generated data (Fig.1a), left panel).

The latent hypersphere can be visualized in 3D as shown in Fig.1b), left. This was done by averaging the 128 latent dimensions into three (first 42, second 42, and the remaining 44), and normalizing each of the resulting 3D vectors to the sphere. Each latent vector could thus be plotted as a point in 3D, and shows a uniform-like distribution on the 2D sphere as expected.

*This visualisation allows us to directly see the entropy.* A k-NN classifier for the 10 classes of MNIST from the latent had an accuracy of 0.95, and 10 clusters can readily be seen when projecting

the 128 latent dimensions into 2 (Fig.1c), left) using t-SNE. However, no particular clustering can be observed on the 3D visualisation (Fig.1b), left). Such a direct visualisation of the latent space *cannot* display clusters, because they are buried into the 'disorder' of the entropy of the hypersphere equators, where most of the samples are located. There are many different possible microstates (points in latent space) that can realize the same macrostate (the clustering).

### 4.4 IMPROVED GENERATION WHEN THE LATENT MANIFOLD IS COMPRESSED

Next, we train the same VAE using the same dataset but now with the KLD-type loss in hyper-spherical coordinates (6). The prior was set to *compress* the (hyper-)volume in latent space by using $a_{\mu,k} = 1, \forall k$, which pushes *all* the $\varphi_k$ towards 0. They could not be exactly 0 because the reconstruction would all be the same. The reconstruction term balances the KLD term to spread the latent samples away from the angles 0. Recall from section 1.3 that, in the replica symmetry breaking phase, two replicas in that case lie in a very thin band or ring centered at a deep minimum of the energy function, center which is not in the initial high volume equatorial region; this ring is a $(n-2)$-hypersphere, submanifold of the initial $(n-1)$-hypersphere. This situation corresponds to sending the angle $\varphi_1$ towards 0, the rest free. Thus, by sending all the angles towards 0, and given the geometrical interpretation of the hyperspherical coordinates, we aim to induce a similar replica symmetry breaking transition also in the mentioned $(n-2)$-hypersphere, as well as in *all* the remaining sub $(n-k)$-hyperspheres, $\forall k$. We call this process 'Nested Replica Symmetry Breaking' (NRSB), and it is only in this regime where we get the results described below.

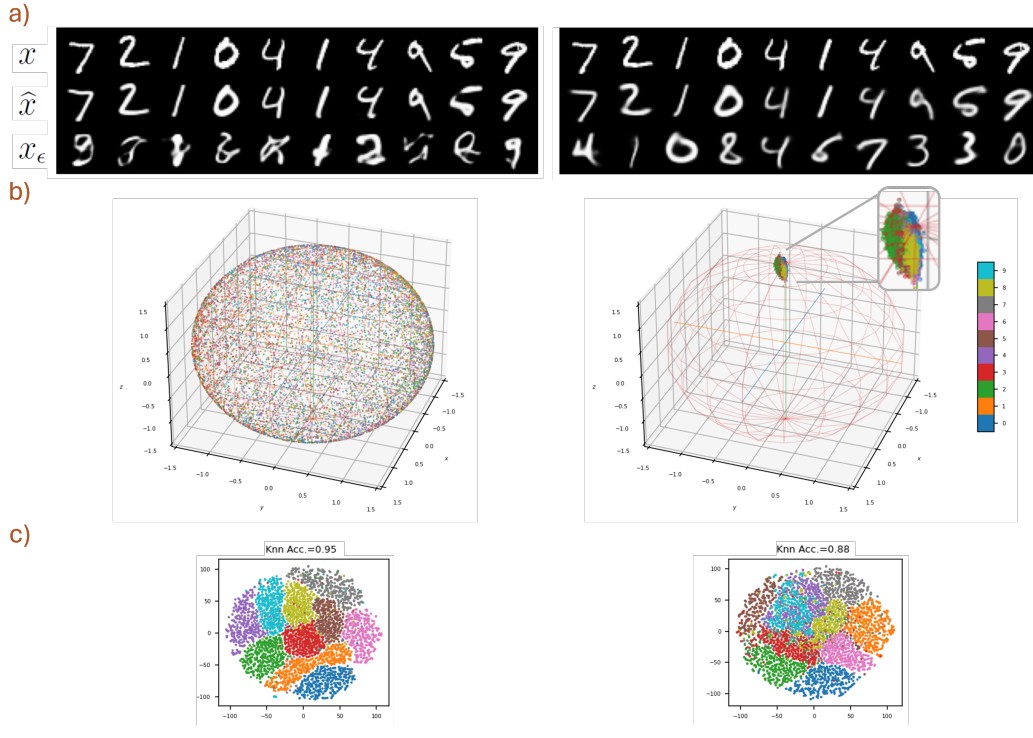

Figure 1: Comparison using MNIST between the standard $\beta$VAE (left) and the proposed compressed version (right). The top panel (a) shows the original data ($x$), the reconstruction ($\hat{x}$), and the generation sampling from the prior ($x_\epsilon$). The middle panel (b) shows the 3D projection on the latent 2D-sphere of the test dataset: the $\beta$VAE posterior is a uniform distribution whereas the proposed method compresses the latent vector on a small volume within an island of the hypersphere. The bottom panel (c) shows that in both cases the classes are clustered in the latent (using t-SNE) and that a k-NN classifier achieves good performance, with the compression $\beta$VAE resulting in lower accuracy (0.88 Vs 0.95) because the lower volume of the latent manifold forces the classes to overlap more (as seen on the clustering of panel c).

In this configuration, for generating new data the latent was not randomly sampled on the whole hypersphere, but from a von Mises–Fisher distribution with the same mean and covariance as the ones empirically calculated from the latent embedding of the full test dataset. These decoded random samples generated data with a quality close to the actual training dataset (Fig.1a), right panel), to be compared to the meaningless decoding of the previous experiment when random sampling from the prior was done (Fig.1a), left panel). By compressing the latent using hyperspherical coordinates, the VAE became a functional generative model, despite having 128 latent variables.

Furthermore, the $3-$dimensional visualization shows something remarkable (Fig.1b), right): besides showing that the latent samples are compressed towards a small 'island' on the hypersphere and away from the equator, the classes are actually *visible*. We believe that it is because the samples are now located away from the equator, in a region with a much lower entropy, where there are many fewer possibilities to realize this clustering in terms of different possible microstates. In other words, the VAE's latent space has made a phase transition. The same k-NN classifier from the latent space shows a similar accuracy, of $0.88$, and the class clusters can also be seen in a t-SNE 2D projection (Fig.1c), left).

## 4.5 Trade off between reconstruction and generation

The reconstruction quality of a VAE improves as the number of latent dimensions increases, as measured by the MSE between $x$ and $\hat{x}$. However, the quality of data generation (from decoding random sampling of the latent) decreases as the number of latent dimensions increases. We have argued in this paper that the later is due to the increased sparsity of the latent, as demonstrated qualitatively in the previous experiment when that sparsity is reduced by compressing the latent using our proposed method.

The quality of randomly generated data can be measured using the Frechet Inception Distance (FID) (Heusel et al., 2017). FID compares the distribution of features between the images of the training/testing dataset and an equivalent number of randomly generated images. We used in this experiment CIFAR10 (Krizhevsky, 2009), a more challenging dataset, and an FID computed using 10,000 samples (we compare the random decoded samples with the *reconstructed* testing set). We call this way of measuring the generation as 'self-FID'.

In a VAE, the quality of the reconstruction, still measured by the MSE, also varies with the gain $\beta$ of the loss: the more weight for the KLD term, the more the latent matches the prior and the worse the reconstruction (Cf. $\beta$VAE Higgins et al. (2017)). We show in Appendix A.5 this behaviour by comparing the results for several values of $\beta$.

We can now explore quantitatively the quality of the reconstruction (using MSE) and the quality of the generation (using self-FID) when the number of latent dimensions increases and the $\beta$ varies. We compared the standard VAE with our proposed compressed VAE.

These two metrics (MSE and self-FID), should give us a good idea regarding how good our models are for the general generative task: the fist measures how 'crisp/sharp' (i.e., not 'blurry') the reconstructed images are, while the second how close the random decoded images resemble images from the (reconstructed) training dataset. *A good VAE-based generative model should minimize both of these metrics* **simultaneously***: that is, to be able to generate random samples which are* **in-distribution** *wrt the reconstructed dataset (low self-FID), and such that the latter actually resembles the original dataset (low MSE)*. See appendices A.6 and A.14 for gaining intuition about the relation between the qualitative look of the decoded images and the corresponding MSE and self-FID values for those models.

Fig.2 sumarizes the results with more details provided in Appendix A.5. As expected, the MSE decreases as one increases the latent space dimension, while the exact opposite is true for the self-FID. To obtain a good generative model in the way we defined it can be a very difficult task, involving a very delicate balance between these two opposing trends we just described, and often relying on off-equilibrium configurations.

The results show that the compression VAE version improves on absolute terms over the standard VAE over any combination of $\beta$ and dimension of the latent.

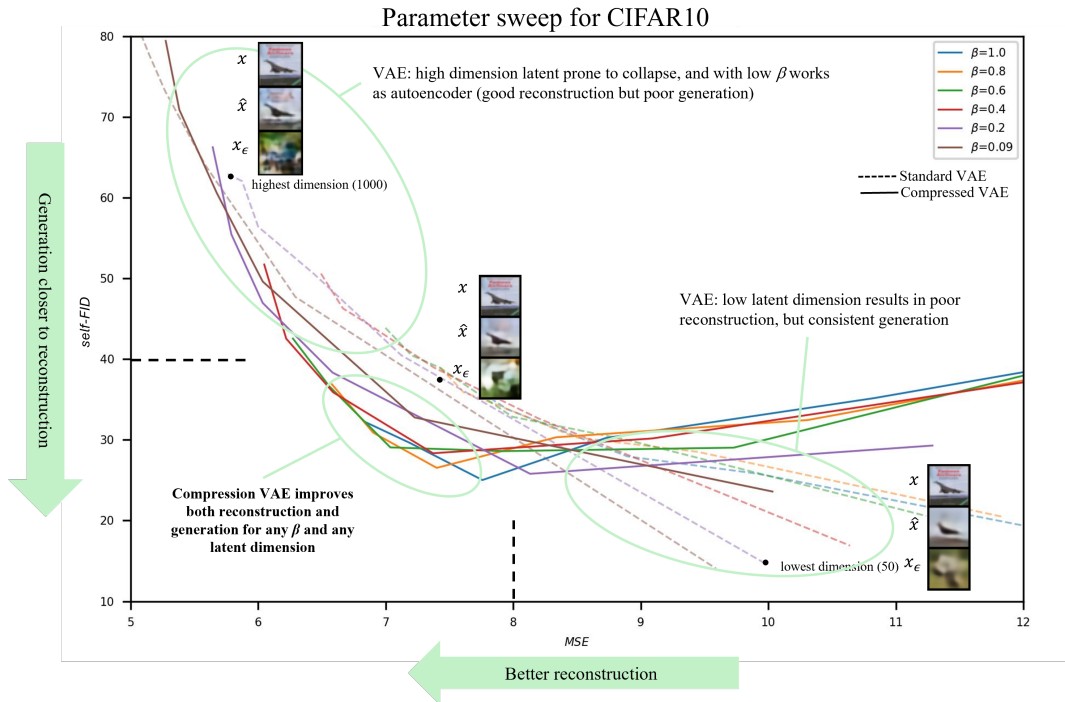

Figure 2: Effect of latent dimension and $\beta$ on the trade off between reconstruction and generation on CIFAR10. Each curve represents a VAE for a given $\beta$ while spanning the number of latent dimensions from low (50, bottom right endpoints) to high (1000 top left corner endpoints). The standard VAEs are shown using dashed lines, whereas the compressed versions are shown using solid lines. We excluded from our discussion the regimes where generation was of very poor quality (self-FID$> 40$) or the reconstruction was too blurry (MSE$> 8$), with the best trade off close to the bottom left corner. In that useful area, the compressed VAEs outperformed their standard equivalent for any combination of $\beta$ and latent size (solid lines closer to the bottom left corner than the dashed lines).

## 5 CONCLUSION

We propose to convert the latent variables of a VAE to hyperspherical coordinates. This allows to move the latent vectors on a small island of the hypersphere, reducing sparsity. We showed that this improves the generation quality of a VAE. The following points will require further attention in regards to the present work:

- the improvement in generation was only evaluated for the purposes of hypothesis testing, and not as absolute performance. Furthermore, the FID metric that we used may also have limitations sometimes (Stein et al., 2023).

- we did not evaluate the method for high resolution and larger datasets such as Imagenet.

- the extra computing time is about 32 per cent more per epoch for 200 latent dimensions. In (much) higher dimensions, the added computation increases and might become prohibitive.

- future research can focus in optimizing this method (or other method that takes into account the hypothesis about sparsity) for obtaining state-of-the-art results in generation and other tasks, in VAEs and other models.

- the use of latent representations in hyperspherical coordinates can also be further explored in several other applications (perhaps unrelated to compression and generation), by the use of the provided script for the conversion and inspired by its proof of concept of practical feasibility in the present paper.

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
