## A APPENDIX

### A.1 CONVERSION BETWEEN CARTESIAN AND HYPERSPHERICAL COORDINATES

For reference, we note here the standard formulas for converting between cartesian and spherical coordinates (as they appear in https://en.wikipedia.org/wiki/N-sphere).

In $n$ dimensions, given a set of cartesian coordinates $x_k$ with $k \in \{1, \ldots, n\}$, the hyperspherical coordinates are defined by a radius $r$ and $n-1$ angles $\varphi_k$ with $k \in \{1, \ldots, n-1\}$; $\varphi_k \in [0, \ldots, \pi]$ for $k \in \{1, \ldots, n-2\}$ and $\varphi_{n-1} \in [0, \ldots, 2\pi)$.

From hyperspherical to cartesian conversion:

$$
\begin{aligned}
x_1 =& r\cos(\varphi_1) \\
x_2 =& r\sin(\varphi_1)\cos(\varphi_2) \\
x_2 =& r\sin(\varphi_1)\sin(\varphi_2)\cos(\varphi_3) \\
&\vdots \\
x_{n-1} =& r\sin(\varphi_1)\sin(\varphi_2)\ldots\sin(\varphi_{n-2})\cos(\varphi_{n-1}) \\
x_n =& r\sin(\varphi_1)\sin(\varphi_2)\ldots\sin(\varphi_{n-2})\sin(\varphi_{n-1})
\end{aligned}
\tag{9}
$$

From cartesian to hyperspherical conversion:

$$
\begin{aligned}
r =& \sqrt{x_n^2 + x_{n-1}^2 + \ldots + x_2^2 + x_1^2} \\
\cos(\varphi_1) =& \frac{x_1}{\sqrt{x_n^2 + x_{n-1}^2 + \ldots + x_2^2 + x_1^2}} \\
\cos(\varphi_2) =& \frac{x_2}{\sqrt{x_n^2 + x_{n-1}^2 + \ldots + x_2^2}} \\
&\vdots \\
\cos(\varphi_{n-2}) =& \frac{x_{n-2}}{\sqrt{x_n^2 + x_{n-1}^2 + x_{n-2}^2}} \\
\cos(\varphi_{n-1}) =& \frac{x_{n-1}}{\sqrt{x_n^2 + x_{n-1}^2}}
\end{aligned}
\tag{10}
$$

### A.2 VECTORIZED CODE FOR CONVERTING BETWEEN CARTESIAN AND HYPERSPHERICAL COORDINATES

This code is accessible here and provided below for reference.

```python
import torch

def r (x):

    r = torch.linalg.norm (x, dim=1)

    return r

def cart_to_cos_sph (x, device):

    m = x.size(0)
```

```
756     n = x.size(1)
757
758     mask = torch.triu(torch.ones(n, n)).to(device)
759
760     mask = torch.unsqueeze(mask, dim=0)
761
762     mask = mask.expand(m, n, n)
763     X = torch.unsqueeze(x, dim=1).expand(m, n, n)
764
765     X_squared = torch.square(X)
766
767     X_squared_masked = X_squared * mask
768     denom = torch.sqrt(torch.sum(X_squared_masked, dim=2)+0.001)
769
770     cos_phi = x / denom
771
772     return cos_phi[:, 0:n-1]
773
774  def cart_to_sin_sph (x, device):
775
776     return torch.sqrt (1 - cart_to_cos_sph (x, device).pow(2))
777
778  def cart_to_sph (x, device):
779
780     m = x.size(0)
781
782     n = x.size(1)
783     mask = torch.triu(torch.ones(n, n)).to(device)
784
785     mask = torch.unsqueeze(mask, dim=0)
786
787     mask = mask.expand(m, n, n)
788     X = torch.unsqueeze(x, dim=1).expand(m, n, n)
789
790     X_squared = torch.square(X)
791
792     X_squared_masked = X_squared * mask
793     denom = torch.sqrt(torch.sum(X_squared_masked, dim=2)+0.001)
794
795     phi_plus = torch.arccos (x / denom)
796
797     phi_minus = 2*3.141592654 - phi_plus
798     phi = phi_plus
799
800     phi[:, n-2] = torch.where (x[:, n-1] >= 0, phi_plus[:, n-2],
801         phi_minus[:, n-2])
802     return phi[:, 0:n-1]
803
804
805  def sph_to_cart (R, phi, device):
806
807     m = phi.size(0)
808     n = phi.size(1)+1
809
       mask = torch.tril(torch.ones(n-1, n-1)).to(device)
```

```
      mask = torch.unsqueeze(mask, dim=0)

      mask = mask.expand(m, n-1, n-1)

      PHI = torch.unsqueeze(phi, dim=1).expand(m, n-1, n-1)

      sin_PHI = torch.sin(PHI)

      mask_ = torch.unsqueeze(torch.triu(torch.ones(n-1, n-1),
          diagonal=1).to(device), dim=0).expand(m, n-1, n-1)

      sin_PHI_masked = sin_PHI * mask + mask_

      sin_prod = torch.prod (sin_PHI_masked, dim=2)

      ones = torch.ones(m).to(device)

      sin_PROD = torch.column_stack((ones, sin_prod))

      cos_R = torch.mul (torch.column_stack((torch.cos (phi), ones)),
          torch.unsqueeze(R, dim=1))

      x = torch.mul (sin_PROD, cos_R)

      return x
```

### A.3 HYPERVOLUME ELEMENT IN HYPERSPHERICAL COORDINATES

The hypervolume element of the hypersphere $\mathbb{S}_R^{n-1}$ is given by the following expression when using hyperspherical coordinates (see https://en.wikipedia.org/wiki/N-sphere):

$$\mathrm{d}V_{\mathbb{S}_R^{n-1}} = R^{n-1} \sin^{n-2}(\varphi_1) \sin^{n-3}(\varphi_2) \cdots \sin(\varphi_{n-2}) \mathrm{d}\varphi_1 \mathrm{d}\varphi_2 \cdots \mathrm{d}\varphi_{n-1} \tag{11}$$

In the small angle regime, where $\sin \varphi \approx \varphi$, we can approximately integrate this expression for an angular coordinate hypercube $[0, \varphi_0]^{n-1}$, and the result is proportional to $v_0 = R^{n-1} \varphi_0^{n(n-1)/2}$. If now we reduce the size of the angular coordinate hypercube by a schedule of the form $\varphi_t = \varphi_0(1 - t)$, $t \in [0, 1]$, then we can compare the percentage of hypervolume being reduced from the initial value, while keeping $R$ fixed, to the percentage obtained by reducing the size of the hypersphere by an schedule of the form $R_t = R(1 - t)$, $t \in [0, 1]$, while keeping $\varphi_0$ fixed (this second case is equivalent to reducing all the Cartesian coordinates at once, because $r^2 = x_n^2 + x_{n-1}^2 + \ldots + x_2^2 + x_1^2$). Indeed, we get, respectively, $v_t = v_0(1 - t)^{n(n-1)/2}$ and $v_t = v_0(1 - t)^{n-1}$. In Fig. 3 we plot the behavior of $v_t/v_0$ in terms of the reduction of the coordinate, given by $(1 - t)$, for three, increasing values of dimension $n$. As we can see, already in dimension 20 (bottom figure in the panel), there is a sharp decrease in volume in the angular case as soon as one decreases the angular coordinates by a minimal amount; in comparison to the radial/Cartesian coordinate case, the abrupt decrease in volume looks almost discontinuous.

### A.4 CONCENTRATION OF MEASURE EFFECTS

In this appendix, we collect the results of simple experiments that clearly show the concentration of measure effects that occur in high dimensions. In Fig. 4a), we show the distribution of a simple Normal distribution in 2 dimensions (left), and the histogram for the norm of the samples (right). In b), the same but for a Normal distribution in 100 dimensions. In Fig. 5a), we show the histogram for the angle between two random samples from a Normal distribution in 2 dimensions (left), and the same but for a Normal distribution in 100 dimensions (right). In b), we display a schematic diagram of the mass concentration of the uniform measure of the hypersphere in very high dimensions. The intuition in this diagram comes from the more precise result (Wainwright, 2019) which states that, for *any* given $y \in \mathbb{R}^n$, if we define on the hypersphere an 'equatorial' slice of width $\epsilon > 0$ as

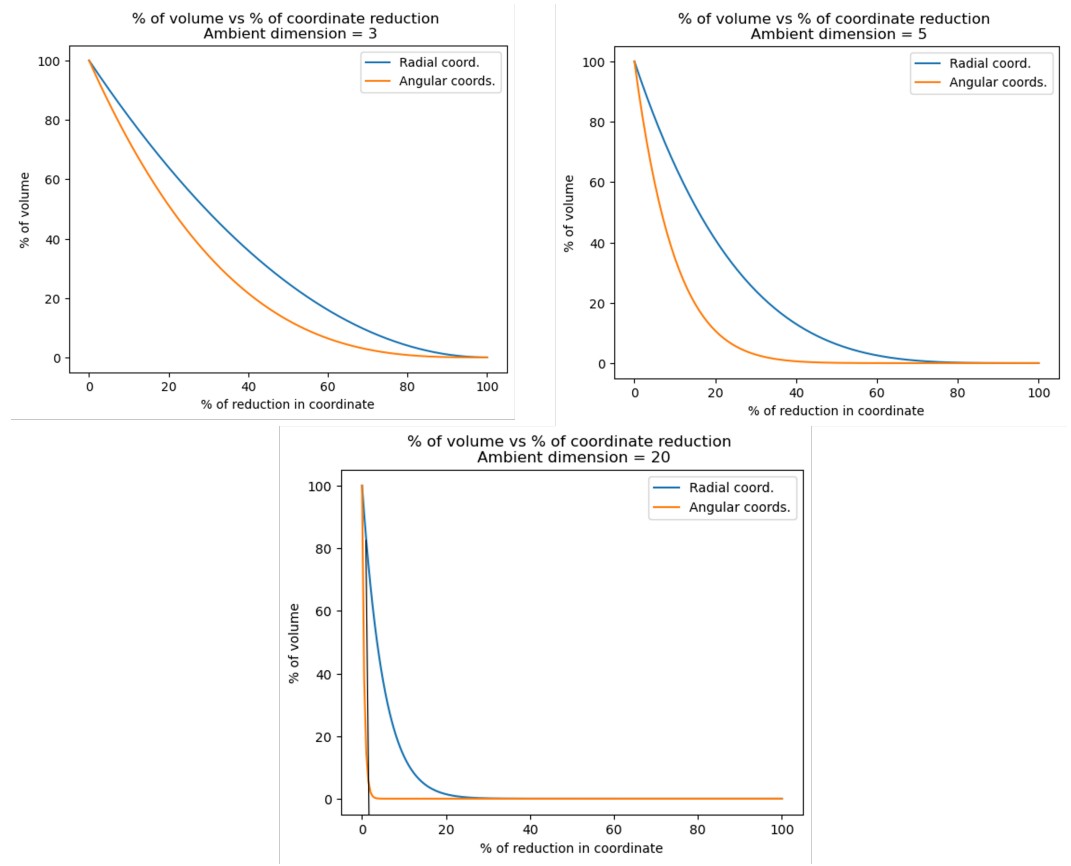

Figure 3: Hypervolume element reduction comparison: $(1-t)^{n-1}$ vs. $(1-t)^{n(n-1)/2}$.

$T_y(\epsilon) \doteq \left\{ z \in \mathbb{S}^{n-1}/ \mid (z,y) \mid \leq \epsilon/2 \right\}$, then its volume according to the uniform measure satisfies the following concentration inequality:

$$\mathbb{P}\left[T_y(\epsilon)\right] \geq 1 - \sqrt{2\pi}\exp(-\frac{n\epsilon^2}{2}). \tag{12}$$

The previous inequality shows that, in very high dimensions, the equatorial slice $T_y(\epsilon)$ occupies a huge portion of the total volume, even for a very small width.

Finally, with this in place, we can understand the peculiar shape that a high dimensional Normal distribution takes when expressed in hyperspherical coordinates (Fig.6).

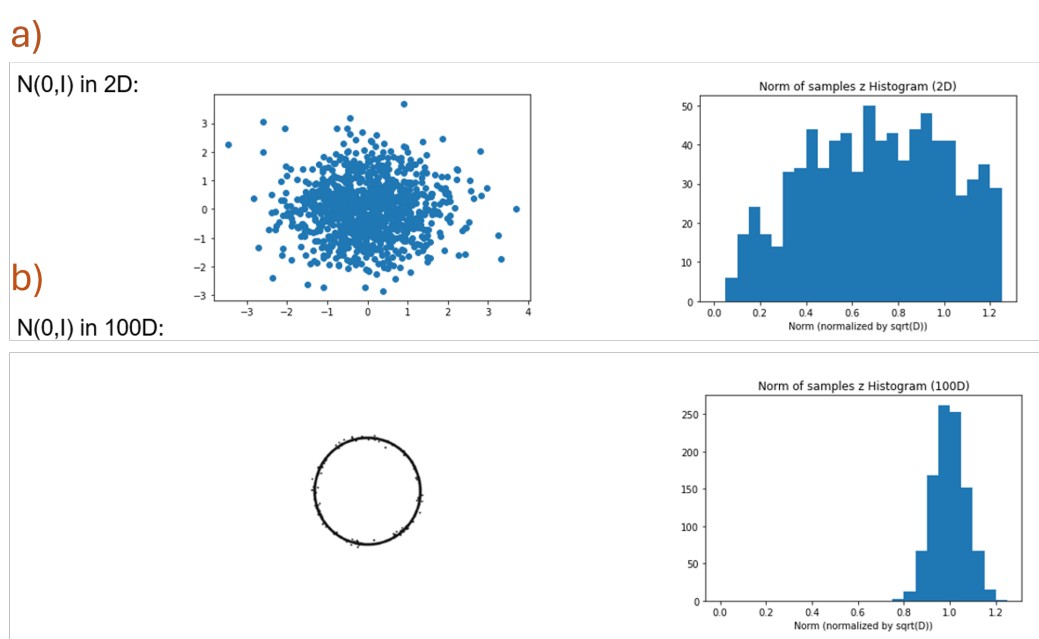

Figure 4: Measure concentration, norm (left image in b), adapted from Vershynin (2018))

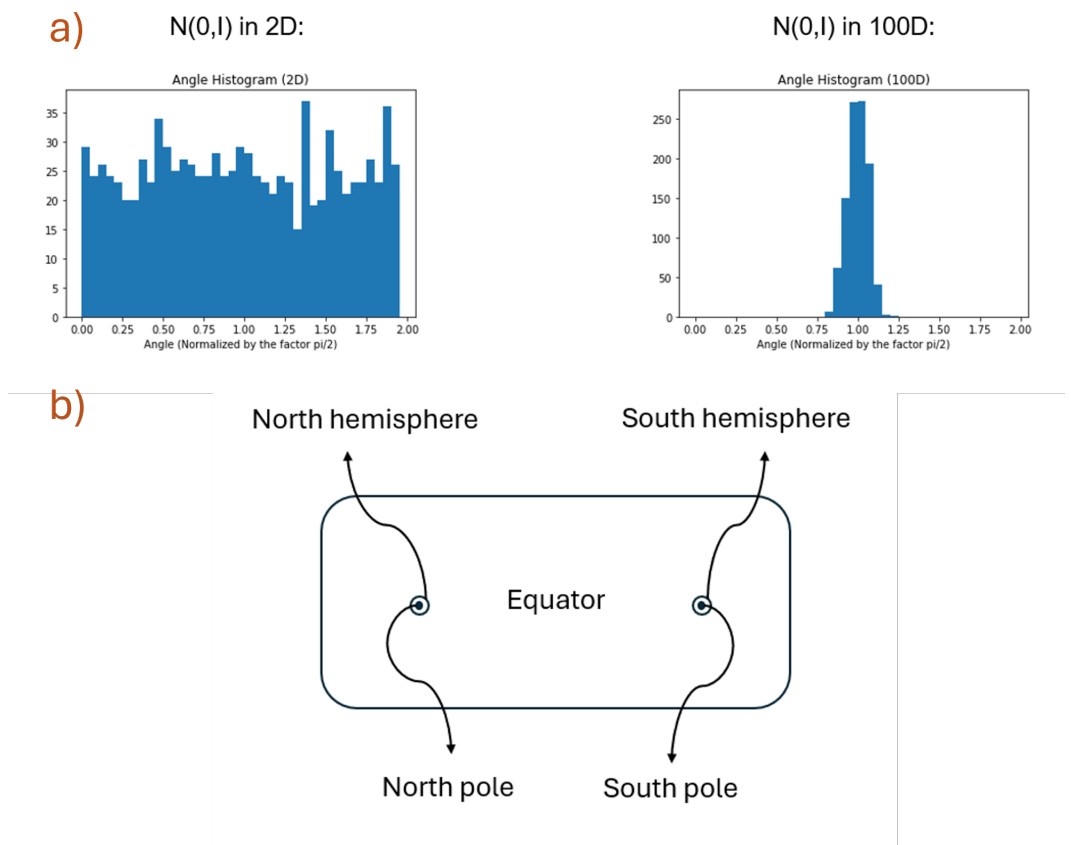

Figure 5: a) Measure concentration, angle; b) Schematic diagram of the mass concentration of the uniform measure of the hypersphere in very high dimensions: most of the volume is in the equator.

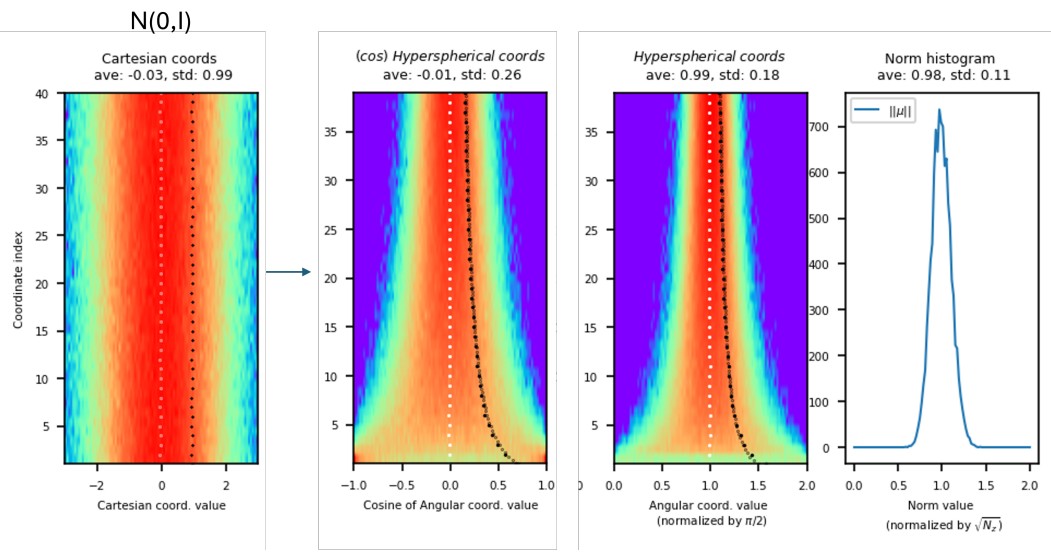

Figure 6: High dimensional Normal distribution in hyperspherical coordinates. For the first three images from the left, each horizontal slice at some vertical index value shows the color coded histogram (red, high density; blue, low density) for the range of the coordinate of that index; the vertical axis stacks all the histograms for all the dimensions (in this example, 40). The white dots represent the mean and the black dots represent the standard deviation of the corresponding histogram. The numbers on top are the total mean and standard deviation of all these previous values taken together.

## A.5 ADDITIONAL ANALYSIS OF CIFAR10 RESULTS

Here, we continue the analysis of the experimental results that we obtained for CIFAR10. Fig.7 shows the same results as Fig.2, but we now make a more detailed breakdown of the dependencies of both the MSE and self-FID wrt both the number of latent space dimensions and the total gain $\beta$ (as in Fig.2, solid lines correspond to the compression model, while dashed lines to the standard one). In the standard VAE, for a fixed $\beta$, as we increase the latent dimension, the self-FID increases (worse generation), but the MSE decreases (more sharp, less blurry images); for a fixed latent dimension, as we increase $\beta$, the self-FID decreases (better generation), but the MSE increases (less sharp, more blurry images).

Fig.8 shows the typical training of a standard VAE in one of our experimental rounds. In the upper panel we show, from left to right, the histograms of $\mu$, $\sigma$, and $z$, respectively, using the same conventions as in Fig.6. The fourth histogram in this panel shows the norm histograms of $\mu$ and $z$, as well as the 'replica angle' (dashed red lines) between the testing samples and the mean for all the test set (this value should give an idea about the angular size of the island as well as to signal if there is an overall replica symmetry breaking in our model; in this particular example, there is no such phase transition, since the mean value of the replica angle is close to $\pi/2$). The second, middle panel shows the behavior of the MSE and KLD loses during training for the test set. The bottom panel corresponds to the histogram of the cosine of the hyperspherical coordinates of $\mu$ (cf. Fig.6).

Fig.9 shows the typical training of our compression VAE in one of our experimental rounds. The conventions are the same as in Fig.8. Of note is that the replica angle value in this case shows the desired phase transition. The middle panel shows the annealing schedule used for training our model. Finally, we can see how in the histogram of the cosine of the hyperspherical coordinates all of them are shifted towards a cosine value of $1$, which corresponds to an angle equal to $0$, as expected.

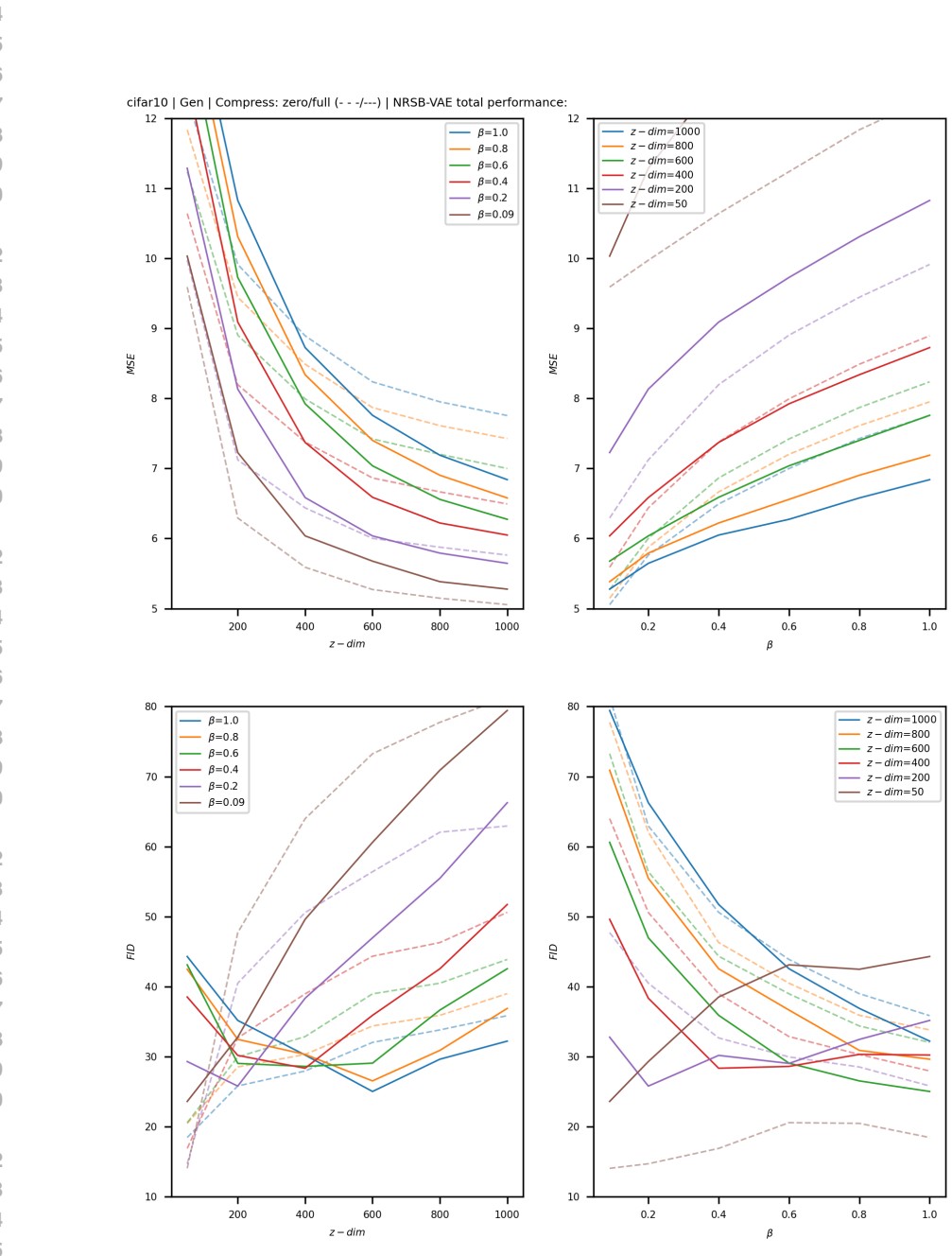

Figure 7: CIFAR10 results breakdown. MSE and self-FID in terms of both the number of latent space dimensions and the total gain $\beta$ (cf. Fig.2).

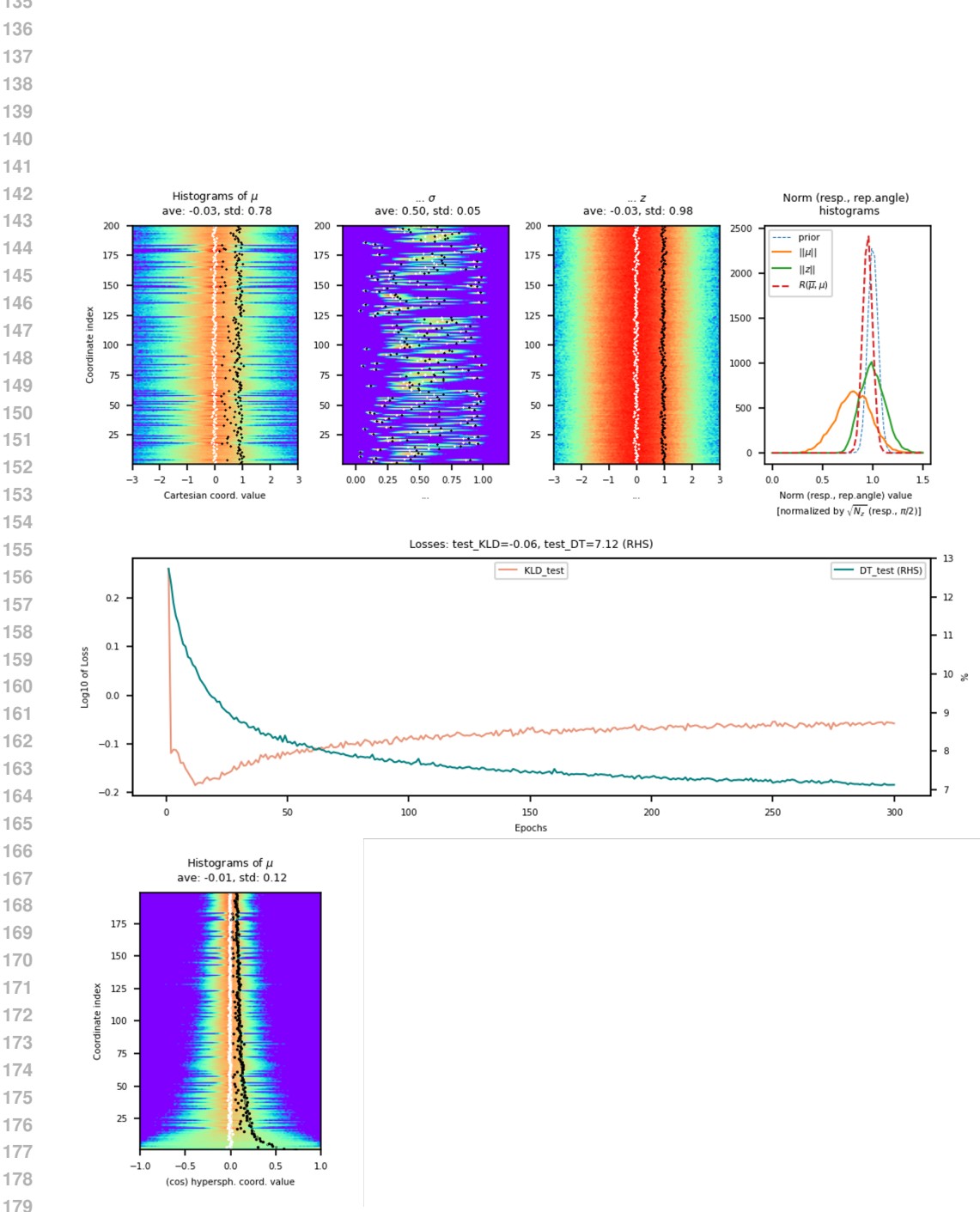

Figure 8: Results of standard VAE training with a balanced $\beta$.

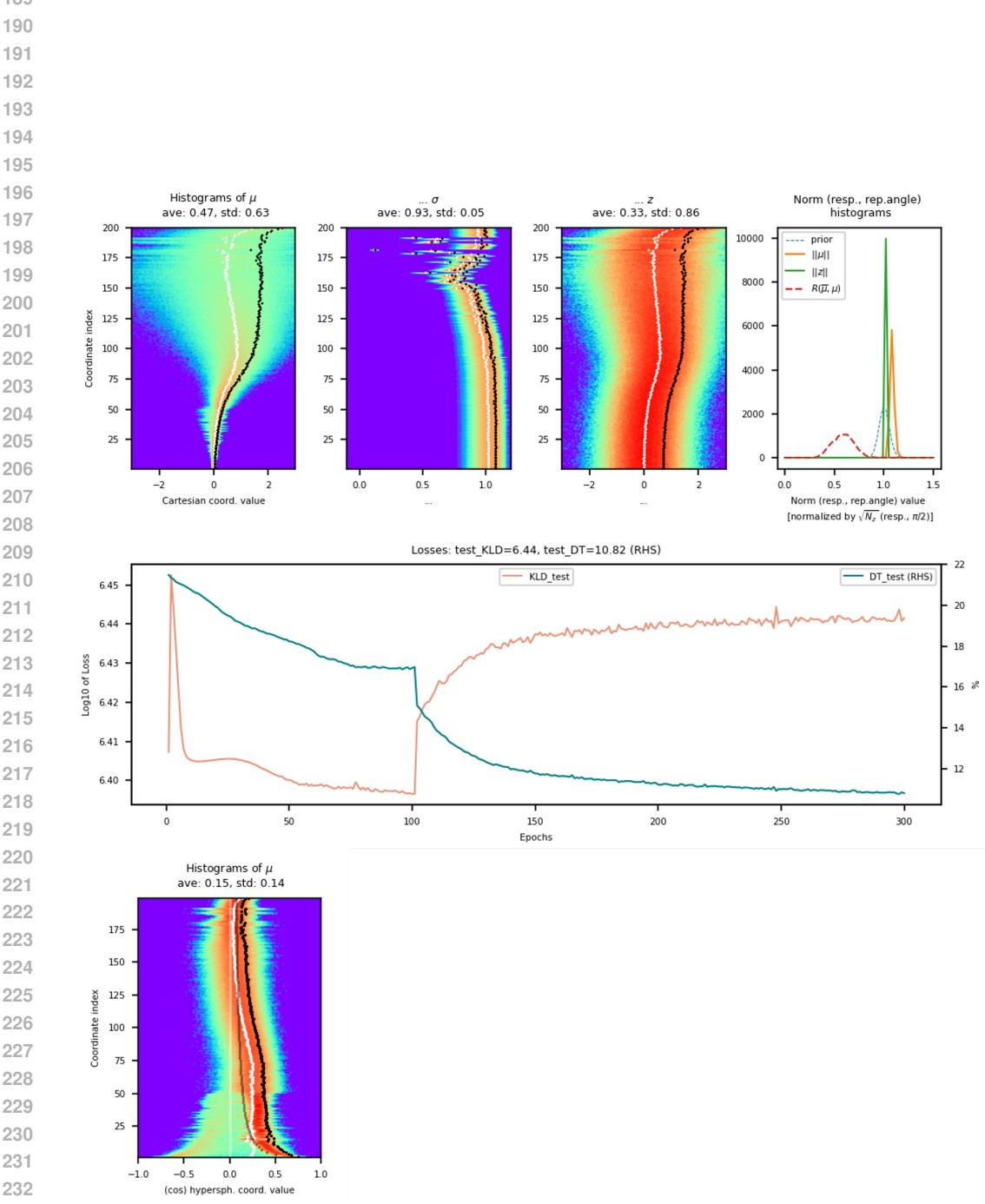

Figure 9: Results of a compressed VAE training.

## A.6 THE DIFFERENT REGIMES OF THE STANDARD $\beta$VAE IN HD

In this appendix, we illustrate with several examples from our experiments the different regimes in which a $\beta$VAE can operate according to the value of the parameter $\beta$, while maintaining the dimension of the latent space fixed but high enough. This is important, since it is known (see, e.g., Cinelli et al. (2021), section 5.5.2) that HD VAEs are prone to exhibit a phenomenon known as posterior collapse when $\beta$ is too high: "[...] [a] state where the variational posterior and true model posterior collapse to the prior, the posterior encodes no information about the input $x$, and no useful latent representation was learned" (quoted from the mentioned reference). This of course, is a problem, since the collapsed latent dimensions become inoperative for the model and in-utilizable for other tasks. Furthermore, if used, they can introduce errors in those analysis.

In practice, a simple solution to avoid this issue that often works is to simply reduce the value of $\beta$, which acts as a gain for the KLD term in the VAE loss function. One can check for any collapse by inspecting the histograms of the means $\mu$ of the latent encoding and making sure that the standard deviation (std) there is appreciably away from zero for each latent dimension. A threshold value can be implemented, but we will keep the discussion qualitative in that aspect.

In Fig.10 we show a standard VAE trained with a high $\beta$ ($= 1.00$) in HD ($n = 200$), it has more than half of its dimensions collapsed yet the self-FID remains the lowest for the examples (for the standard VAE, that is) in this dimension as we decrease the $\beta$ (cf. Fig.7, second row, right; this is the case for all the dimensions we checked except the lowest, $n = 50$; see A.12 for this latter case). Thus, posterior collapse here acts as an effective dimensional reduction mechanism for the generation, since the collapse actually improves the self-FID profile (we believe that what happens here is that the weights of the network corresponding to these dimensions are inactive or close to 0 and, therefore, the decoder simply ignores the dimensions in question; see also Dai et al. (2018); Rolínek et al. (2019)). Nevertheless, since many dimensions are ignored, the model's latent space lacks representation capacity, which translates into poor reconstructions ($MSE = 9.92$): the model works similarly to a non-collapsed one with a much more lower latent dimension.

In Fig.11 we show a standard VAE trained with a medium/balanced $\beta$ ($= 0.20$). In this case, there are more functional dimensions than collapsed or almost collapsed ones. Thus, the model has more representation capacity and this is reflected in a lower reconstruction error ($MSE = 7.12$). Nevertheless, since the decoder now actually operates with a much higher number of dimensions, then the sparsity and high hypervolume of HD spaces becomes an issue, and this is reflected in a worse generative performance (higher self-FID than the previous case). In Fig.12, we show a standard VAE trained with a low $\beta$ ($= 0.09$) VAE. In this example, the mentioned trends continue and intensify, now with a much better reconstruction ($MSE = 6.32$), but very poor generation.

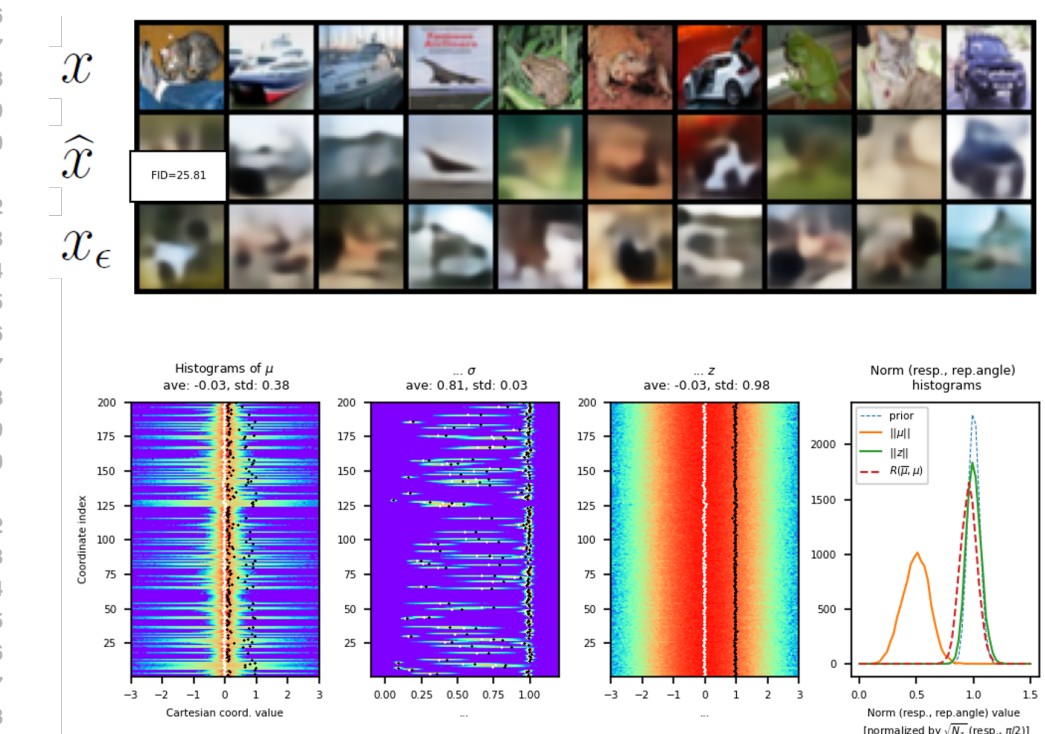

Figure 10: Results of a high $\beta$ (= 1.00) VAE training ($MSE = 9.92$, poor). Notice the collapsed dimensions in the histograms for $\mu$ (the variance, black dots, for each of those dimensions is very close to 0). Good generation.

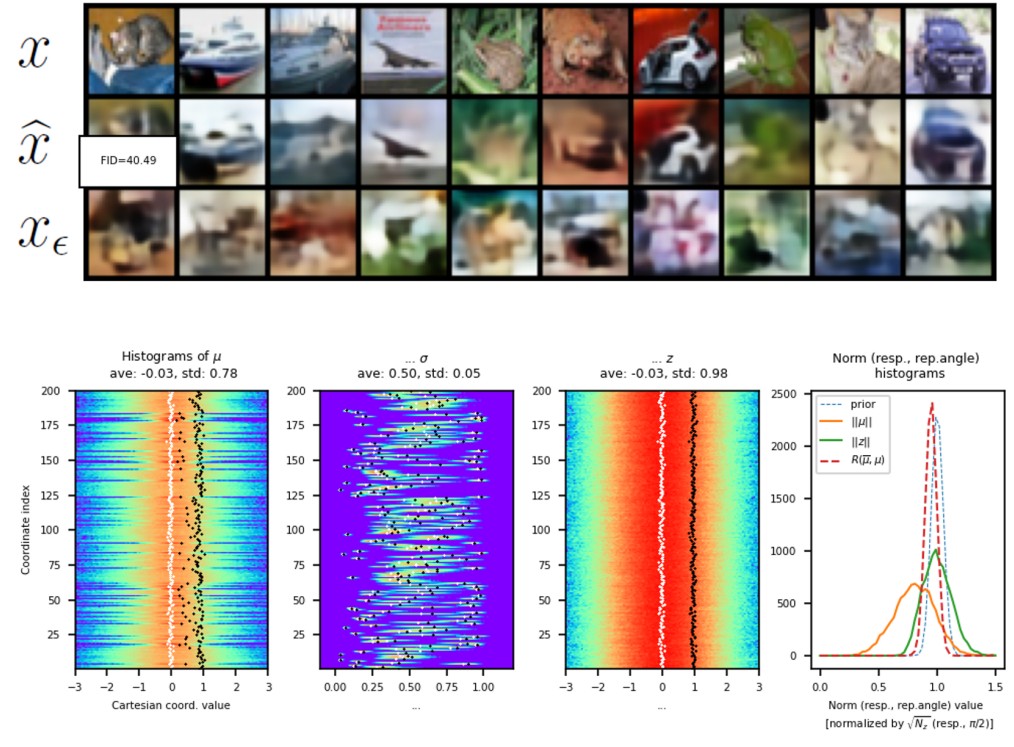

Figure 11: Results of a medium/balanced $\beta$ (= 0.20) VAE training ($MSE = 7.12$, regular). There are more functional dimensions than collapsed or almost collapsed ones. Regular generation.

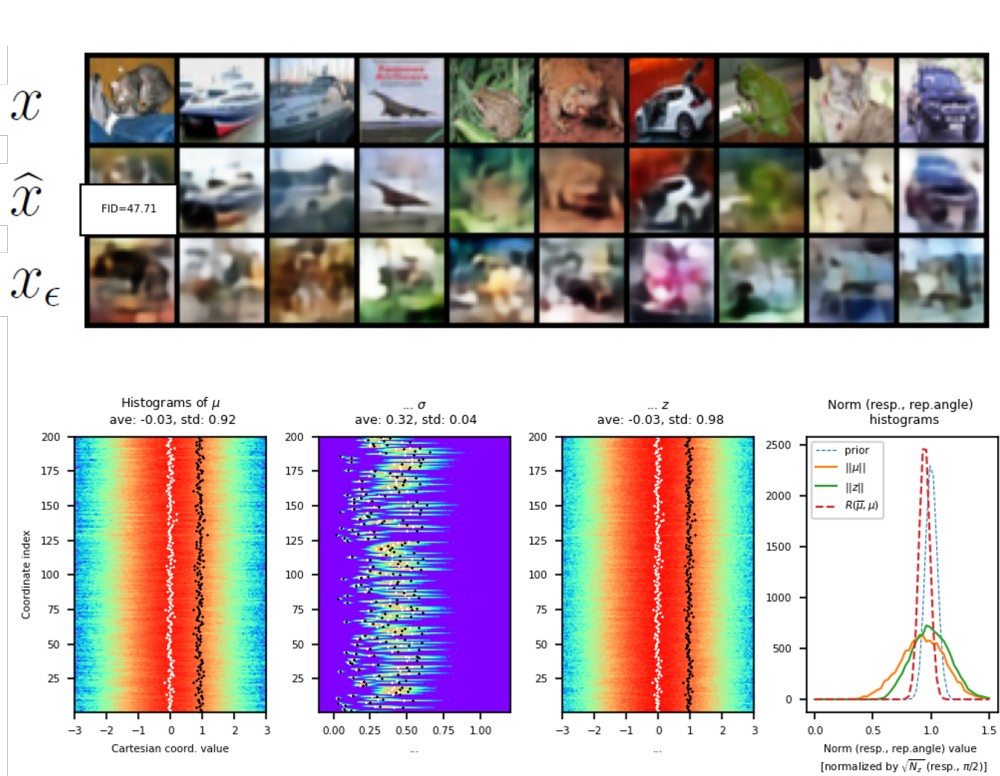

Figure 12: Results of a low $\beta$ $(= 0.09)$ VAE training ($MSE = 6.32$, good). There are no collapsed dimensions, but the model becomes almost an autoencoder (i.e., the VAE's $\sigma$ is close to 0). Bad generation.

## A.7 AVOIDING POSTERIOR COLLAPSE IS NOT ENOUGH TO IMPROVE GENERATION IN A HD VAE

In this appendix, we show an example in which we encourage the mean of the radial coordinate $\overset{\mu}{r}$ of the encoded means $\mu$ to lie on the hypersphere of radius $\sqrt{n}$, i.e., $a_{\mu,r} = \sqrt{n}$, and the means of the (cosine) hyperspherical angles $\overset{\mu}{\phi}_k$ to lie in the equators, i.e., $a_{\mu,k} = 0$, $\forall k$; furthermore, we also balance the variance of the (cosine) angles $\overset{\mu}{\varphi}_k$ by encouraging it to be in the same direction as the vector whose Cartesian coordinates are $(1,\dots,1)$, i.e., $b_{\mu,k} = 1/\sqrt{k+1}$, $\forall k$. With this setup, our experiments show that posterior collapse is avoided (in both the Cartesian and hyperspherical coordinates representations), while the distribution of $\mu$ is still similar to a uniform distribution on the hypersphere, like in the standard VAE (cf. Bardes et al. (2021)). Nevertheless, as expected from the discussion in the previous section, this is not enough to guarantee good generation (Fig.13).

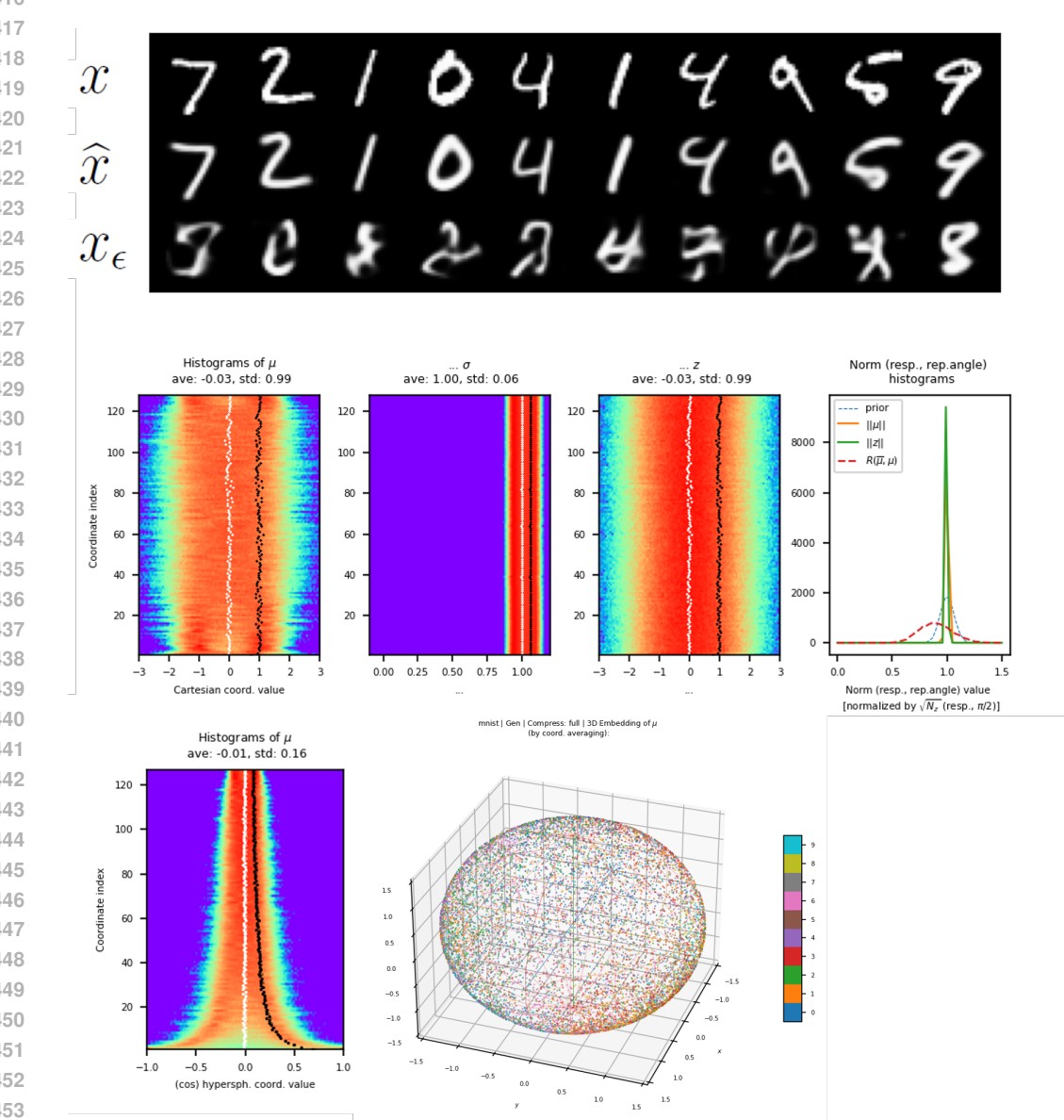

Figure 13: Results of a non-collapsed, non-compressed VAE training. We repeated the experiments for several target values for $\sigma$ and $\beta$, but the results were qualitatively the same as in the present figure.

### A.8 HIGH HYPERVOLUME COMPRESSION REDUCES SPARSITY AND IMPROVES GENERATION IN HD VAEs

Continuing the analysis of the previous section and figure, then, now that we are sure that we don't have any collapsed latent dimensions and thus are using the full representation capacity of the HD space, we can try to improve the poor generation. If our hypothesis about the sparsity introduced by the exponentially (wrt the dimension) diverging hypervolume in the equators being the root cause of this issue is true, then by implementing our compression via hyperspherical coordinates we should be able to improve this generation while remaining on the hyphersphere, un-collapsed and thus retaining the full expressive capacity of the HD space (unless we compress too much and the excessive overlap hinders the reconstruction).

In Fig.14 we start with a moderate amount of compression by encouraging the mean of the (cosine) angles $\overset{\mu}{\phi}_k$ to be in the same direction as the vector whose Cartesian coordinates are $(1, \ldots, 1)$, i.e., $a_{\mu,k} = 1/\sqrt{k+1}$, $\forall k$. Indeed, recall from appendix A.3 that the closer we get to the north pole, the lower the volume. Nevertheless, this moderate compression is not enough to significantly improve the generation. Thus, in Fig.15 we go to full compression mode by setting $a_{\mu,k} = 1$, $\forall k$, which encourages all the points to converge and condense at the north pole. it is only in this regime of very high compression that we get a significantly appreciably improvement in the generation. Furthermore, we consider this a direct proof of our hypothesis regarding the sparsity of HD spaces and their impact on generation. In Fig.2 of the main text we showed our experimental results for the more challenging dataset CIFAR10 regarding how we can use this to systematically take advantage of the better representation capacity of un-collapsed HD latent spaces to maintain a good and stable reconstruction, while we use our method of volume compression to improve at the same time the quality of the generation. This allowed us to reach more valuable zones of the MSE-self-FID plane which are not accessible via the standard VAE in any combination of the parameters $n$ (latent dimension) and $\beta$.

As an additional comment, by looking at the histogram for $\mu$ in Cartesian coordinates in Fig.15, one may think that the lower (in coordinate index) latent dimensions seem heavily collapsed. But this is not the case: the latent data distribution lies exactly on the hypersphere, and this forces correlations in the Cartesian coordinates, reason by which the fact that one or many more Cartesian coordinates (and their variance) are close to 0 is not conclusive of the irrelevance of many of the latent dimensions; indeed, if we now check the histogram for the (cosine) angles $\overset{\mu}{\phi}_k$ in hyperspherical coordinates (which are a set of uncorrelated coordinates on the hypersphere, by construction), then we see that there is no collapse in any dimension there. Adding to this point, we can see in Fig.8 that, in the standard VAE, the collapse in Cartesian coordinates (e.g., around index 20 in the first histogram to the left in the first row) translates into a collapse in the (cos) hyperspherical coordinates (third row histogram, same index), while this is not the case in our compression VAE in Fig.9, where the apparent collapse in Cartesian coordinates around, e.g., index 20, doesn't translate into an analogous collapse in the (cos) hyperspherical coordinates: we believe that the reason for this is that, in the standard VAE, we are not exactly on the hypersphere (in the fourth histogram to the right in the first row in Fig.8, we can see that the norm of $\mu$, in orange, has a non-zero variance, since the prior is still a multivariate Gaussian, not exactly a uniform distribution on the hypersphere), while our compression VAE is indeed exactly on the hypersphere (analogous norm histogram in Fig.9), since we explicitly encourage the variance of the radial coordinate of $\mu$ to be 0. Thus, we emphasize that the improvements in generation by our compression method cannot be explained by selective posterior collapse (as in Fig.10), where the HD collapsed latent representation is effectively equivalent to a non-collapsed one in lower dimensions, since this comes at the cost of loosing reconstruction quality; but our method is able to improve generation while retaining some amount of better reconstruction, and this is why some of the best performative examples in Fig.2 cannot be re-obtained by a standard VAE with a different combination of parameters $n$ and $\beta$ (possibly in a selective collapsed mode). The improvement in our method is coming from the reduction of the sparsity by compression of the latent hypervolume and by performing this in a key angular way due to the peculiar equatorial nature of the volume in HD spaces.

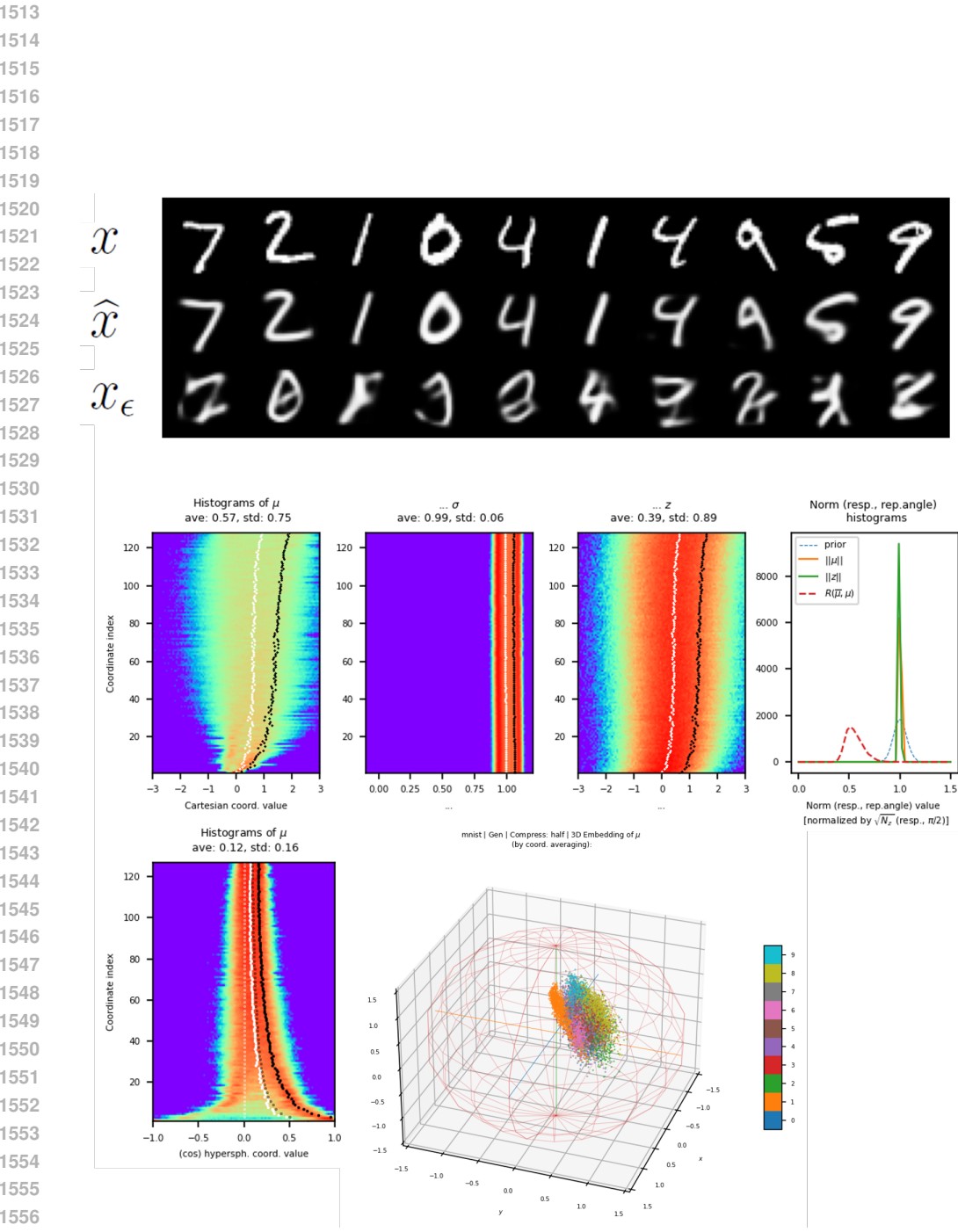

Figure 14: Results of a moderately compressed VAE training.

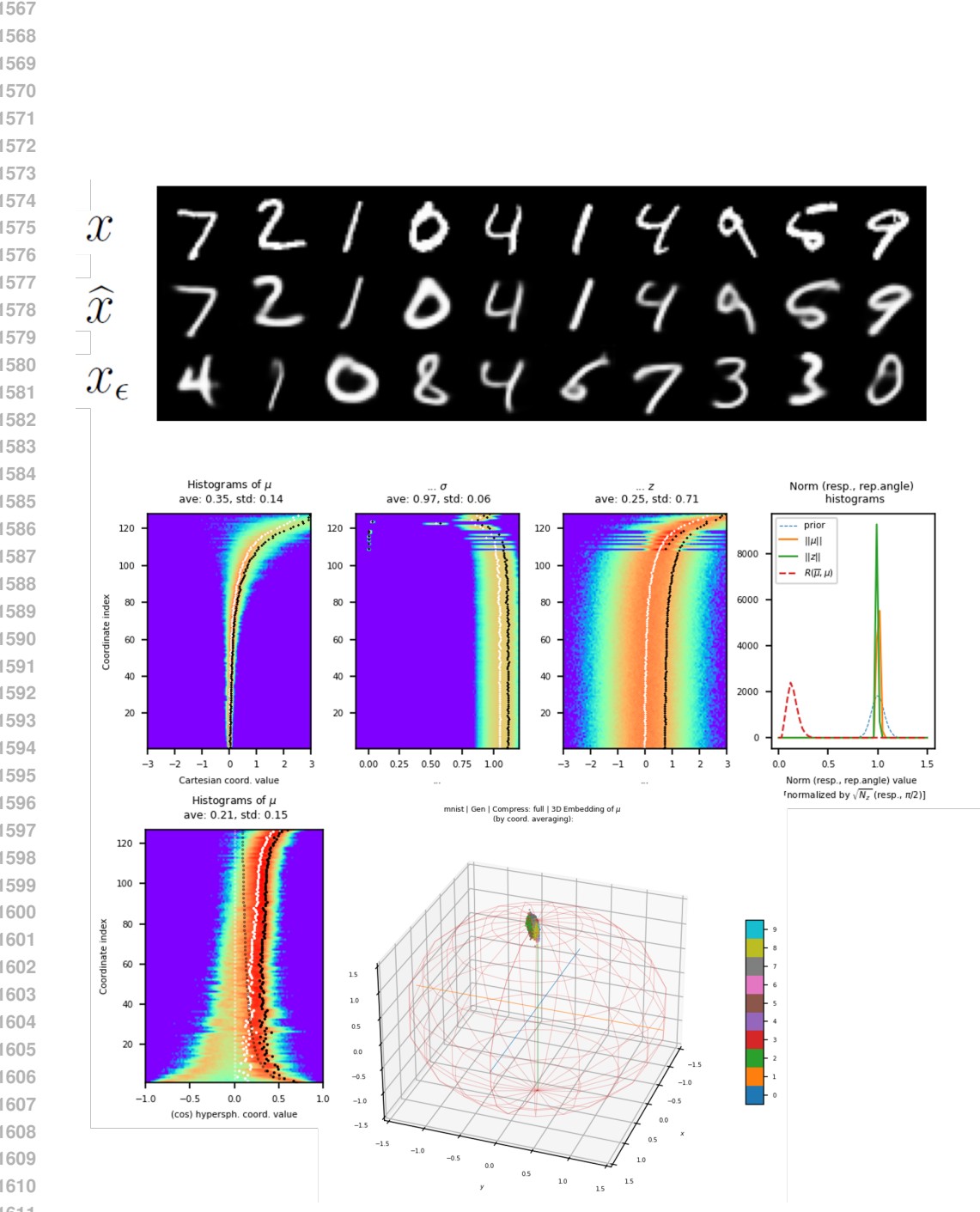

Figure 15: Results of a fully compressed VAE training.

## A.9 THE SPIN GLASS ANALOGY DURING TRAINING

We described in Section 4.1 of the main text the following training schedule and the reasons behind this choice: "*[...] we use an annealing schedule for the gain $\beta$ of the KLD-like loss, consisting of an initial stage which increases proportionally with $\sqrt{epoch}$ for the first 100 epochs, and is constant afterwards. This was necessary because we observed that too much compression of the volume was detrimental to the performance, while a strong compression was still necessary at the initial stage[...]*". The gain $\beta$ here has the role[2] of the inverse temperature, $\beta = 1/T$. In spin glasses and complex systems, the energy function has exponentially many local minima in the equatorial region of the hypersphere. To overcome them, a very strong signal or bias towards the desired region is necessary at the beginning, together with a rapid cooling or quenching. Thus, our initial high $\beta$ (i.e., very low temperature $T$) setting, and in the presence of the high intensity (regulated by the $\beta^{-1}$ factor in front of the MSE) hyperspherical external magnetic fields as bias in directions away from the equator, should make the gradient descent dynamics to quickly tend towards a low temperature distribution with replica symmetry breaking. Indeed, this is what we observed in our experiments, since we check for the replica angle, as mentioned before. This initial strong compression helps escaping those undesirable equatorial minima (Fig.16). Nevertheless, the obtained state shows too much overlapping between samples, so we then perform the annealing (i.e., lower the $\beta$, or increase the temperature $T$, and also lower the intensity of the magnetic fields) in order to allow the system to relax the strong order introduced by the initial bias and, in this way, transition to a replica symmetry breaking state with a bigger angle between replicas (that is, to go back up a bit in the ultrametricity tree/hierarchy of the replica angle values; cf. Mourrat (2024)). This decreases the MSE and makes the decoded images more sharp, at the cost of some generation quality (Fig.17). Note how the replica angle (red dashed lines in fourth histogram to the left in second row) doesn't fully go back to $\pi/2$, even when the KLD term (where the external magnetic fields are) stops optimizing at this stage of the training process (red line in third row), but instead jumps to a different value, higher than the initial one but still below $\pi/2$. This is fully consistent with the spin glass analogy in a quenched and then annealed system, where the glass, always in the replica symmetry breaking phase, jumps from one so-called 'pure state' to a different pure state, i.e., goes back up a bit in the ultrametricity tree/hierarchy of the replica angle values, as mentioned before. But the system has escaped the zone with exponentially many local minima in the equator.

---

[2]$\mathcal{L} = \beta \left( \beta^{-1}\mathrm{MSE}(x, x_z) + \mathrm{KLD}_{\mathrm{HSphCoords}}^{w/Prior}(\varphi_k, r) \right) = \beta\mathcal{H}$. cf. footnote 1, where $\nabla\mathcal{L} = \beta\nabla\mathcal{H}$ for the gradient descent dynamics on $\mathcal{L}$.

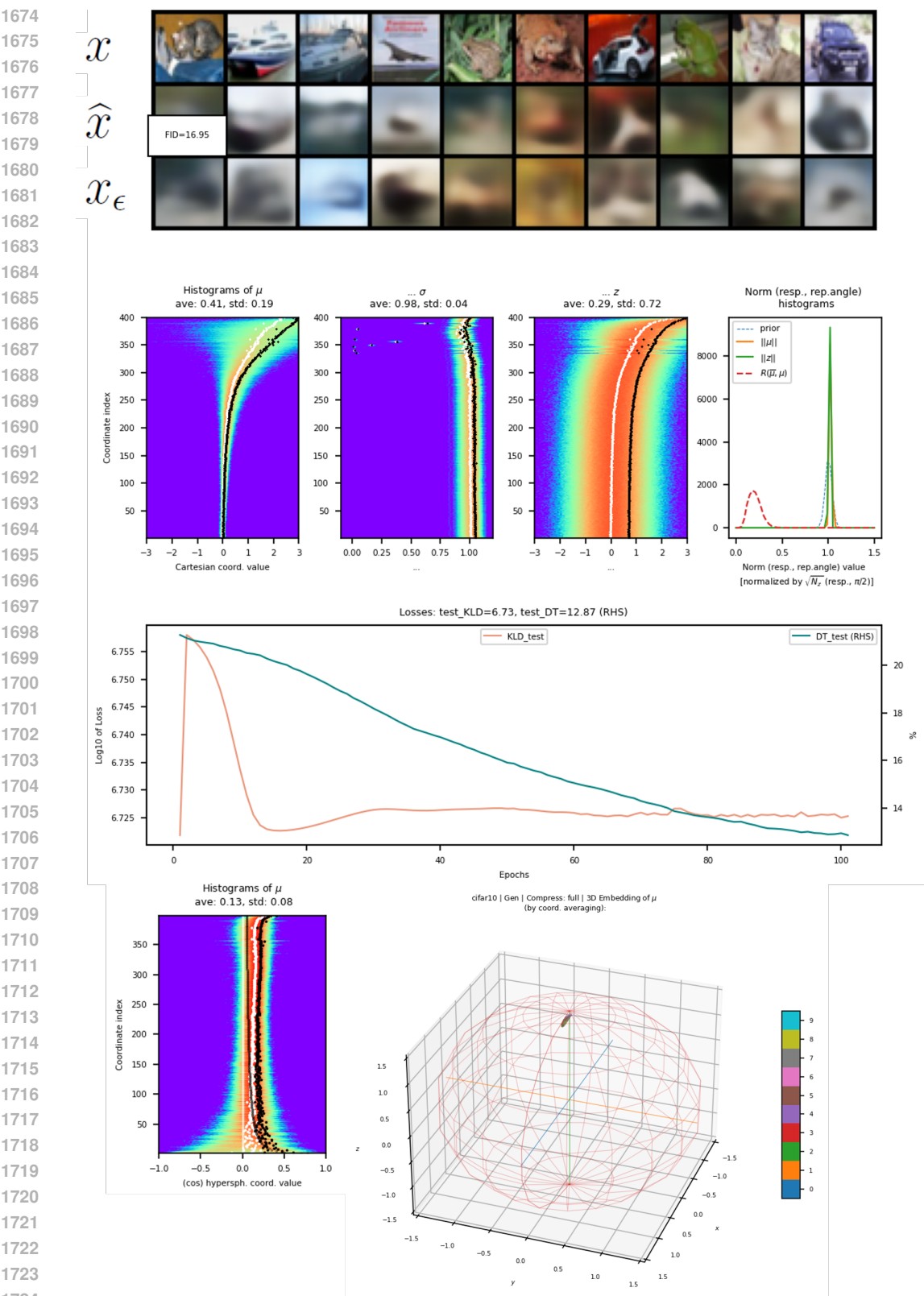

Figure 16: Results of a typical fully compressed VAE training at epoch 100.

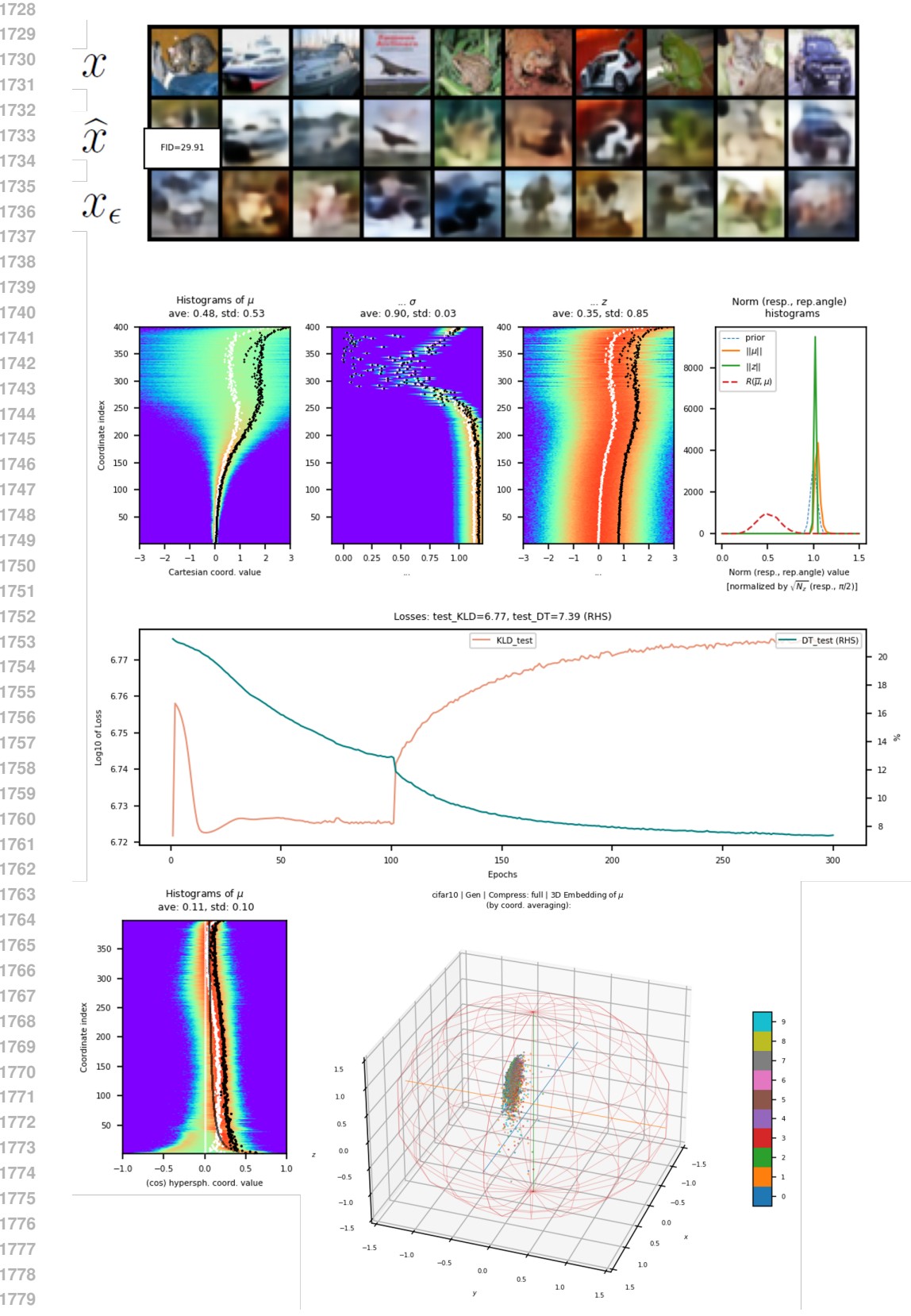

Figure 17: Results of the same fully compressed VAE training at final epoch 300.

## A.10 RESULTS ON CELEBA64

In this appendix we include additional experimental results conducted on the dataset CelebA (Liu et al., 2015), resized to a $64 \times 64$ image size.

The analysis is of the same type as the one we performed on CIFAR10 (cf. Figs.2, 7), and the results show qualitatively the same trends (Figs.19, 20).

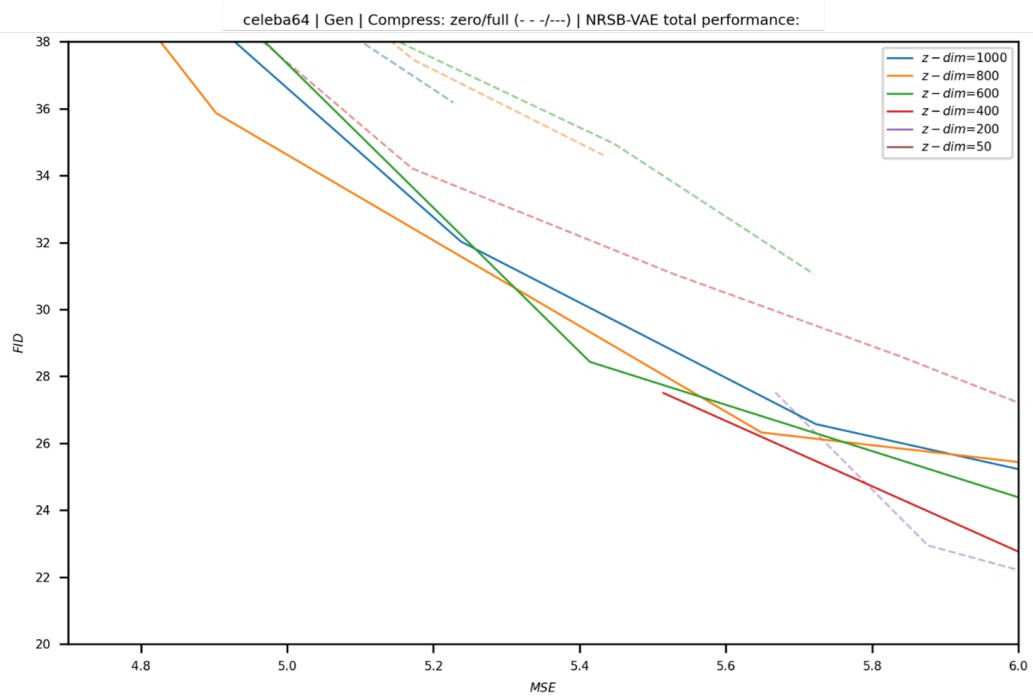

Figure 18: Effect of latent dimension and $\beta$ on the trade off between reconstruction and generation on CelebA64 (as in CIFAR10, solid lines closer to the bottom left corner than the dashed lines).

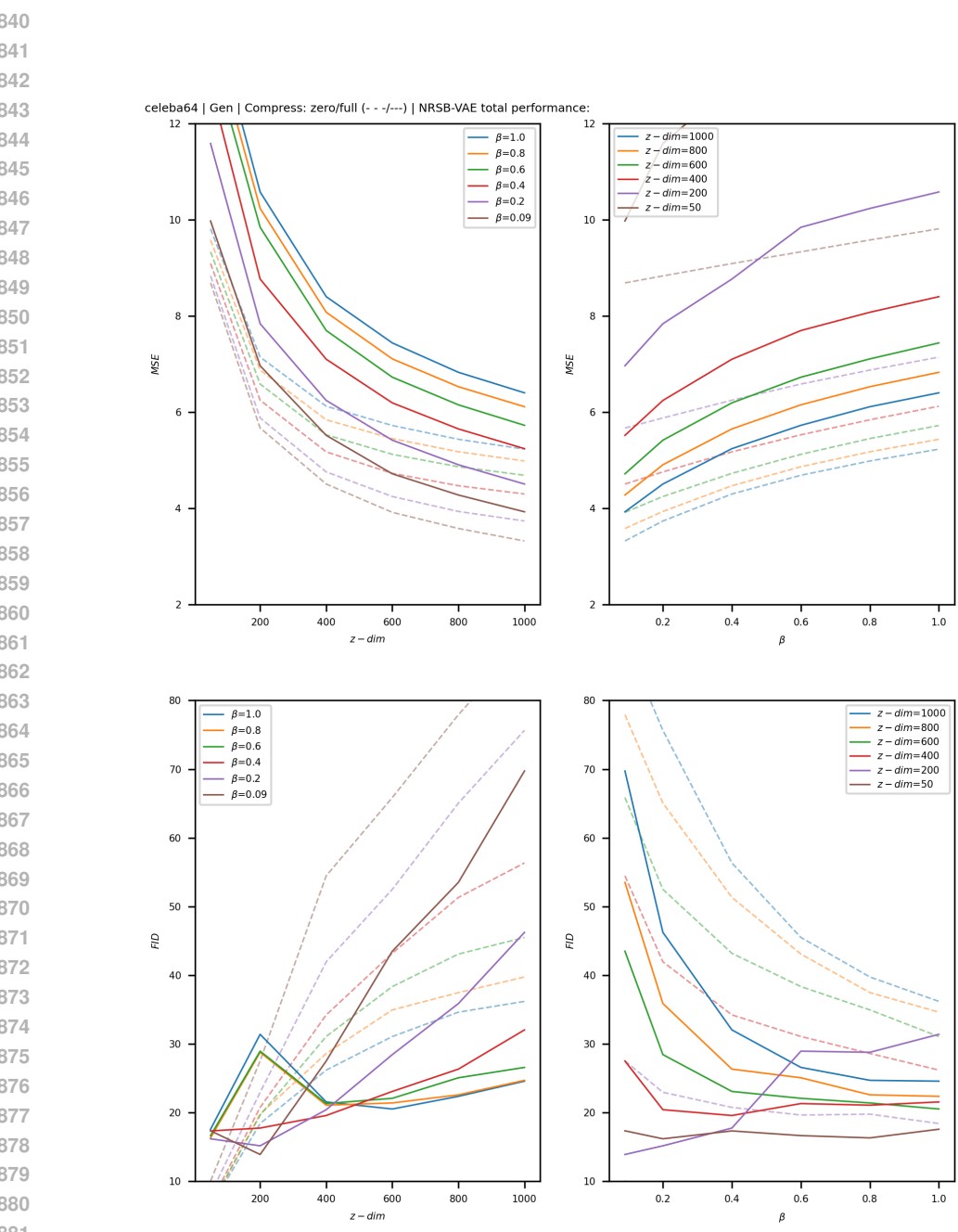

Figure 19: CelebA64 results breakdown. MSE and self-FID in terms of both the number of latent space dimensions and the total gain $\beta$.

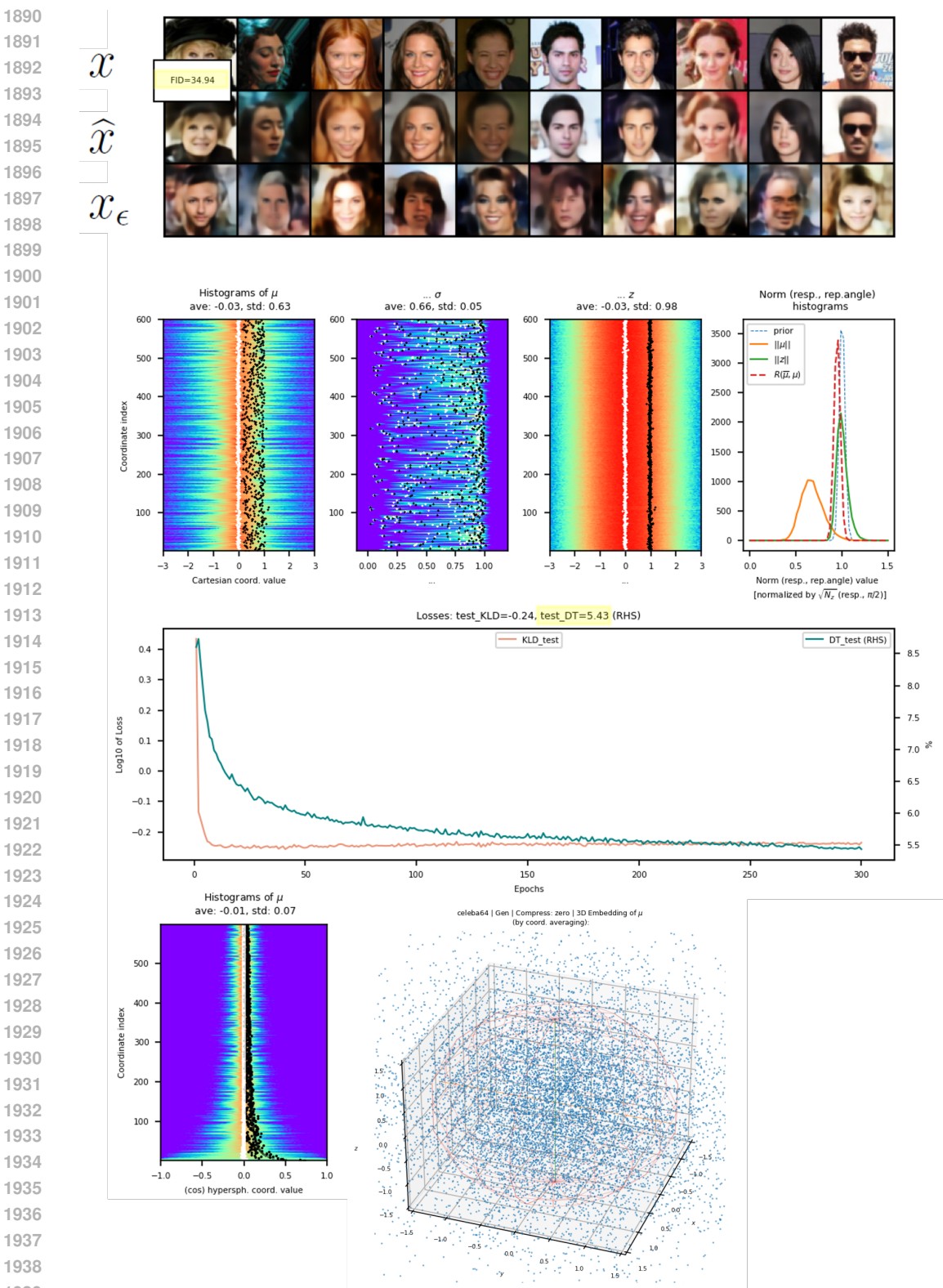

Figure 20: Results of standard VAE training with a balanced $\beta$ (in the 3-D embedding diagram, the samples are normalized by the overall mean of the radial coordinate, rather than set exactly to the sphere; thus, rather than looking like a uniform-like distribution on the 2-D sphere, it looks like a normal distribution in 3-D, but this difference is only merely in the convention being used regarding the radial normalization for the 3-D embedding). $MSE = 5.43$ and $self - FID = 34.94$, $n = 600$.

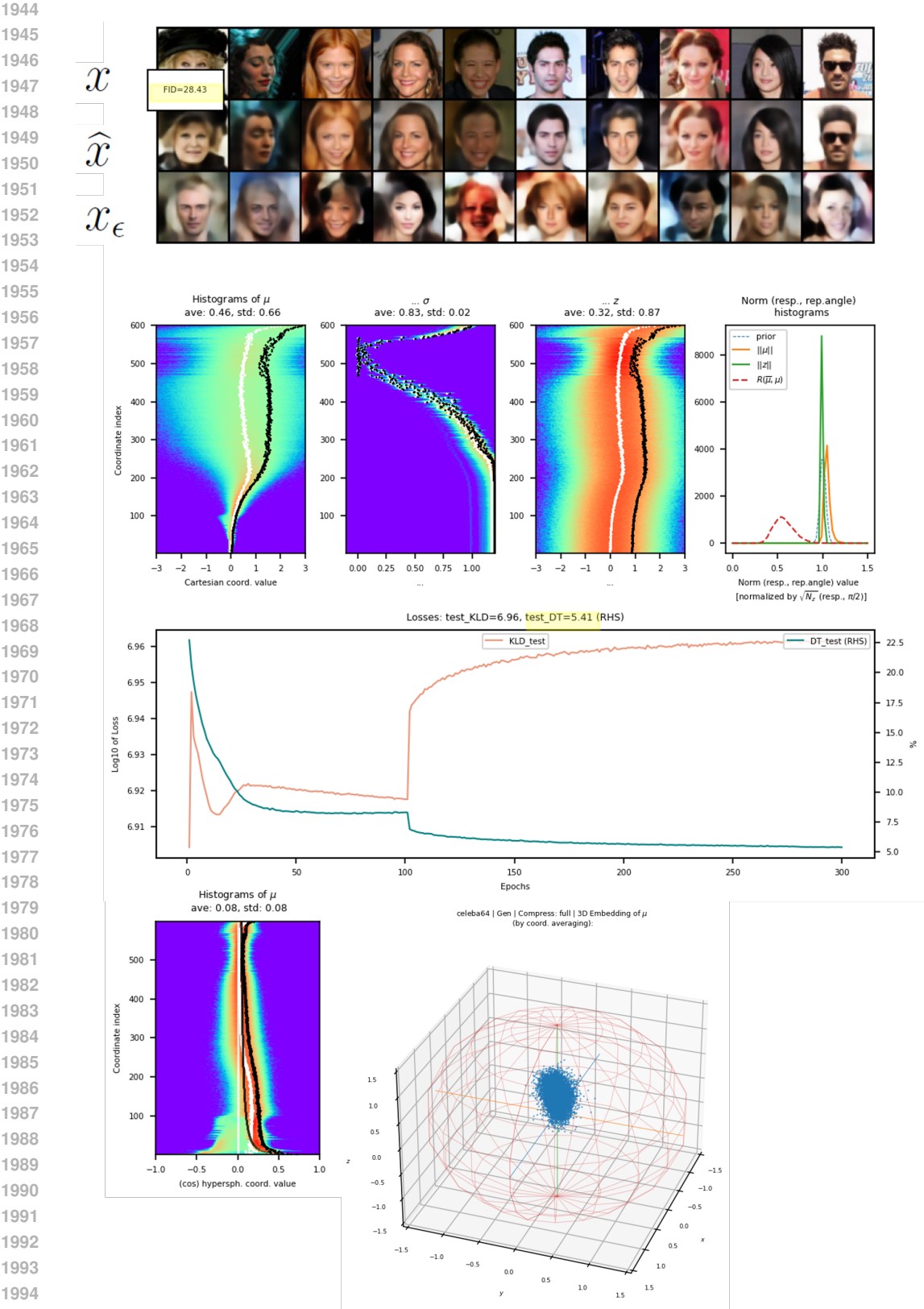

Figure 21: Results of a compressed VAE training. $MSE = 5.41$ and $self - FID = 28.43$, $n = 600$.

## A.11 INTERPOLATIONS ON MNIST

Here we complement the results and associated claims of Fig.1 with interpolations experiments on the same models. They highlight the lack of continuity in the standard VAE case (Fig.22), while they show the gained continuity and how densely packed the clusters are in our compressed version (Fig.23).

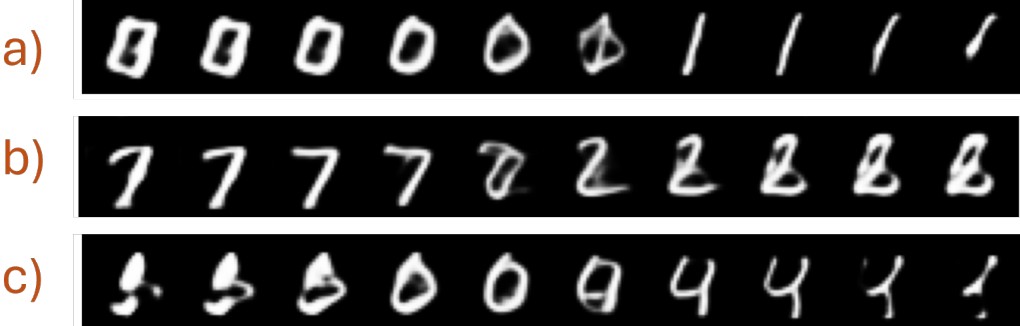

Figure 22: Interpolations on the standard VAE. a) from 0 to 1; b) from 7 to 2; c) from 0 to 4.

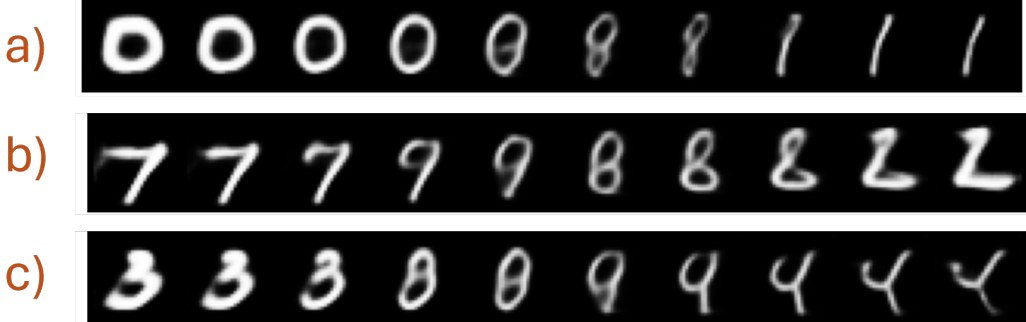

Figure 23: Interpolations on the compression VAE. a)-b) Idem as previous figure.

## A.12 THE DIFFERENT REGIMES OF THE STANDARD $\beta$VAE IN LOW DIMENSIONS

In this appendix, we perform a similar analysis as the one in A.6, but now for the model with the lowest latent dimension ($n = 50$).

In this situation, the trends actually reverse: de-collapsing the model (that is, going from Figs.24 to 25 and so on) improves the generation as measured by the self-FID (cf. Figs.10 to 11 and so on, in the HD case, where it becomes worse). See also Fig.7, second row, right.

Nevertheless, these cases are pathological and not very useful, since all of them have very high MSE, that is, the images are too 'blurry'. Thus, both the collapsed and the non-collapsed cases fall into the bottom far right of Fig.2, way outside the more useful area of the MSE-self-FID plane.

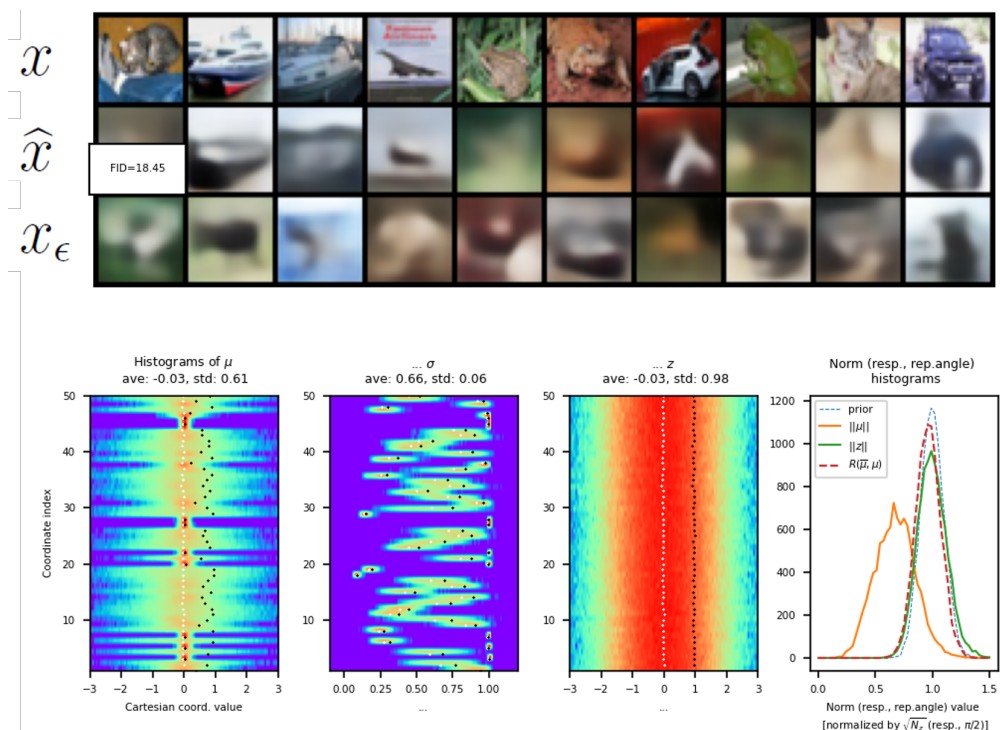

Figure 24: Results of a high $\beta$ ($= 1.00$) VAE training ($MSE = 12.27$, very poor). Notice the collapsed dimensions in the histograms for $\mu$ (the variance, black dots, for each of those dimensions is very close to 0). Good generation.

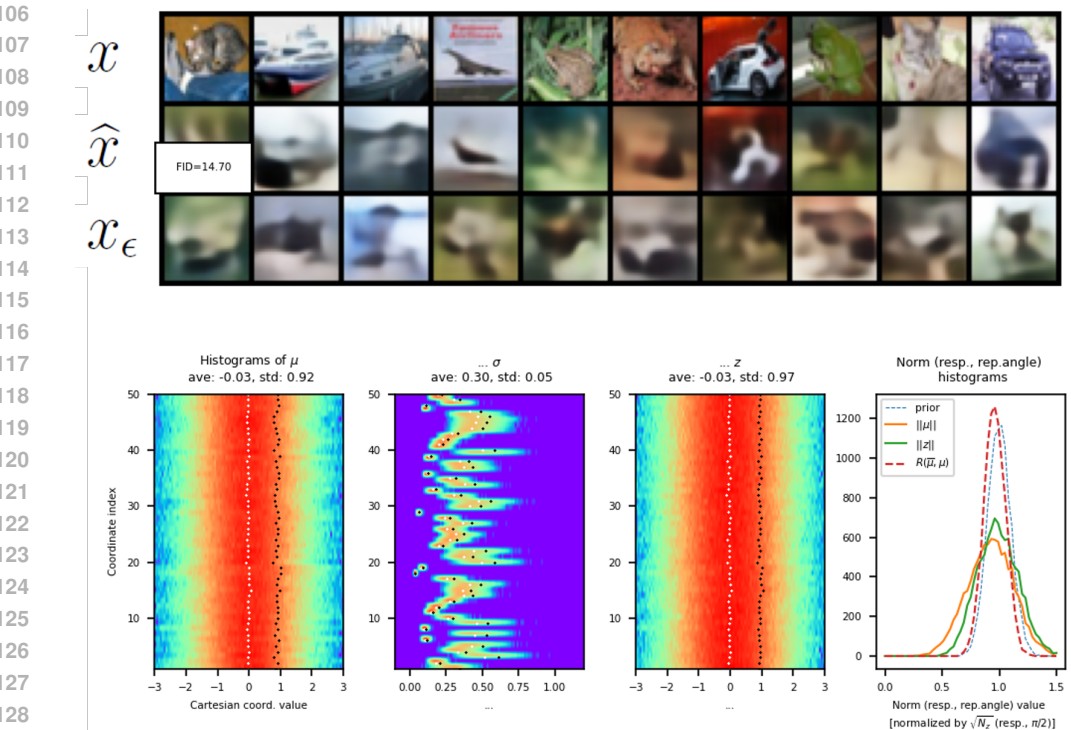

Figure 25: Results of a medium/balanced $\beta$ ($= 0.20$) VAE training ($MSE = 9.97$, poor). There are more functional dimensions than collapsed or almost collapsed ones. Better generation than previous figure.

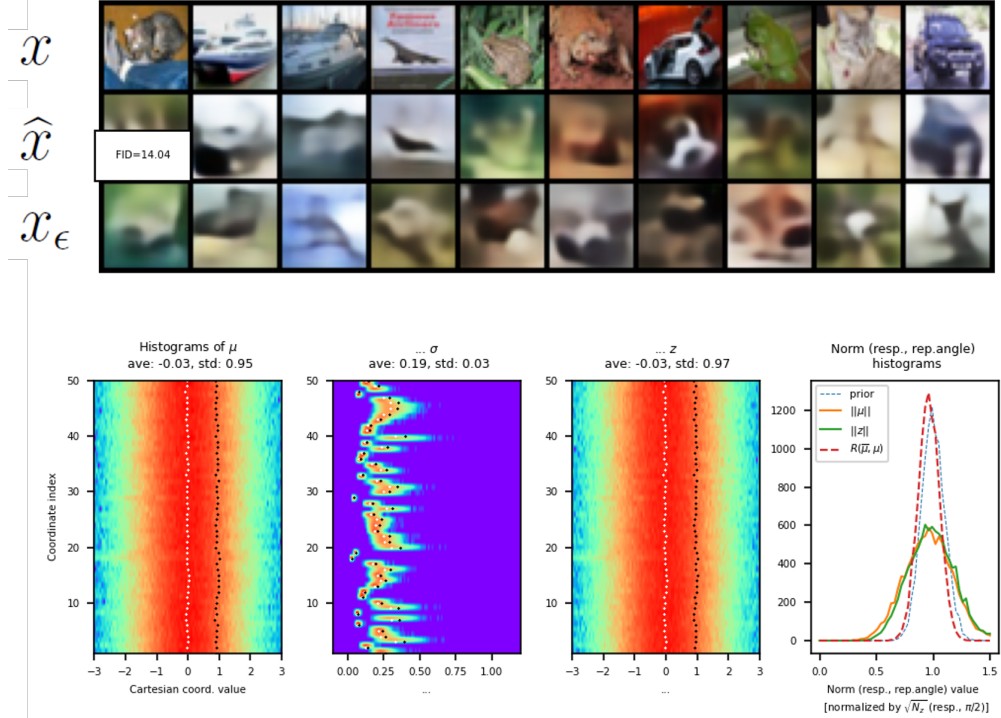

Figure 26: Results of a low $\beta$ ($= 0.09$) VAE training ($MSE = 9.59$, poor). There are no collapsed dimensions, but the model becomes almost an autoencoder (i.e., the VAE's $\sigma$ is close to 0). Even better generation than previous figure.

## A.13 RE-WRITING OF THE KLD TERM

In this appendix, we make explicit the steps to go from the standard form of the KLD term in the VAE to the one we used as a starting point for our own KLD in hyperspherical coordinates.

In Cartesian coordinates, the KLD divergence between the estimated posterior defined by $\mu_k$ and $\sigma_k$ and the prior defined by $\mu_k^p$ and $\sigma_k^p$ is (Odaibo, 2019):

$$\text{KLD}_{\text{CartCoords}}^{w/Prior} = \frac{1}{2} \sum_{k=1}^{n} \left[ \left( \frac{\sigma_k}{\sigma_k^p} \right)^2 - \log \left( \frac{\sigma_k}{\sigma_k^p} \right)^2 - 1 + \frac{(\mu_k - \mu_k^p)^2}{(\sigma_k^p)^2} \right] \tag{13}$$

A Taylor approximation (up to second order) of the part for sigma around its prior yields for some constants $\gamma_k$ and $\widetilde{\gamma}_k$:

$$\text{KLD}_{\text{CartCoords}}^{w/Prior} \approx \sum_{k=1}^{n} \left[ \gamma_k \left( \sigma_k - \sigma_k^p \right)^2 + \widetilde{\gamma}_k \left( \mu_k - \mu_k^p \right)^2 \right] \tag{14}$$

In practice, the optimization is performed over mini batches of data (of size $N_b$), using the objective below:

$$\text{KLD}_{\text{CartCoords}}^{w/Prior} \approx \frac{1}{N_b} \sum_{l=1}^{N_b} \sum_{k=1}^{n} \left( \gamma_k \left( \sigma_{k,l} - \sigma_k^p \right)^2 + \widetilde{\gamma}_k \left( \mu_{k,l} - \mu_k^p \right)^2 \right) \tag{15}$$

If we denote the corresponding batch statistics as $\mathbb{E}_b$ and $\sigma_b$, then, by using the basic formula,

$$\mathbb{E}_b[X^2] = \mathbb{E}_b[X]^2 + \sigma_b[X]^2, \tag{16}$$

we can write this objective as (we omit the constants for ease of reading)

$$\text{KLD}_{\text{CartCoords}}^{w/Prior} \approx \sum_{k=1}^{n} \left( (\mathbb{E}_b[\sigma_k] - \sigma_k^p)^2 + \sigma_b[\sigma_k]^2 + (\mathbb{E}_b[\mu_k] - \mu_k^p)^2 + \sigma_b[\mu_k]^2 \right) \tag{17}$$

## A.14 THE DIFFERENT REGIMES OF THE STANDARD $\beta$VAE IN HD: CELEBA64

This is a complete analogue of A.6 but for the CelebA64 dataset with $n = 1000$.

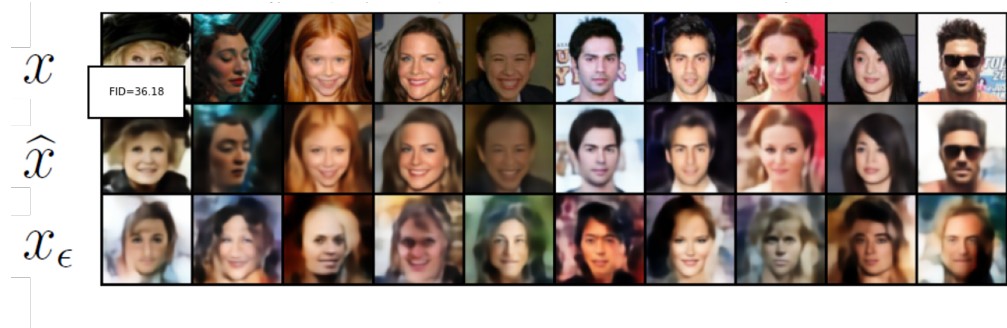

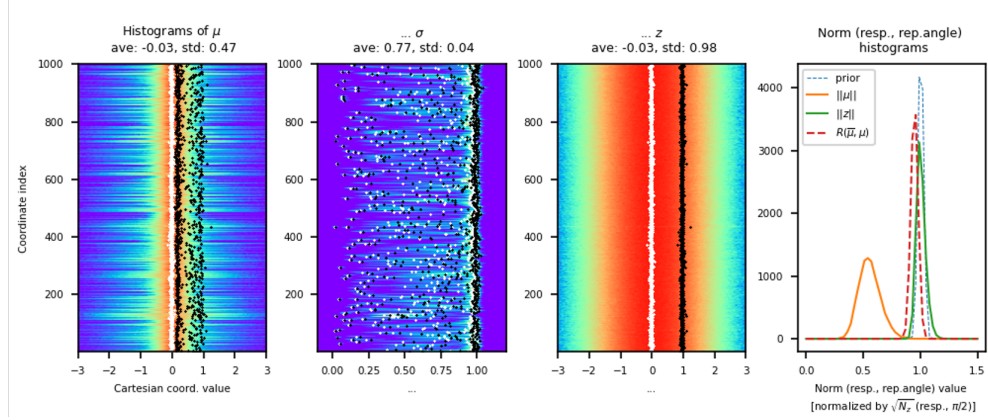

Figure 27: Results of a high $\beta$ ($= 1.00$) VAE training ($MSE = 5.22$, good). Regular to bad generation.

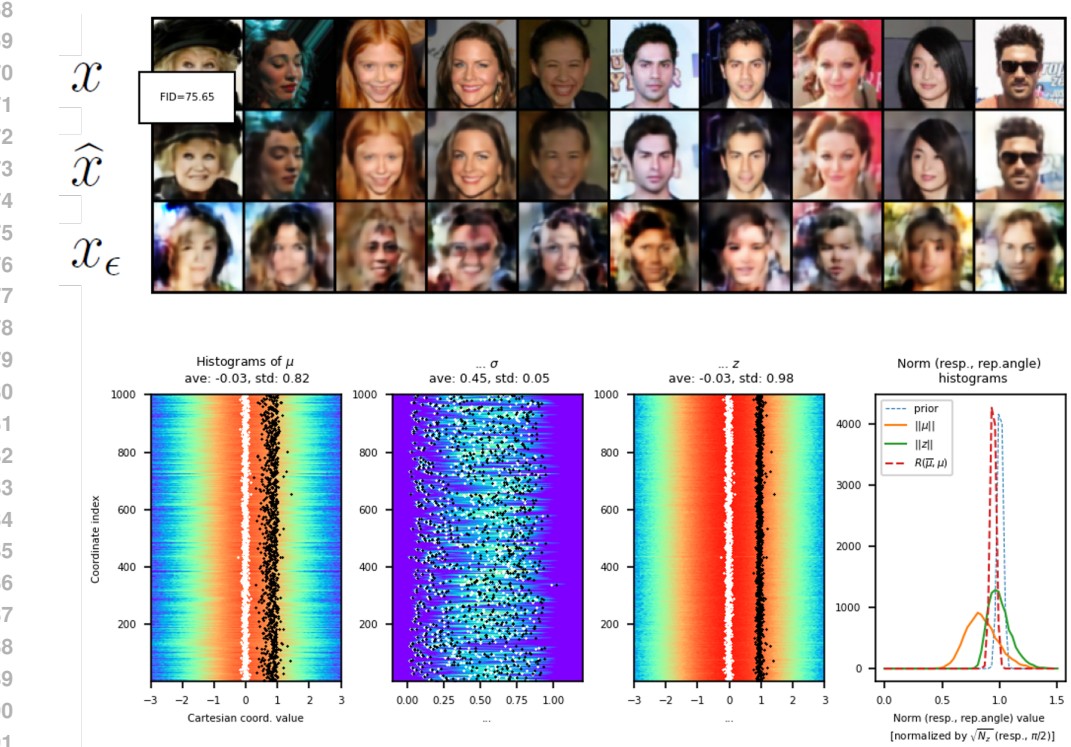

Figure 28: Results of a medium/balanced $\beta$ (= 0.20) VAE training ($MSE = 3.74$, good). Bad generation.

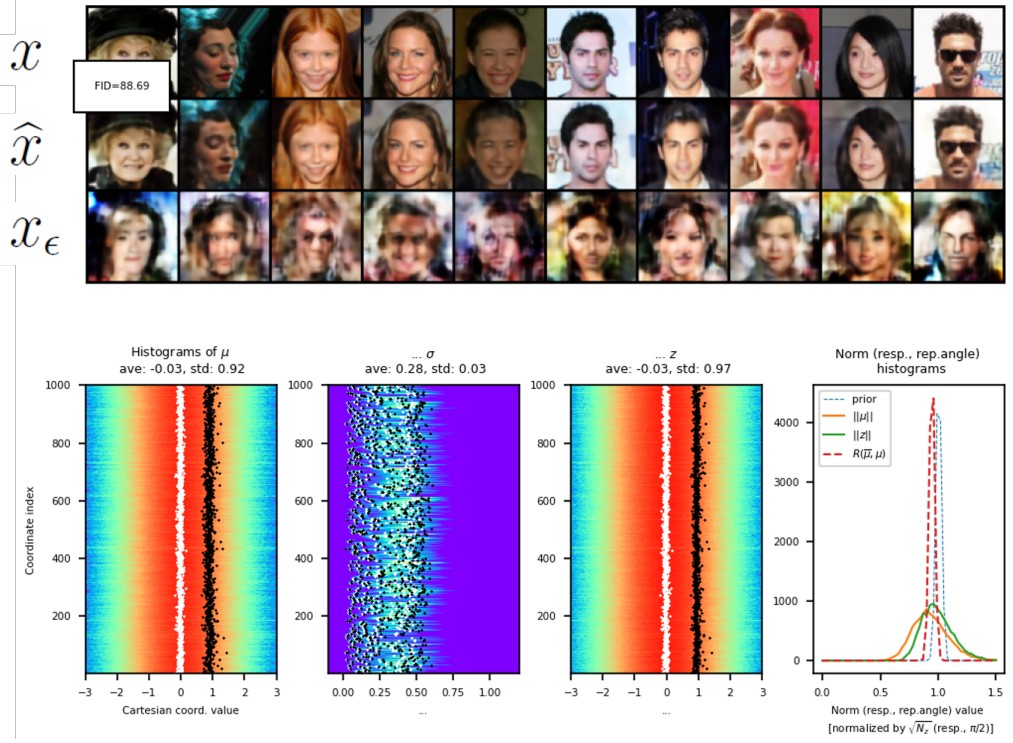

Figure 29: Results of a low $\beta$ (= 0.09) VAE training ($MSE = 3.31$, good to very good). Very poor generation.

### A.15 IT IS THE SPARSITY!

In this apppendix, we collect further experimental evidence, from different directions, that complements our findings in Fig.1, as well as the discussion in A.7, A.8, and A.11, that supports our hypothesis about the sparsity of HD latent spaces as the root cause of the poor generation for the standard VAE.

Continuing the analysis of A.7 and Fig.13, to further confirm that this poor generation is caused by the sparsity, we started with a very low value for the radial mean of the VAE's $\sigma$, from $a_{\sigma,r} = 0.1 \times \sqrt{n}$ (this is similar to an autoencoder) in Fig.30, to $a_{\sigma,r} = 1.0 \times \sqrt{n}$ (this is similar to a VAE regime) in Fig.13, and finally to $a_{\sigma,r} = 2.0 \times \sqrt{n}$ (this is similar to a VAE regime with very wide encoded latent distributions for each encoded data point $x$) in Fig.31. Actually, in this latter case, after the reparameterization trick and normalization of $z_x$ to the hypersphere, $z_x$ could be, for each $x$, in *any* location on the hypersphere, not just a portion of it.

The logic behind this is that the bigger $a_{\sigma,r}$ is, then more of the latent space is covered during training because of the reparameterization trick; thus, if sparsity is the cause of the poor generation, we should be able to see some improvement as we increase $a_{\sigma,r}$. And this is indeed what our experiments show. Nevertheless, the amount of improvement in generation by this method quickly reaches a limit for the extreme case $a_{\sigma,r} = 2.0 \times \sqrt{n}$, since there is probably too much overlapping between the encoded distributions for all the different datapoints. On the other hand, our method which consists in the angular coordinates compression of the distributions (rather than the previous radial expansion of $\sigma$) gives better, more consistent and efficient results (we believe this is because of the natural angular behavior of the hypervolume in HD; cf. A.3, A.4), cf. Fig.15.

In Fig.32, we show a standard VAE with decoded random samples from the aggregate approximate posterior distribution, rather than the prior. We can see that there is no much improvement in the generation, and thus this cannot be attributed to a mismatch between the prior and aggregate approximate posterior distribution in this case (as in some of the references mentioned in 2). In Fig.14 of A.8, we take a moderately compressed model (by our method) and show the $3D-$embedding of the approximate aggregate posterior, which is a von Mises-Fisher-like distribution, and it shows no holes or cracks (since our method, by compressing the data, makes the clusters more packed and closer to each other; thus, it could also be applied to solve or mitigate the prior hole problem, but we will leave it at that for now, since the main point of this paper is the sparsity issue in HD). If the philosophy of the mismatch between prior and posterior as root cause of bad generation were true, then sampling from such an approximate aggregate posterior von Mises-Fisher-like distribution should produce better generation. But we do exactly that in that figure and we don't see any improvement. It's only when we do a full compression mode that we get the improvement in generation.

In Fig.33, we computed the *persistence diagrams* corresponding to the experiments in Figs.13, 14, 15 ( *a)*, *b)*, *c)* in Fig.33, respectively). We used the GUDHI library (Maria et al.; https://gudhi.inria.fr/) for producing the persistence diagrams presented here.

This is a key tool from Topological Data Analysis (TDA) (Carlsson, 2009), which allows us to explicitly and directly assess the precise amount of sparsity in our HD latent spaces. Consider the encoded means $\mu$ of our dataset in latent space as a point cloud, then we take a cover of $n$-balls of a fixed radius $\varrho$ around each point in the cloud. At 'time' of radius $\varrho = 0$, all the balls are obviously non-intersecting with each other, this is called the 'birth time' $B_\mu$ of the points $\mu$ (thus, all points have birth time $B_\mu = 0$). We then start increasing the value of $\varrho$ with 'time'. If at some moment of radius $\varrho_\mu$ the ball around a point $\mu$ *intersects* other balls, this is called the 'death time' of that point, $D_\mu$, and has the value $D_\mu = \varrho_\mu$. In this way, each point $\mu$ is characterized here by its corresponding pair $(B_\mu, D_\mu) = (0, \varrho_\mu)$, and it is these pairs that we plot in the persistence diagrams, for all the points in the cloud.

Thus, it is not difficult to see why this directly assess the sparsity of the cloud: the longer the death times for a given cloud, this means that the points are more separated from each other and that the point cloud is more sparse.

In this way, we can see a very interesting correlation between the sparsity measured in this manner and the quality of the generation. When we go from Fig.13 to 14, we see that there is indeed some compression (this can be seen in the replica angle and the $3D-$embedding in those figures), but there is no appreciable improvement in the generation. In the corresponding persistence diagrams

of Fig.33 ( $a$) and b), respectively), we can see that there is a small diminution of the death times (of around $\Delta\varrho_\mu \approx 1000$ for each point) between diagrams. Nevertheless, the difference in the death times between the shortest living point and the longest one in the same diagram remains almost unchanged, around $\Delta\varrho \approx 9000$ for each diagram. Thus, this moderate compression only achieves a reduction in death times by a constant translation factor of $\Delta\varrho_\mu \approx 1000$ on all death times.

It is only when we pass to the full compression mode, Fig.15, that we see an appreciable improvement in the generation. In the corresponding persistence diagram of Fig.33 ( $c$) ), we can see that there is a considerable greater reduction in the life time of the longest living point, from around $D_{\mu_{max}} \approx 10000$ in the previous two diagrams, to around $D_{\mu_{max}} \approx 2000$. Furthermore, we also get a considerable decrease of the difference in the death times between the shortest living point and the longest one in the same diagram, from around $\Delta\varrho \approx 9000$ in the previous diagrams, to around $\Delta\varrho \approx 2000$ in the diagram in c). Thus, our analysis here shows that this dramatic reduction in sparsity is strongly correlated to the appreciable improvement in the quality of the generation. It was the sparsity!

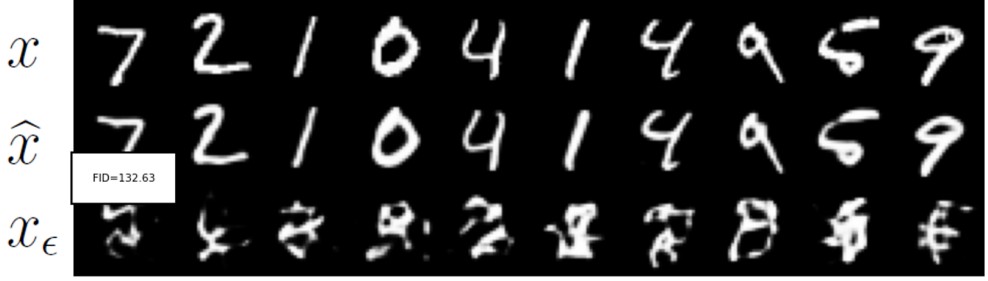

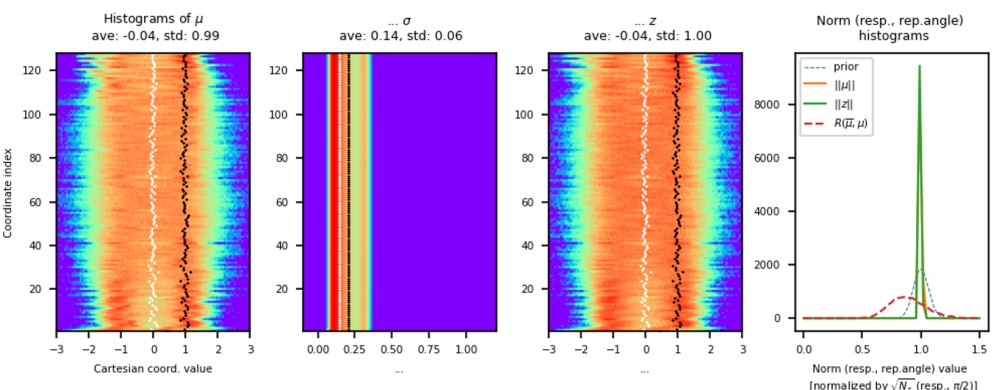

Figure 30: Hyperspherical VAE with $a_{\sigma,r} = 0.1 \times \sqrt{n}$.

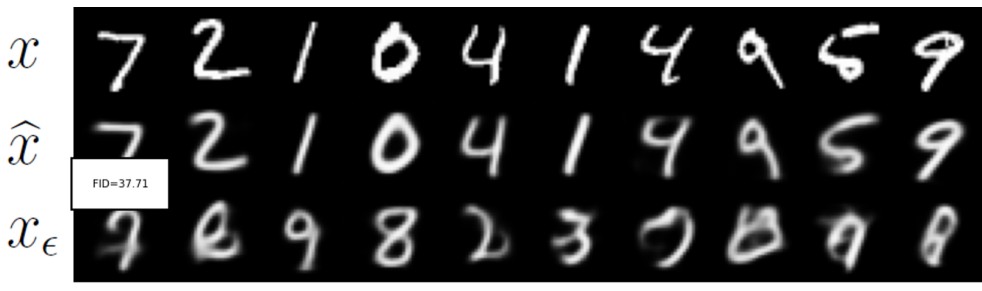

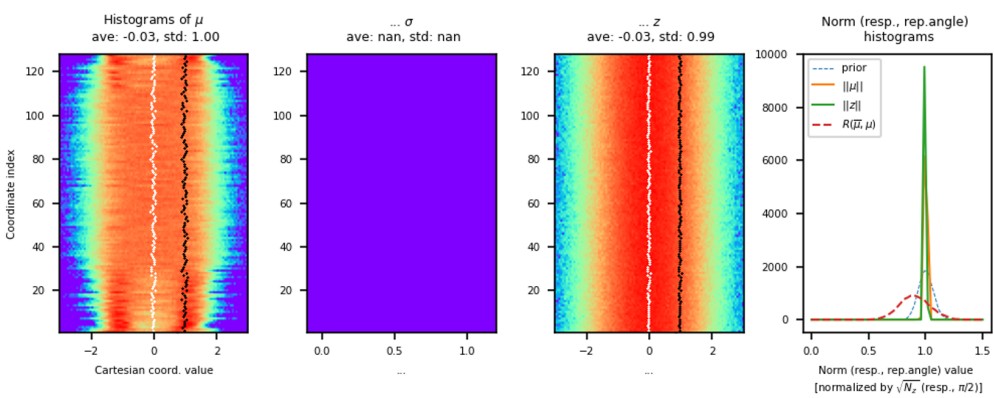

Figure 31: Hyperspherical VAE with $a_{\sigma,r} = 2.0 \times \sqrt{n}$ (histograms for $\sigma$ out of scale here).

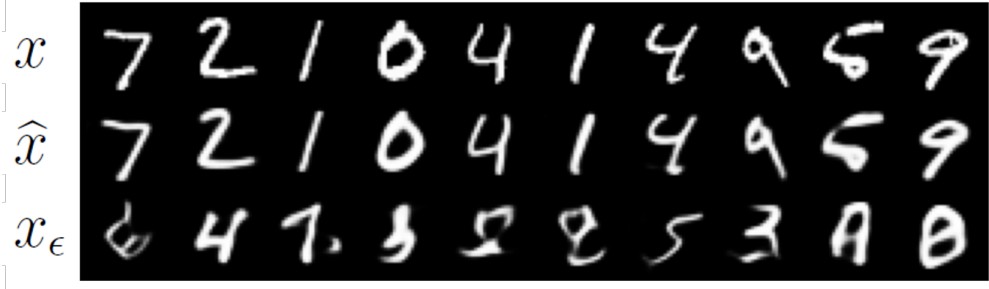

Figure 32: Standard VAE with decoded random samples from the aggregate approximate posterior distribution.

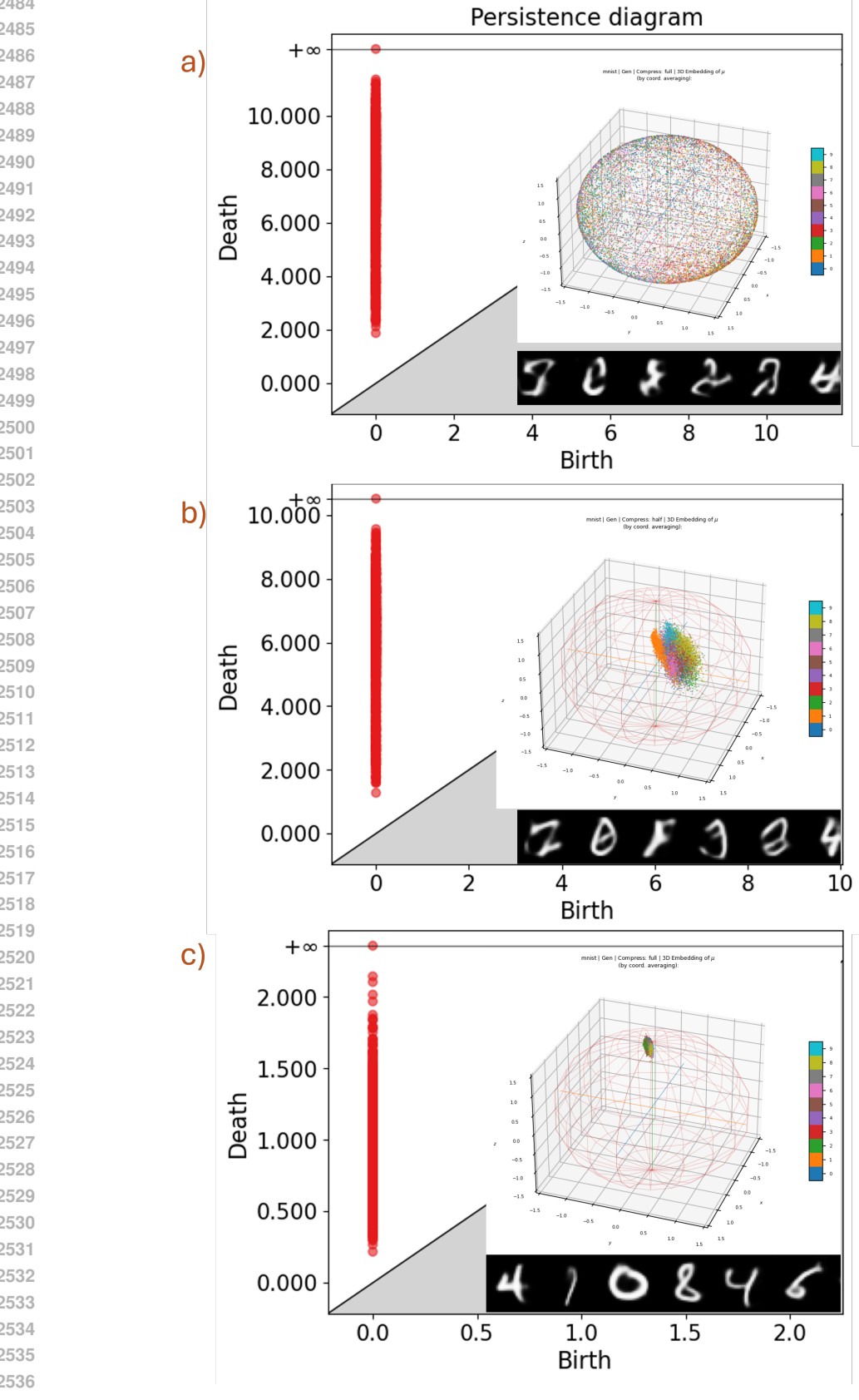

Figure 33: Persistence diagrams.

### A.16 DIFFERENCES IN TRAINING SPEED

We provide here data regarding the differences in the training speeds between the standard VAE and our compression VAE via hyperspherical coordinates. The origin of this difference mainly lies in the extra calculations needed for the coordinate transformations in A.1, which are implemented via the script in A.2.

The measurements were done during typical trainings in a NVIDIA H100 GPU. In Fig.34 we show the results for the case of trainings with CIFAR10, with a batch size of 200 samples, and the changes in training speed (measured as how many batches per second are being processed) in terms of the dimension $n$ of the latent space. After $n = 200$, until $n = 800$, the decay is almost linear in $n$, with a decay rate in the speed of 20 batch/s every 200 latent dimensions.

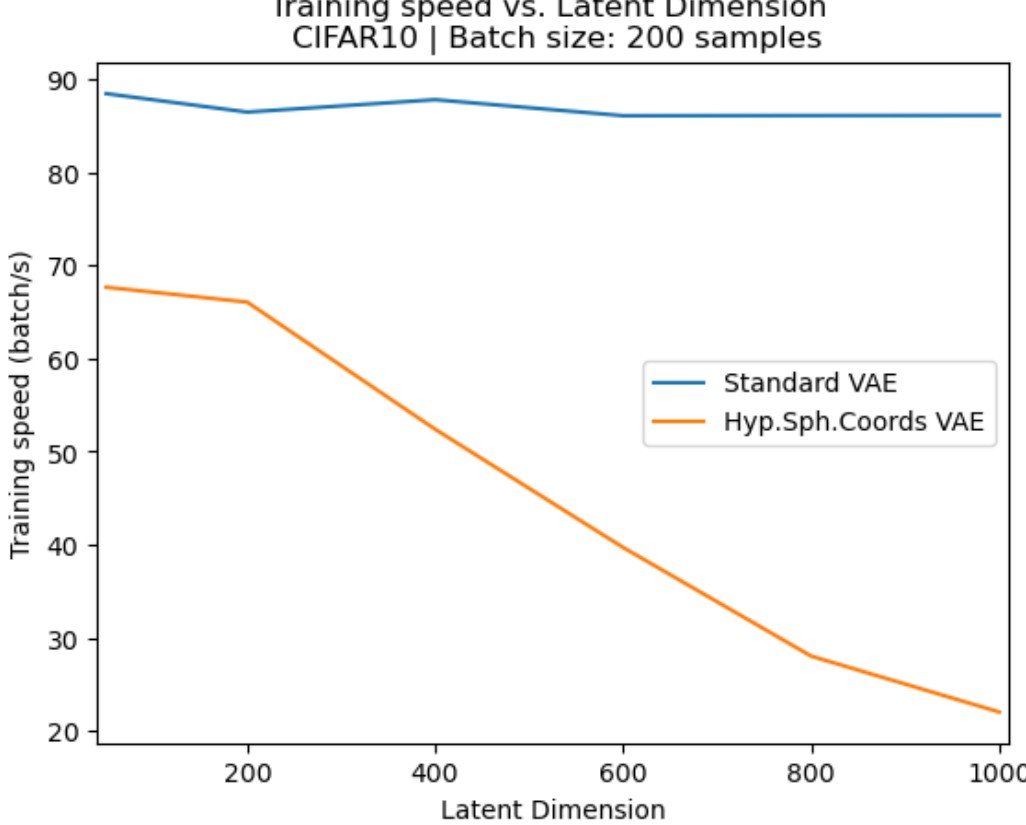

Figure 34: Differences in training speed.