# OpenReview forum: "Is the sparsity of high dimensional spaces the reason why VAEs are poor generative models?"
_ICLR.cc/2025/Conference — Submitted to ICLR 2025_

### Official Review · Reviewer_8unt · 2024-10-30

**Soundness:** 2
**Presentation:** 3
**Contribution:** 3
**Rating:** 6
**Confidence:** 4

**Summary:**

This paper explores the poor image generation quality of VAEs at inference time when the latent representation is high dimensional. After hypothesising, based on spin glasses theory from statistical physics, that the issue comes from the sparsity of high-dimensional spaces. Based on this, they propose to apply a mechanism akin to the quenching process used to reduce the entropy of such systems. Practically, this is done by implementing a change of coordinates from Euclidean to hyperspherical during the KL divergence computation and setting the priors of each dimension of these new coordinates such that the latent samples are pushed away from highly entropic regions. This results in an improved generation quality for MNIST while keeping latents that are interpretable enough to perform clustering.

**Strengths:**

Originality
========
- I like the creative approach of solving the issue of poor image quality generation by applying a solution to a similar problem from physics.
This is quite different from previous publications about hyperspherical VAEs, where the main motivation (as far as I know) was to provide a better prior than the standard multivariate Gaussian.
- The proposed model also differs from others in the hyperspherical VAE literature. Usually, changing the prior and posterior distribution can be complex as the reparametrisation trick and the KL divergence need to be updated accordingly. Here, the proposed solution is an elegant change of coordinates done after the reparametrisation trick during the KL divergence computation, making it easy to implement and intuitive to understand.

Quality/Clarity
===========
- The section on spin glasses is well vulgarised, intuitive, and reads very well for non-physicians.
- The paper is generally well-written and easy to follow.

Significance
=========
- Given the simplicity of the implementation, practitioners looking for better image generation quality with VAEs could easily adopt the proposed model.
- The explanation of why the generation is bad at inference time is interesting for the research community working on the learning dynamics of VAEs.

**Weaknesses:**

While I really like this paper's approach, I have some concerns about its empirical soundness and found several mistakes in the given equations (see major comments below). Clarifying some aspects of the paper would also strengthen it (see major and minor comments below). If these concerns were addressed, I would happily raise my score.

Major comments
=============

Experimental soundness
---------------------------------
- The experiment compares the results of $\\beta$-VAEs with change of coordinates + annealing with the results obtained with $\beta$-VAEs without annealing. As a result it is not possible to see if the improved generation comes from the annealing or the change of coordinates. Additional results with $\\beta$-VAEs using the same annealing schedule would be needed to ensure that the improvement is really due to the change of coordinates.
- While the experiment is partially done on CIFAR 10, it is hard to assess how several results generalise. For example, a comparison of the classification accuracy, examples of images generated, and a comparison of the projection into spheres using both models would be nice to have on CIFAR 10 as well.

Mathematical soundness
----------------------------------
In section 1.1, several equations were incorrect.
- In Eq. (1) $KLD(q_{\\phi}(z) || p_{\\theta}(z))$ should be  $KLD(q_{\\phi}(z|x) || p_{\\theta}(z))$
- In Eq. (2) the given formula is not for $KLD(z,\\epsilon)$ but for $-KLD(z,\ \epsilon)$. Furthermore, subscripts from the summations are missing. I would suggest either removing the second summation and using a matrix form like $KLD(z,\\epsilon) = \\frac{1}{2} \sum^{N_b} \\bigl(Tr(\\sigma) + || \\mu||^2_2 - \\log det(\\sigma) - n\\bigr)$, or rewriting it with subscripts as  $KLD(z,\\epsilon) = \\frac{1}{2} \sum^{N_b} \sum_{k=1}^n \\bigl(\\sigma_i^2 + \\mu_i^2 - \\log(\\sigma_i^2) -1 \\bigr)$.
- In Eq. (3), this is not the ELBO $\\mathcal{L}$ from Eq. (1) but its negative approximation $- \\tilde{\\mathcal{L}}$ which is minimised by the VAE. While the original formulation of VAEs by [1] does not contain a $\\beta$ term and is equivalent to setting $\\beta=1$, it would be interesting to briefly discuss what is the impact $\\beta$ when $\\beta > 1$ and when $\\beta < 1$. Indeed, the motivation for both settings is very different: the first is to provide "disentangled representations" [2] and to force the VAE to learn in a polarised regime (a.k.a selective posterior collapse), which is akin to a PCA-like behaviour [3,4,5], while the second aims at mitigating posterior collapse and is often used together with annealing [6]. Thus, the choice of $\\beta$ has a practical impact on the proposed experiment (see further discussion on the questions part below).

Clarity
---------
- To facilitate the understanding of the paper, it would be great to have the derivations from Eq. (4) to Eq. (5) and from Eq (8) to Eq (9) in appendix.


Minor comments
=============

Mathematical notation
------------------------------
The current notation is sometimes confusing. For example, l. 77 $\\mathcal{N}(z; \\mu, \\sigma)$ reads as "the univariate Gaussian with mean $\\mu$ and standard deviation $\\sigma$" while it is in fact a multivariate Gaussian. A suggestion to improve this is to use a different notation for numbers, vectors, and matrices, following, for example the notation suggested in the math_commands.tex file of the ICLR template.

Clarity
---------
- l. 478, I found the sentence "the quality of AE and VAE [...]" confusing as AEs are not discussed anywhere else in the paper and are not used in the experiment. I would suggest removing the part about AE to make the argument clearer.
- I struggle to see what 32% of computation time of the KLD represent in term of additional training time. It would be easier to see with an average run time with and without change of coordinates over n seeds and k epochs. Furthermore, if this increases with the number of dimensions, an estimate of the increase rate using big O notation would be very useful for practitioners to assess whether this implementation is suitable to their needs.
- The seminal papers on VAEs are [1,7] which are different from the ones references. I would suggest updating this.

Typos
--------
- l. 91 weighs -> weights
- l. 100 teh -> "that the" ?
- l. 133 there's -> there is
- l. 133 have -> has
- l. 157 it's -> it is
- l. 422 gven -> given
- KL divergence is inconsistently refered to as "KL divergence" and "KLD" in the paper.
- l.591-592, the title of Higgins et al. is capitalised while others are not.

**Questions:**

- As mentioned in the weaknesses section, a VAE usually learns in a polarised regime when $\beta$ is sufficiently high. In this setting, the latent representations contain two kinds of variables: active and passive. The passive variables are kept as close to the prior as possible and used to lower the KL divergence, while the active variables contain the information needed for reconstruction (or further use in downstream tasks). Active variables typically do not follow the prior as the KL is kept low by the passive variables. Instead, they have a very low $\\sigma$ such that during the reparametrisation trick, $z \approx \\mu$. One would usually remove passive variables for downstream tasks, only keeping the small subset of variables containing some information [8]. I wonder if keeping only active variables would change the sphere projection to something more akin to what is obtained with hyperspherical coordinates?
- What is the impact of using different values of $a_{\mu, k}$ (keeping $a_{\mu, k} \neq 0$ of course)? Was there a specific reason to choose $a_{\mu, k}=1$?

References
=========
- [1] Kingma, D. P. and Welling, M. (2014). Auto-Encoding Variational Bayes. In International Conference on Learning Representations, vol. 2.
- [2] Higgins, I., Matthey, L., Pal, A., Burgess, C., Glorot, X., Botvinick, M., Shakir, M. and Lerchner, A. (2017). $\\beta$-VAE: Learning Basic Visual Concepts with a Constrained Variational Framework. In International Conference on Learning Representations, vol. 5.
- [3] Rolinek, M., Zietlow, D. and Martius, G. (2019). Variational Autoencoders Pursue PCA Directions (by Accident). In Proceedings of the IEEE/CVF Conference on Computer Vision and Pattern Recognition (CVPR).
- [4] Dai, B., Wang, Y., Aston, J., Hua, G. and Wipf, D. (2018). Connections with Robust PCA and the Role of Emergent Sparsity in Variational Autoencoder Models. Journal of Machine Learning Research, 19(41), pp. 1–42
- [5] Lucas, J., Tucker, G., Grosse, R. B. and Norouzi, M. (2019a). Don’t Blame the ELBO! A linear VAE Perspective on Posterior Collapse. In Advances in Neural Information Processing Systems, vol. 32.
- [6] Bowman, S. R., Vilnis, L., Vinyals, O., Dai, A., Jozefowicz, R. and Bengio, S. (2016). Generating Sentences from a Continuous Space. In Proceedings of The 20th SIGNLL Conference on Computational Natural Language Learning.
- [7] Rezende, D. J., Mohamed, S. and Wierstra, D. (2014). Stochastic Backpropagation and Approximate Inference in Deep Generative Models. In Proceedings of the 31st International Conference on Machine Learning, Proceedings of Machine
Learning Research, vol. 32.
- [8] Bonheme, Lisa, and Marek Grzes. "Be more active! understanding the differences between mean and sampled representations of variational autoencoders." The Journal of Machine Learning Research 24.1 (2023): 15423-15452.

---

> ### Author Response · Authors · 2024-11-21
>
> Re annealing: We did conduct experiments along those lines, but we found little change by including the annealing in the standard $\beta$-VAE (it simply moves to a different point in the standard VAE curves, dashed lines, in Fig.2 of the paper, but the overall shape of the curves doesn't change). This is expected if our hypothesis is correct, since the samples in that case are still in the equators.
> ***
> Re generalization of other tasks: The classification task was not really included to support our higher density claims, but to complement our interpretation of the lower entropy states. The aim was not to improve or compare classification metrics, and this task is only tangential to our main claims. This is why we stopped those comparisons after we considered that we made our point in that aspect. Our code always generates those types of figures at the end of the training for any of the datasets anyway, and the results are qualitatively similar to the ones presented in the paper for MNIST (although, with lower classification accuracy for all cases, standard or compression VAE, due to the increased complexity of those other datasets). The same with the projections on the $2-$sphere.
> ***
> Re equations, typos, etc.: thank you for pointing out these mistakes, we will update with a corrected version of the paper in the coming days.
> ***
> Re regimes of the VAE and posterior collapse: We added an extensive discussion related to the different regimes of the VAE and posterior collapse in new appendices, A.6, A.7, A.8, and A.12. Our finding is that, in the standard VAE, a high $\beta$ value produces considerable collapse. Nevertheless, in the high dimensional case, for each dimension, these collapsed models produce the best results (for the standard VAE) in terms of FID, i.e., the generative model is good in this sense. The price to pay for this is a low reconstruction accuracy, or `blurriness' in the decoded images. Thus, these collapsed models simply behave like non-collapsed ones in a much lower dimension for the latent. It's only in the lowest dimension that we considered ($50$), that the de-collapsing of the model improves the FID; nevertheless, in these cases, due to the low dimension, *both* the collapsed and de-collapsed models have very low reconstruction accuracy, so they fall very far away from the useful zone in the MSE-FID plane of our Fig.2 in the paper.
>
> It's simply not possible to produce non-collapsed standard VAE trainings in high dimensions and with a high $\beta$. This explains why there's a limit on how much it can go in the MSE-FID plane, as we show in Fig.2 of our paper for CIFAR10 (and now also for CelebA64 in new appendix A.10).
>
> One could try to forcefully de-collapse the models, but this alone only helps in maintaining more stable reconstructions but at the price of making the FID worse. We experimentally prove that this has to be because of the sparsity gained by the use of all the available dimensions (which make the reconstruction more expressive), since the FID only improves when we start the volume compression with our method. We discuss this in detail in A.7 and A.8.
> ***
> Re extra computation time: We measured an increase of 32 per cent in the training time required *per epoch* for training a CIFAR10 model with $n=200$ and mini batch size$=200$, in comparison to the same situation for the standard VAE. We will update the paper to provide the increase rate using big O notation in terms of the dimensions in the coming days, thank you for pointing this out.
> ***
> Re projection of active variables: We already mentioned the different modes of the VAE and how they reflect in the metrics we studied. Regarding the sphere projection, since we do this projection by first diving the (Cartesian) coordinates of $\mu$ into three groups (e.g., if $n=90$, from index $0$ to $29$ in the first group, $30$ to $59$ in the second, and $60$ to $89$ in the third) and then we average all coordinates within each group to obtain the corresponding $3-$D coordinates $x$, $y$, and $z$ for that $\mu$, then this automatically only takes into account the active, non-collapsed coordinates (since a value of $\mu_i=0$ for a same index $i$ for *all* $\mu$ becomes irrelevant in the sum performed in the averaging, only the active variables contribute to the projection).
> ***
> (continues in next comment...)

---

> ### Author Response · Authors · 2024-11-21
>
> (... continuation)
> ***
> Re impact of differetnt values of $a_{\mu,k}$: Thank you, this is a very useful question for us to better explain the more subtle points and results of our method. In the new appendix A.8 we start with a moderate amount of compression by encouraging the mean of the (cosine) angles $\overset{\mu}{\varphi_{k}}$ to be in the same direction as the vector whose Cartesian coordinates are $(1,\dots,1)$, i.e., $a_{\mu,k}=1/\sqrt{k+1},\,\forall k$. Indeed, recall that the closer we get to the north pole, the lower the volume. Nevertheless, this moderate compression is not enough to significantly improve the generation. Thus, we go to full compression mode by setting $a_{\mu,k}=1,\,\forall k$, which encourages all the points to converge and condense at the north pole. It's only in this regime of very high compression that we get a significantly appreciably improvement in the generation.

---

> > ### Comment · Reviewer_8unt · 2024-11-22
> >
> > I thank the authors for the additional analysis they performed on Celeba and CIFAR as well as the provided clarifications.
> >
> > I still have some questions and comments:
> >
> > - *Re FID and posterior collapse:* is FID a good metric for measuring generation quality? In the case of a fully collapsed VAE, the decoder will generate random noise ignoring the latent representations (which will always have a mean close to 0 and a variance close to 1) so one would expect a low FID. Despite this, the generation quality is poor (random noise) so I am not sure one could say that a low FID is always representative of a better generation (as indicated in Fig. 2 for example).
> >
> > - *passive/collapsed variables are ignored by the decoder*: the authors mentioned in A.6 that "they believe that [...] the decoder simply ignores the dimensions in question". This has been shown by [1,2] which the authors may want to cite there.
> >
> > - there are several typos in the new appendix, I encourage the authors to proof read them carefully during any subsequent revisions.
> >
> > - all my comments from the mathematical soundness and minor comments section are yet to be adressed.
> >
> > - if the authors do another revision, could they provide a diff file to highlight the new parts and facilitate further review?
> >
> > Overall, despite the updates provided by the authors, I am still not convinced that the paper quality is sufficient for acceptance in its current state. I will thus leave my score as it is for now.
> >
> > - [1] Rolinek, M., Zietlow, D. and Martius, G. (2019). Variational Autoencoders Pursue PCA Directions (by Accident). In Proceedings of the IEEE/CVF Conference on Computer Vision and Pattern Recognition (CVPR).
> > - [2] Dai, B., Wang, Y., Aston, J., Hua, G. and Wipf, D. (2018). Connections with Robust PCA and the Role of Emergent Sparsity in Variational Autoencoder Models. Journal of Machine Learning Research, 19(41), pp. 1–42

---

> ### Author Response · Authors · 2024-11-25
>
> Please note that the FID does not compare generation with original data (as usually done). In our paper FID compares the reconstruction and the generation. So generated images could be blurrier but still have lower FID if they are closer to the reconstruction images (i.e. the FID in our paper is relative to the reconstruction, not to the original data).
>
> This is a very important point that some of the reviewers seem to have overlooked.
>
> Thank you for the additional references, we will update the paper.
>
> We wanted to provide the additional materials ASAP for this review discussion. We will update the manuscript to fix the errors/typos that you pointed out, thank you. We will save room by moving the VAE description in appendix (fixing the equations issue). EDIT: many of these issues are now fixed in the new uploaded version. We are still working in fixing/adding the references and typos.

---

> > ### Comment · Reviewer_8unt · 2024-11-25
> > **clarification on FID**
> >
> > I am not sure I understand what the authors mean regarding FID here?
> >
> > Given some dataset $X$, an encoder $E$, a decoder $D$ and the output reconstructed/generated by the decoder $D(E(X)) = \\hat{X}$, I assumed that the authors compared $\\hat{X}$ with $D(E(\\hat{X}))$, hence the low FID of collapsed models.
> >
> > Was this assumption incorrect? If so could the authors define what they mean by reconstruction and generation in that context? Both terms are often used interchangeably in the literature.

---

> ### Author Response · Authors · 2024-11-25
>
> We call "reconstruction" of $X$ to $\hat{X}$. We measure how good this reconstruction is by the standard Mean Squared Error (MSE). This is correlated with the perceived level of "blurriness" of the reconstructed images w.r.t. the originals.
>
> We now take an equal number of random samples, $Z_{\epsilon}$, from the prior in latent space and decode them, $X_{\epsilon}=D(Z_{\epsilon})$ . We compute the FID between $\hat{X}$ and $X_{\epsilon}$. This in order to avoid the `blurriness', which was measured separately via the MSE, to interfere here. We call this "generation" measurement.
>
> After much consideration, we decided on this approach because of the following. A good VAE-based generative model should minimize both of these metrics *simultaneously*: that is, to be able to generate random samples which are *in-distribution* w.r.t. the reconstructed dataset (low FID), and such that the latter actually resembles the original dataset (low MSE).
>
> Re low FID in collapsed model: first, none of the models are fully collapsed, they all still have some functional latent dimenions. This makes them similar to a fully non-collapsed model in much lower dimensions. Thus, one can have a low FID (in our sense) in those situations (we measured the FID in both the HD semi-collapsed, Fig.10, and low D non-collapsed, Fig.26), but the reconstruction suffers (high MSE). Note the similitude of the MSE in both of these examples (the FID is a bit higher in the HD case, but still the lowest for that dimension as we vary $\beta$).

---

> ### Comment · Reviewer_8unt · 2024-11-25
>
> I thank the authors for the clarification. This would be very useful to include in the paper for clarity.
> However, my comment about good FID in collapsed models still stands. Indeed, as mentioned in my intial comment, fully collapsed models will generate random noise and ignore the latent. Sampling from $Z_\\epsilon$ will lead to the same result as the decoder will ignore the latents in both cases. Thus one will get low FID but a very poor image quality. As mentioned by reviewer Ww1f, it has previously been observed that low FID may not be representative of good image quality and collapsed VAEs are an illustration of such case.
>
> While I understand the author's idea of considering that a good model is able to simultaneously reconstruct correctly images from the original dataset and also good quality images in inference mode (i.e., using $Z\\epsilon$), my initial concern was that FID is referred to as representative of the generation quality in the paper, see Fig. 2 for example. Given what was discussed above and also with reviewer Ww1f, I find this misleading as collapsed models do not provide good generation quality. I thus think several updates of the papers would be beneficial. Specifically, I suggest that the authors:
> 1) Clearly define what the goal of FID is in their case
> 2) How they compute it
> 3) Change generation quality to a better defined term to avoid confusion.
>
> At the moment I believe that the paper's idea is promising but the amount of work needed to make the argument clear is too important to accept it in its current state. I thus leave my score as it is.

---

> ### Author Response · Authors · 2024-11-25
>
> We added an edit to the previous comment in reference to collapsed models: Re low FID in collapsed model: first, none of the models in our experiments are fully collapsed, they all still have some functional latent dimensions. This makes them similar to a fully non-collapsed model in much lower dimensions. Thus, one can have a low FID (in our sense) in those situations (we measured the FID in both the HD semi-collapsed, Fig.10, and low D non-collapsed, Fig.26), but the reconstruction suffers (high MSE). Note the similitude of the MSE in both of these examples (the FID is a bit higher in the HD case, but still the lowest for that dimension as we vary $\beta$). In Fig.2, these cases appear to the lower right: low FID, but high MSE. Therefore, too blurry to be considered good. Thus, by keeping track of both FID and MSE, we can make a good assessment.
>
> All of our experimental results follow qualitatively what one would expect by looking at the obtained FIDs-MSE pairs and the corresponding plotted images. See A.6, A.9, and the new A.14 (for the CelebA64 case). Thus, for the purposes of our experiments and claims, we have extensive evidence that the FID in our sense works perfectly well as a measure of in-distribution wrt the reconstructed data.
>
> 1-2) We described this in section 4.5 of the paper. We highlighted the relevant interpretations now.
>
> 3 ) Yes, this is something we could consider (particularly the naming of the vertical axis in Fig.2).

---

> ### Author Response · Authors · 2024-11-26
> **The argument about mode collapse is not correct.**
>
> We agree that a "**fully** collapsed models will generate random noise and ignore the latent". But this is a completely useless model with a terrible maximal MSE.
>
> We don't understand why this reviewer still brings up this point.
>
> None of our models have full mode collapse. None of the standard VAE are fully collapsed in our experiments, and we did an extra experiment with a VAE that has no mode collapse at all. In all cases, it shows that the compressed VAE provides a better alternative in the practical area of the MSE/FID plane.
>
> We do not consider FID on its own, but only in the context of the reconstruction quality.
>
> The edit suggested of the paper makes sense and are easy to do. We will emphasize more that the FID in our paper is not the usual FID by calling it self-FID.
>
> We don't understand what is the extra work the reviewer is referring to.

---

> > ### Comment · Reviewer_8unt · 2024-11-26
> >
> > *Regarding FID*
> > I understand that the authors don't have any collapsed models and that this case is not relevant to their analysis. My point about collapsed models was to illustrate why using FID as a "good generation quality metric" was misleading in some cases. Another example was also discussed by reviewer Ww1f.
> > Overall my main point about this and some of my previous points is that the paper is that the paper lacks clarity in some aspects.
> >
> > *Regarding additional work*
> > I have carefully reviewed the latest update and I acknowledge the amount of updates made by the authors and I thank them for their dedication as I believe this represents many hours of work. This has done a lot to improve the original version of the paper and I do not mean that the paper needs more work regarding experiments.
> > However, I believe that the paper still needs some work to improve its overall clarity.
> > I am sorry I cannot raise my score now, as I enjoyed the paper and the creative idea the authors had. I believe this paper has a nice potential but still lacks the clarity needed to convincingly make its point.

---

> > > ### Author Response · Authors · 2024-11-26
> > >
> > > Thank you for your sustained interest in the discussion and the paper. We appreciate that.
> > >
> > > Since we don't have too much time remaining, could the reviewer, if possible, pinpoint exactly which part of, for example, Section 4.5 of the paper (where we say what we mean by quality of the generation as measured by the FID and quality of the reconstruction measured by the MSE) could be made more clear? We simply state exactly what we mean there, and is the same interpretation we have been saying here. We really do not see how it could be even more clearer. Furthermore, the examples we added for CelebA64 in A.14 illustrate pretty convincingly that our interepretations of those FID values are sound (we will add a reference to that appendix in Section 4.5).
> > >
> > > We also changed the name to self-FID and the vertical axis of Fig.2 to "generation closer to reconstruction" (we will upload this new version of the pdf tomorrow, plus some other minor corrections). The use of "generation" in the title is correct because it does refer to generation in the absolute sense there (which is assessed by looking at *both* the MSE and the self-FID for each model). We will keep looking in the paper for any ambiguous use of the term "generation" and replace it with "generation closer to reconstruction" if it actually refers to improvement in the self-FID. But beyond that, we really cannot think of any other clarification regarding this particular issue.
> > >
> > > We agree that there was some potential for confusion in the terminology, but we consider that there should not be any confusion now after the highlights we made in Section 4.5: it's as explicit as it could possibly be. We also explained our rationale for this choice of metrics to evaluate our proposal.
> > >
> > > Finally, a general detailed discussion about the FID as a metric for evaluating generation is obviously beyond the scope of this paper. At some point, an evaluation metric has to be chosen to do the evaluation. We chose a rather popular and widely used one. We provided several examples in different datasets so that the reader can get an intuition from the images and the corresponding self-FID values. The interpretation we make is sound. Pathological cases are avoided in our experiements.
> > >
> > > As any proposed metric, FID certainly may have its drawbacks (as pointed by one reviewer), but those don't seem to be having too much of a role in our results, and, in any case, will be generic to anyone using it, not just our paper. We will add a word of caution and cite the reference suggested by the other reviewer, but we don't see any relation between this and some supposed insurmountable clarity issues in our paper.
> > >
> > > So, we ask, with all honesty, what can we do to clarify more? Can the reviewer please clarify?
> > >
> > > Best regards.
> > >
> > > The authors.

---

> ### Comment · Reviewer_8unt · 2024-11-30
> **Sorry for the late answer**
>
> Dear authors, sorry for my late answer, health issues prevented me from participating during the last few days.
> I have carefully read the latest version and believe that my concerns about clarity are now addressed. I am thus happy to raise my score. I thank the authors for engaging in the discussion and their dedication. I believe this paper is very interesting and, as I said before, I really enjoyed the creative approach there, I am thus very happy to raise my score to accept.
> Best wishes

---

### Official Review · Reviewer_Ww1f · 2024-10-31

**Soundness:** 2
**Presentation:** 3
**Contribution:** 2
**Rating:** 6
**Confidence:** 3

**Summary:**

The authors propose expressing the latent variable of a standard VAE in hyperspherical coordinates, reformulating the KL term of the ELBO loss accordingly, to reduce the sparsity in high-dimensional latent spaces and improve the generative performance of the model. Their approach draws on a parallel between high-dimensional spaces in statistical physics, specifically spin glasses, and the training dynamics of a VAE.

**Strengths:**

- To the best of my knowledge, this work introduces a novel approach to improve generative performance in VAEs by constraining latent representations, exploiting the hyperspherical coordinates formulation to reduce sparsity in the high-dimensional latent space.
- I find the connection between VAE training and physical systems, such as spin glasses, particularly valuable, as it provides a novel perspective for understanding the model's training dynamics.

**Weaknesses:**

- The authors should revise the references (e.g., replace arXiv preprints with published versions where available, add access dates for blog references, and consider replacing Wikipedia links with more reliable sources). Additionally, some typos were noted in the main text, and Figures 4 and 5 appear in low resolution, making the text difficult to read.
- I think the experimental section could be strengthened. Although the primary aim of this work is to improve the model's generative performance, the authors present generated samples only on MNIST, which is a simple dataset. Also, they do not compare the generative performance of the method with any other baseline (in the introduction they mention methods that improve generative performance by using more flexible priors but do not show if the results achieved are comparable).
- The paper lacks a concluding discussion and does not explore potential limitations of the method or directions for future research, which I believe would add significant value.

**Questions:**

- What are the main limitations of this method?
- Could the authors provide a comparison with additional baselines and training times? It would be valuable to compare the generative performance and training times of the proposed method with other state-of-the-art approaches to evaluate whether the additional computations required for using hyperspherical coordinates are justified by the improvements in generation performance.
- The authors state that ‘the random samples of an independent multivariate Gaussian distribution fall in the equator of a hypersphere, and thus none of them is near the singularities of the hyperspherical coordinates’. However, once the latent samples are forced away from the equator, could it be possible to fall near the singularities of the hyperspherical coordinates?
- The authors have presented generated images only on MNIST, a relatively simple dataset that does not require a high-dimensional latent space to capture its features. As a result, introducing additional constraints in the latent space does not appear to limit its capacity to represent information. However, with more complex datasets, how can these constraints affect the expressivity of the model given that the representations tend to overlap (as stated in Figure 1)?
- Comparing the results in Figure 2 is challenging because the configurations (dimensionality of the latent space) are positioned at different points (not aligned). Consequently, it is difficult to determine in which configurations or regimes the VAE with hyperspherical coordinates surpasses the vanilla VAE and vice versa.
- In section 4.4, the authors generate new data sampling from a von Mises–Fisher distribution with the same mean and covariance as the ones empirically calculated from the latent embedding of the full test dataset. What is the motivation to use the empirical statistics of the test set instead of the training set for generation?
- Could this method be extended/generalized to other distributions?

---

> ### Author Response · Authors · 2024-11-20
>
> Re references, typos, etc.: Thank you for pointing this out, we will correct and update the paper.
> ***
> Re experimental section: We addressed similar comments brought up by reviewers 1, 2, and 3 above.
>
> We provided explicit samples in the case of MNIST because it is easy to appreciate the differences qualitatively and thus convey the main idea of the paper. Nevertheless, we have updated the paper now to include several more examples in the other datasets that we studied (CIFAR10 and now also CelebA64).
>
> The main quantitative performance metrics were the mean square error (MSE, to measure the quality of the reconstructions) and the Frechet Inception Distance (FID, to measure the quality of the generation). FID was calculated between the reconstructed samples and the decoded random samples. *These metrics were systematically evaluated in a full parameter sweep* in CIFAR10 (and now also CelebA64), where our method shows a consistent improvement in line with our claims (Fig.2 of the paper).
>
> We stress that the aim of this work is not to produce state-of-the-art generation, but to understand why VAEs are poor generators in high dimension. It should be viewed as an investigative inquiry, hence the title of the paper. The method of volume compression via hyperspherical coordinates was specifically designed to test whether sparsity is the problem.
>
> For this reason comparing with other VAE variations that aim to improve generation is not relevant. For example, the references using flexible priors are focused on the so-called `prior hole problem' (mismatch between prior and the posterior). This can cause problems for the generation but is not related to our hypothesis: even with perfect match, the sparsity problem is present.
> ***
> Re conclusions section: This is a fair point as we ran out of space. We will expand our conclusions section in the paper, adding the following remarks.
>
> - the improvement in generation was only evaluated for the purposes of hypothesis testing, and not as absolute performance.
>
> - we did not evaluate the method for high resolution and larger datasets such as imagenet.
>
> - the extra computing time is about 32\% more per epoch for 200 latent dimensions. In higher dimensions, the added computation increases and might become prohibitive.
>
> - future research can focus in optimizing this method (or other method that takes into account the hypothesis about sparsity) for obtaining state-of-the-art results in generation and other tasks, in VAEs and other models.
>
> - The use of latent representations in hyperspherical coordinates can also be further explored in several other applications (perhaps unrelated to compression and generation), by the use of the provided script for the conversion and inspired by its proof of concept of practical feasibility in the present paper.
> ***
> Re singularities of the hyperspherical coordinates: This is a very interesting point and where one can appreciate the very peculiar nature of HD spaces. In low dimensions (e.g., the $2-$sphere), if one were compressing all the samples towards the north pole, the result would indeed be an island-like condensation there, centered at the pole, with many of the samples in the inner part of the island falling into the singularities of the hyperspherical coordinates. In HD spaces, however, the situation is very different: rather than a full island, one usually gets a ring-shaped distribution, with no samples at the center. This can be seen in all our histograms for angular variables when we display the results of the compression mode in several of the appendices of the paper. It is a generic effect, sometimes called the `ring or band effect' (see [1]). Thus, the north pole singularities will always be avoided in HD, even if one is deliberately trying to reach them!
>
> [1] Eliran Subag. The geometry of the Gibbs measure of pure spherical spin glasses. Inventiones
> Mathematicae, 210(1):135–209, 10 2017. ISSN 00209910. doi: 10.1007/s00222-017-0726-4.
> ***
> (continues in next comment...)

---

> ### Author Response · Authors · 2024-11-20
>
> (... continuation)
> ***
> Re figure 2: This figure contains our central experimental results, so we have now included more details. Results are plotted as they are, we don't understand how they can be aligned. We will include the extra explanation below in the appendix A.5.
>
> One could start first by looking at Fig.7 in appendix A.5, to familiarize with the basic trends (we offer in that figure an explicit break up of the results in terms of both latent dimension and KLD gain $\beta$). In the standard VAE, for a fixed $\beta$, as we increase the latent dimension, the FID increases (worse generation), but the MSE decreases (more sharp, less blurry images); for a fixed latent dimension, as we increase $\beta$, the FID decreases (better generation), but the MSE increases (less sharp, more blurry images).
>
> In the compressed version, we know, from the MNIST example in Fig.1, that it can improve the generation in HD cases where the standard VAE produces meaningless images for decoded random samples. But this comes at the price of losing some reconstruction quality. On the other hand, we also know that, in high dimensions and with a high $\beta$, the standard VAE is prone to collapse and then give better generation but with worse reconstruction quality (like a non-collapsed model in lower dimensions).
>
> It is important to plot all of these different regimes in a single MSE-FID plane, to be able to compare the collapsed HD, high $\beta$ standard VAE with a non-collapsed results in lower dimensions. It is also interesting to compare all of these cases with the compressed version: there are regions on the MSE-FID that are only accessible via the compression VAE.
>
> This is what we find: if we select an acceptable value for the MSE (e.g., $7.5$; higher values are considerably more blurry), the lowest FID value that any standard VAE model is around $39$ (for any latent dimension and any $\beta$, and any mode, collapsed or not). For the same MSE value, the compressed versions can achieve lower FIDs, of around $28$.
>
> Thus, the compressed model is able to retain a given `good' reconstruction in HD while keeping a low FID, something not possible with a standard VAE.
> ***
> Re empirical statistics: We tried both and there was only a marginal difference, not relevant enough. We preferred the test version since it takes less time (fewer samples) and was easier and more natural to implement in the code (since this calculation is done during testing).
> ***
> Re other distributions: Interesting question. The method is by construction adapted to distribution on the hypersphere and the KLD expression is only valid for Gaussian prior. A different prior would probably require a sampling-based divergence method (E.g. sliced Wasserstein), which we want to investigate in future work.

---

> ### Comment · Reviewer_Ww1f · 2024-11-23
> **Response**
>
> Thank you for your response! I appreciate the additional experimental results and your clarifications. However, I still have some concerns:
>
> - While I acknowledge your response regarding the conclusions, I have noticed that there is no dedicated conclusions section in the revised paper. I think it is important that the authors include this. As the paper has already reached the page limit, it would be helpful to see how they plan to restructure the main text.
>
> - As Reviewer 8unt pointed out, the equations, typos, and references have not yet been corrected in the revised version. I would also recommend highlighting these corrections in a different color to make the revision process easier.
>
> - Regarding the CIFAR10 generation examples, I share Reviewer 8unt’s concerns that FID alone may not be the best way of evaluating generation performance. In Figure 10, we observe lower FID scores but blurry generated images, while in Figure 12, higher FID scores correspond to more visually appealing images with greater detail. A similar discrepancy can be seen between Figures 16 and 17. This aligns with findings in prior work suggesting that FID does not always correlate with human evaluations, potentially due to its reliance on the pre-trained Inception-V3 model [1]. Since the goal of the paper is to investigate why VAEs perform poorly as generators in high-dimensional regimes, it would strengthen the claims if the Authors provided a more comprehensive evaluation of generation performance.
>
> If these concerns are adequately addressed, I would be happy to reconsider and raise my rating. For now, however, I am maintaining my current score.
>
> **References**
>
> [1] Stein, George, et al. "Exposing flaws of generative model evaluation metrics and their unfair treatment of diffusion models." Advances in Neural Information Processing Systems 36 (2024).

---

> > ### Author Response · Authors · 2024-11-26
> >
> > We invite the reviewer to read the interchange with reviewer 8unt below, for a follow-up on these same raised issues. Thank you.

---

> > > ### Comment · Reviewer_Ww1f · 2024-11-26
> > >
> > > I would like to thank the authors for their response and acknowledge the effort they have put into improving the manuscript during the rebuttal phase. As a result, I am slightly increasing my score. My comment on the FID metric was meant to point out that most of the discussion about improved generation performance in the paper focused on FID scores.  Since this metric has known limitations, I believe the discussion should not rely **entirely** on it, especially given that the main claim of the paper focuses on generation performance in VAEs. Adding and discussing examples, such as the CelebA case, provides helpful context and makes the results easier to interpret. However, as Reviewer 8unt says, the text should be clearer regarding the metric employed and what the authors mean when they say they improve generation.
> > >
> > > Additionally, I believe it would be helpful to include samples from both the prior and the aggregated posterior in a visualization similar to Figure 1 (or any other visualization the authors deem appropriate). This would show how well they align and further illustrate that the issue is not related to the alignment of the posterior and prior, but rather to sparsity. In the end, introducing flexible priors that better match the aggregated posterior has been shown to improve generation performance in the literature, so I think a discussion in this regard could be helpful. Please let me know if you have already included a discussion on this topic and I may have missed or overlooked it
> > >
> > > Minor comment: I still have trouble reading the text in Figures 4 and 5 given the low resolution, and authors should revise the references and links included in the paper (as I mentioned in my first review).

---

> ### Author Response · Authors · 2024-11-25
>
> We agree re the conclusion, we are working on an update. We will save room by moving the VAE description in appendix (fixing the equations issue).  EDIT: all of this is now fixed in the new uploaded version.
>
> **Please note that the FID does not compare generation with original data (as usually done). In our paper FID compares the reconstruction and the generation**. So generated images could be blurrier but still have lower FID if they are closer to the reconstruction images. This is a very important point that some reviewers seem to have overlooked (Cf. our answer to reviewer TsTD about the performance metrics used in the paper: "*FID, to measure the quality of the generation; this metric was calculated between the reconstructed, i.e., decoded, original data and the decoded random samples, in order to avoid the `blurriness', which was measured separately via the MSE, to interfere here*").
>
> We are not proposing a better generator, we are arguing that sparsity is an issue. In the standard VAE, there is a trade off between reconstruction and generation. When the latent is compressed, this trade off is much reduced in the case of MNIST (good reconstruction and good generation). It is still present in more complex images (CIFAR10, CelebA) but always with better performance for both reconstruction and generation in the compressed version.
>
> Note that the blurry generated images in Fig.10 (third row in the images panel) look more like the reconstructed data (second row of that panel), than the analogue situation in Fig.12. This is correlated with the FID values displayed there. The greater details in Fig.12 come from a lower MSE; nevertheless, those details are out-of-distribution w.r.t. the reconstructed dataset, they don't correspond to details that could be present in an image in the original dataset, it's just noise fabricated by the decoder, hence the greater FID. We have CelebA results in which, for the case of latent dimension $n=1000$ and $\beta=0.09$, the trend just described for Fig.12 is even more clear (the noise in each pixel is very sharp, almost no face can be seen, but the reconstruction is very good). We will try to include this in that appendix in the limited time remaining. EDIT: We included these results in new appendix A.14 in the new uploaded version.

---

> ### Author Response · Authors · 2024-11-28
>
> Hello.
>
> Thank you for your engagement and willingness to raise the score.
>
> Regarding the FID, your initial concern with this metric is based on what we consider a clear misunderstanding and reading of the presented figures. We provided an explicit rebuttal to your comment in an edit, so we will repeat it in case it was missed: Note that the blurry generated images in Fig.10 (third row in the images panel) look more like the reconstructed data (second row of that panel), than the analogue situation in Fig.12. This is correlated with the FID values displayed there. The greater details in Fig.12 come from a lower MSE; nevertheless, *those details are out-of-distribution w.r.t. the reconstructed dataset*, *they clearly don't correspond to details that could be present in any image in the original dataset, it's just noise fabricated by the decoder, hence the greater FID*. We have CelebA results in which, for the case of latent dimension $n=1000$ and $\beta=0.09$, the trend just described for Fig.12 is even more clear (the noise in each pixel is very sharp, almost no face can be seen, but the reconstruction is very good). We included these results in new appendix A.14 in the new uploaded version.
>
> Thus, whatever issue that the FID may have, is clearly **not** present in our experiments in the range and scope in which we are operating. In all the explicit examples, the values follow the qualitative feel of the images. Then, the choice of FID is sound, good, and enough for our purposes. We consider that any further discussion of this point and unwillingness to raise the score should clearly include an explicit rebuttal to the details in the answer of the previous paragraph, which are the actual points you initialy raised. Furthermore, we already highlighted very explicitly the meaning and way in which we assess the generation quality in Section 4.5, as we already mentioned to reviewer 8unt.
>
> We included a new figure, Fig.32 in A.15, where we show decoded samples from the approximate aggregate posterior in a standard VAE. There is little to no improvement in the generation wrt to the standard VAE. Flexible priors may indeed improve the generation, but for different reasons (reduction of the mismatch). We have provided extensive evidence by now that the main issue in HD is the sparsity, mismatches between prior and posterior are only an additional on top of that. Even more. In Fig.14 of A.8, we take a moderately compressed model (by our method) and show the 3D-embedding of the approximate aggregate posterior, which is a von Mises-Fisher-like distribution, and it shows no holes or cracks (since our method, by compressing things, makes the clusters more packed and closer to each other). If the philosophy of the mismatch between prior and posterior as root cause of bad generation were true, then sampling from such an approximate aggregate posterior von Mises-Fisher-like distribution should produce better generation. But we do exactly that in that figure and we don't see any improvement. It's only when we do a full compression mode that we get the improvement in generation. Thus, it was the sparsity. See also new appendix A.15 for even more elaboration on these points.
>
> Thank you.

---

> > ### Comment · Reviewer_Ww1f · 2024-11-28
> >
> > Thank you for your response. However, the current version of the manuscript does not include the appendices. I would appreciate it if you could upload the complete version.

---

> > > ### Author Response · Authors · 2024-11-28
> > >
> > > Hi.
> > >
> > > Yes, we moved the appendices to the Supplementary Material because there's a limit of 50mb for the main text. The Supplementary Material download icon should be below the abstract and visible for the reviewers, as far as I know.

---

> > > > ### Comment · Reviewer_Ww1f · 2024-11-28
> > > >
> > > > Thank you for the prompt response! I hadn’t noticed the Supplementary Material earlier.
> > > >
> > > > I’d like to thank the authors for their clarifications and the additional experimental results addressing my concerns. I recognize the effort and dedication put into the rebuttal. As a result, I am raising my score to a 6.
> > > >
> > > > That said, I believe the presentation of the paper could still be improved, which is also an important aspect. Specifically, the figures should clearly indicate that the metric being shown is the self-FID. Additionally, the references should be corrected (e.g., "J Tomczak. Priors (blogpost). URL https://jmtomczak.github.io/blog/7/7_priors.html," where the date of access is missing, or published conference articles cited as their ArXiv versions). Finally, the main text should point readers to the additional experimental results provided in the Appendix (e.g., CelebA results).

---

> > > > > ### Author Response · Authors · 2024-11-29
> > > > >
> > > > > Hello.
> > > > >
> > > > > Thank you for your response and raising the overall score from the initial 3 to 6.
> > > > >
> > > > > Regarding the other concerns: yes, unfortunately we didn't have enough time to address all of those since we tried to focuss on the more contentious issues, which required most of our attention in the edits.
> > > > >
> > > > > Best regards.

---

### Official Review · Reviewer_TsTD · 2024-11-01

**Soundness:** 2
**Presentation:** 3
**Contribution:** 2
**Rating:** 6
**Confidence:** 3

**Summary:**

The paper proposes a new way to formulate the latents of VAE on hyperspherical coordinates. They use real MNIST data to show the improved generalization ability of VAE.

**Strengths:**

- The paper is well written.
- The relevant works are clearly listed.
- The idea of drawing insights from high dimensional physical systems is interesting.

**Weaknesses:**

The experiment results section is weak.
- The evaluation is only qualitative comparison. Is there any quantitive metric (e.g. classification/prediction error) that can be used to compare the proposed method with existing methods?
- Lack of comparison with other related hyperspherical VAE work listed in the section 2. related works. e.g. Hyperspherical Variational Auto-Encoder’ Davidson et al. (2018),  Yang et al. (2023), Bonet et al. (2022), etc.
- The paper only shows results on one real dataset. How well does the proposed method generalize to more datasets?

**Questions:**

See the weakness section above.

---

> ### Author Response · Authors · 2024-11-20
>
> The evaluation is only qualitative in the case of MNIST, since it's easy to appreciate the differences there.
>
> The performance metrics in which we were interested are the mean square error (MSE, to measure the quality of the reconstructions) and the Frechet Inception Distance (FID, to measure the quality of the generation; this metric was calculated between the *reconstructed*, i.e., decoded, original data and the decoded random samples, in order to avoid the `blurriness', which was measured separately via the MSE, to interfere here). These metrics were systematically evaluated in a full parameter sweep in CIFAR10 (and now also CelebA64), where our method shows a consistent improvement in line with our claims (Fig.2 of the paper).
>
> It's beyond the purposes and scope of the current paper to evaluate the discussed methods on further downstream tasks, like classification, and optimization of metrics there for state-of-the-art results. Instead, our goal was to try to *understand* why VAEs seem to fail at the generative task in high dimensions. In particular, it was falsifying our main hypothesis, stated in the title of the paper. Our idea is to highlight this point and its connection with statistical physics, so that the broader research community could be benefited from this knowledge. Our points are mainly about high dimensional spaces and how they are becoming ubiquitous in modern machine learning, so that we should investigate more about their nature and effects. We have observed considerable interest on this point in the machine learning community.
> ***
> Despite the fact that some of the models in the mentioned references also work on the hypersphere in latent space, the similitude ends there. We do not really consider them suitable for comparison because both the goals and methods of these works are very different from ours. In particular, those approaches are not concerned with the sparsity issue of HD spaces as well as volume compression as a possible solution to it.
>
> Davidson et al. (2018), for example (and related elaborations), build a VAE with a KLD term between a uniform distribution on the hypersphere and a von Mises-Fisher approximate posterior. Thus, by construction, there cannot be any compression of the type we discuss in our work, it's still highly similar to the standard VAE in the sense in which the posterior distribution ends approaching a uniform distribution on the hypersphere, where, we claim, the sparsity issue arises.
> ***
> We have added now results for one extra dataset (CelebA64) in a new appendix A.10 that further support our claims.

---

> ### Author Response · Authors · 2024-11-28
>
> Hello.
>
> We invite you to engage with our rebuttal and consider changing your score if your points are well addressed.
>
> Thank you.

---

> > ### Comment · Reviewer_TsTD · 2024-12-03
> >
> > I thank the authors for their responses and additional experiments. They address my questions and I'm willing to increase my score to 6.

---

### Official Review · Reviewer_uan5 · 2024-11-01

**Soundness:** 2
**Presentation:** 3
**Contribution:** 2
**Rating:** 3
**Confidence:** 5

**Summary:**

This paper presents an enhancement to Variational Autoencoders (VAEs) by introducing a hyperspherical latent space with a novel loss function. The method aims to improve generative quality by concentrating embeddings in denser latent regions, moving them away from the equatorial band often associated with sparsity. The approach offers compatibility with existing VAE structures, requiring minimal adaptation to the standard VAE framework.

**Strengths:**

• Innovation and Relevance: The use of hyperspherical coordinates in latent space with principles drawn from statistical physics is a unique and potentially impactful innovation.
• Clear Problem Statement: The paper presents the problem of sparse latent spaces in VAEs clearly, providing a well-motivated solution.
• Practical Usability: The proposed method integrates easily with current VAE models and introduces a manageable computational overhead.
• Clarity of Presentation: The visual support, especially in Figure 1b, effectively illustrates the approach, aiding in understanding the latent space adjustments.

**Weaknesses:**

• Limited Experiments: The method is only evaluated on two datasets (MNIST and another dataset). More examples of generated samples and interpolations would better support the claim of improved density in latent space.
• No Comparison with Relevant Competitors: The paper lacks comparative experiments with established hyperspherical VAEs, such as Davidson et al. (2018), or with Riemannian approaches like “A geometric perspective on VAEs” by Chadebec and Allassonniere.
• Inconclusive Results: The results in downstream tasks like classification are mixed, and the paper’s claims of denser latent space do not
consistently reflect in performance metrics.

While the paper offers valuable theoretical insights, additional empirical support is needed to substantiate the generative benefits relative to state-of-the-art sampling methods.

**Questions:**

1. Could you provide comparisons with hyperspherical VAEs or other structured latent space models?
2. Have you explored interpolation in the latent space to support your density claims?
3. Why were only two datasets used? Including more diverse datasets could better validate your method.
4. Why would a latent space as tightly concentrated as shown in Figure 1 be desirable? This remains unclear.

---

> ### Author Response · Authors · 2024-11-20
>
> We have added now results for one extra dataset (CelebA64) in a new appendix A.10, as well as more figures with decoded images for all the considered datasets. We also included interpolations in a new appendix A.11 that further support our claims.
> ***
> Despite the fact that some of the models in the mentioned references also work on the hypersphere in latent space, the similitude ends there. We do not really consider them competitors because both the goals and methods of these works are very different from ours. In particular, those approaches are not concerned with the sparsity issue of HD spaces as well as volume compression as a possible solution to it.
>
> Davidson et al. (2018), in particular, build a VAE with a KLD term between a uniform distribution on the hypersphere and a von Mises-Fisher approximate posterior. Thus, by construction, there cannot be any compression of the type we discuss in our work, it's still highly similar to the standard VAE in the sense in which the posterior distribution ends approaching a uniform distribution on the hypersphere, where, we claim, the sparsity issue arises.
>
> Regarding “A geometric perspective on VAEs” by Chadebec and Allassonniere, our model is only Riemannian in the sense that the hypersphere is a basic example of a Riemannian manifold. The only key insight from differential geometry that we really use is the fact that a smooth manifold can be covered with different types of coordinate systems. This latter fact, which is the central aspect of our method, is not exploited in the mentioned reference, where everything seems to be done in terms of Cartesian coordinates. Furthermore, it also doesn't seem concerned with our main hypothesis about sparsity in high dimensional spaces.
> ***
> The classification task was not included to support our density claims, but to complement our interpretation of the lower entropy states. The aim here was not to improve classification metrics, and this task is only tangential to our main claims. In contrast, the performance metrics in which we were interested are the mean square error (MSE, to measure the quality of the reconstructions) and the Frechet Inception Distance (FID, to measure the quality of the generation). These metrics were systematically evaluated in a full parameter sweep in CIFAR10 (and now also CelebA64), where our method shows a consistent improvement in line with our claims (Fig.2 of the paper).
> ***
> It's not our purpose in this work to beat state-of-the-art sampling methods. Instead, our goal was to try to *understand* why VAEs seem to fail at the generative task in high dimensions. After hypothesizing that the exponentially diverging volume/high entropy localized mainly in the equators introducing too much sparsity could be the reason (or one of the reasons), we devised a method (volume compression via hyperspherical coordinates) that could attack this directly and explicitly in order to see if we could get any improvement. If the answer is positive, then we could conclude that our hypothesis is likely correct.
>
> And this is indeed what our experiments show. The method and the improvement in the metrics are mere tools or means for a goal, which was falsifying our main hypothesis, stated in the title of the paper. Our idea is to highlight this point and its connection with statistical physics, so that the broader research community could be benefited from this knowledge, not just those concerned with VAEs or state-of-the-art sampling methods there. Our points are mainly about high dimensional spaces and how they are becoming ubiquitous in modern machine learning, so that we should investigate more about their nature and effects. We have observed considerable interest on this point in the machine learning community.
>
> As a secondary point, we highlight the use and potential of hyperspherical coordinates representations themselves for manipulating data in latent spaces, for which we provide a simple, vectorized, and relatively cheap (in computing terms, for moderate high dimensions, generally lower than $1000$) script for the conversion; the compression method and its use in this paper serves as one possible application, and by no means we think it exhausts the full potential. As in the sparsity issue, it's about ideas and offering proof of concept for these ideas.

---

> > ### Comment · Reviewer_uan5 · 2024-11-24
> >
> > I thank the authors for their feedback on the reviews.
> > However, I am still not convinced about the benefit if sparcity and  tight concentration in the latent space compared to other approaches like decreasing the latent dimension or manifold learning in the latent space to sample from the estimated posterior. I keep my grade.

---

> > > ### Author Response · Authors · 2024-11-25
> > >
> > > The point of the paper is to improve generation while keeping a high number of latent dimensions.
> > >
> > > We demonstrated that compression allows to improve reconstruction and generation, unlike reduction of dimension as suggested (which affect negatively reconstruction).
> > >
> > > Manifold learning is beyond the point here, we do not try to build a better generator: we just show that sparsity is a problem. We speculate that any manifold learning method would work better if the latent is compressed (future work).
> > >
> > > Please note that the FID does not compare generation with original data. **In our paper FID compares the reconstruction and the generation**.

---

> > > ### Author Response · Authors · 2024-11-28
> > >
> > > Hello.
> > >
> > > Decreasing the dimension can indeed make the generation to look closer to the reconstructed initial data, but this comes at the price of very blurry reconstructions (because of the lack of expressive power in the latent space due to the low dimensionality). A good generative model should produce a generation that looks closer to the reconstructed initial data, but also such that the latter (the reconstructed initial data) is closer to the initial data (i.e., the reconstructions are not blurry). We showed in the parameter sweep of Fig.2 that the models that achieve this are always our compression ones rather than the standard VAE (low dimensional ones included), so we think the reviewer is clearly mistaken regarding this point.
> > >
> > > About manifold learning, the problem is that this will make the samples to be inside the manifold (that's good, since being outside will generate bad images), but if the manifold is sparse (which will happen for any standard choice in HD), then we would be in the same situation as the standard VAE. Only by adding a compression method to the manifold one could compare that with our approach. We didn't encounter such an approach in the literature.
> > >
> > > We invite you to answer to these points and to change your score if these are addressed by us (we think they are; if not, please offer the corresponding rebuttal to our answers).

---

### Official Review · Reviewer_Dgy1 · 2024-11-01

**Soundness:** 2
**Presentation:** 3
**Contribution:** 2
**Rating:** 3
**Confidence:** 3

**Summary:**

The authors propose to convert the latent variables of a VAE to hyperspherical coordinates. This allows them to constrain samples from the prior distribution to a small region in latent space especially if the latent dimension is large. They provide experimental evidence that this improves the performance of the VAE when generating new data. They also provide some theoretical justification arguing that the sparsity of the latent space impairs the smoothness of the latent manifold.

**Strengths:**

1.	The introduction of hyperspherical coordinates to better capture samples obtain from high dimensional Gaussians is useful especially in the context of constraining the latent variables
2.	The paper attempts to establish an ambitious connection between replica symmetry breaking in spin glasses and their proposed modifications of the loss function that avoids ‘high sparsity’ equatorial regions of the hypersphere.

**Weaknesses:**

1.	The connection to spin glasses is made via a formal similarity of the energy function of a spin glass and their regularization term (equation 8). This connection appears weak as it does not explain: a) how the regularization helps escaping local minima that correspond to low-quality outputs of the VAE, b) the effect on Parisi’s order parameter which seems at the heart of the spin glass theory of neural networks, and c) the role of temperature i.e., the learning rate in the proposed scheme by which desired low-entropic states are reached.
2.	I am not sure that sparsity of the latent representation is the root cause of poor generative performance in VAEs. “Posterior collapse” seems a more likely explanation (also of the empirically observed improvements in section 4) as the proposed constraints not only compress the volume but also simply decrease the variance of the prior distribution.
3.	As a minor point, in line 189-190 the authors see sparsity as an impediment to learning a data representation as a smooth manifold. I am not sure I agree unless the objective would be to explicitly construct the manifold (e.g., via simplicial complexes). But the manifold hypothesis (like the spin glass model) is only a conceptual aid (of how to think about VAE representations of data) not a fully developed theory from which algorithms and their convergence properties can be derived.

**Questions:**

Have you tried running experiments without angular constraints but only radius constraints (to test against the lower variance of the prior effect)?

---

> ### Author Response · Authors · 2024-11-20
>
> In the main text we chose to make an emphasis on the regularization term analogy because this aspect was the most clearly noticeable or immediate, and directly related with our proposal of priors in hyperspherical coordinates. Regarding the other specific points, the analogy goes like the following:
>
> b) while we did not mention it explicitly in the main text for lack of space, we do make extensive use of the relevant order parameter here (given by the law of the inner product between replicas) in our experiments. We use it to check if there is an overall replica symmetry breaking in our latent distribution. This parameter is explicitly plotted in all our reported figures in the appendices as a dashed red line in the same part of the figure where we plot the histograms for the norms of $\mu$ and $z$.
>
> a) and c) The gain $\beta$ here has the role of the inverse temperature, $\beta=1/T$.
>
> In spin glasses and complex systems, the energy function has exponentially many local minima in the equatorial region of the hypersphere. To overcome them, a very strong signal or bias towards the desired region is necessary at the beginning, together with a rapid cooling or quenching.
>
> Thus, our initial high $\beta$ (i.e., very low temperature $T$) setting, and in the presence of the high intensity hyperspherical external magnetic fields as bias in directions away from the equator, should make the gradient descent dynamics to quickly tend towards a low temperature distribution with replica symmetry breaking. Indeed, this is what we observed in our experiments, since we check for the replica angle, as mentioned before.
>
> This initial strong compression helps escaping those undesirable equatorial minima. Nevertheless, the obtained state shows too much overlapping between samples, so we then perform the annealing (i.e., lower the $\beta$, or increase the temperature $T$, and also lower the intensity of the magnetic fields) in order to allow the system to relax the strong order introduced by the initial bias and, in this way, transition to a replica symmetry breaking state with a bigger angle between replicas. This decreases the MSE and makes the decoded images more sharp, at the cost of some generation quality.
>
> This is fully consistent with the spin glass analogy in a quenched and then annealed system, where the glass, always in the replica symmetry breaking phase, jumps from one so-called `pure state' to a different pure state, i.e., goes back up a bit in the ultrametricity tree/hierarchy of the replica angle values. But the system has escaped the zone with exponentially many local minima in the equator. More details and figures are provided in a new added appendix A.9.
> ***
> We added an extensive discussion related to posterior collapse in new appendices, A.6, A.7, A.8, and A.12. Our finding is that, in the standard VAE, a high $\beta$ value produces considerable collapse. Nevertheless, in the high dimensional case, for each dimension, these collapsed models produce the best results (for the standard VAE) in terms of FID, i.e., the generative model is good in this sense. The price to pay for this is a low reconstruction accuracy, or `blurriness' in the decoded images. Thus, these collapsed models simply behave like non-collapsed ones in a much lower dimension for the latent. It's only in the lowest dimension that we considered ($50$), that the de-collapsing of the model improves the FID; nevertheless, in these cases, due to the low dimension, both the collapsed and de-collapsed models have very low reconstruction accuracy, so they fall very far away from the useful zone in the MSE-FID plane of our Fig.2 in the paper.
>
> It's simply not possible to produce non-collapsed standard VAE trainings in high dimensions and with a high $\beta$. This explains why there's a limit on how much it can go in the MSE-FID plane, as we show in Fig.2 of our paper for CIFAR10 (and now also for CelebA64 in new appendix A.10).
>
> One could try to forcefully de-collapse the models, but this alone only helps in maintaining more stable reconstructions but at the price of making the FID worse. We experimentally prove that this has to be because of the sparsity gained by the use of all the available dimensions (which make the reconstruction more expressive), since the FID only improves when we start the volume compression with our method. We discuss this in detail in A.7 and A.8.
> ***
> RE Question: Yes, we tried this indeed. We have now updated the paper with explicit examples that make the case for our point, from this perspective too, about the sparsity. In appendix A.7, we show an example in which posterior collapse is avoided (in both the Cartesian and hyperspherical coordinates representations), while the distribution of $\mu$ is still similar to a uniform distribution on the hypersphere, like in the standard VAE. Nevertheless, this is not enough to guarantee good generation, even with a high $\sigma$ as in the mentioned example.

---

> > ### Comment · Reviewer_Dgy1 · 2024-11-23
> >
> > The additional discussion of the posterior collapse is helpful. I still believe that its avoidance is the main driver of any improvement of the generative performance of the VAE. The fact that you achieve this by a judicious choice of the prior in hyperspherical coordinates and a square-root type annealing schedule during training remains interesting.
> >
> > However, no guarantees on the achieved performance improvement can be given at this stage and one wonders if simpler methods like reducing the variance of the prior (or a lower dimensional latent space in the first place) would not yield similar results.
> >
> > I believe the paper's strongest point is to highlight that some regularization constraints act as an external magnetic field (l. 301) and training needs to quickly reach a region in latent space consistent with these constraints followed by a relaxation ("quenching and annealing"). This seems like a more defensible claim (rather than "sparsity implies poor generation") given the material presented in the paper. As such it would already provide valuable insights into the training of VAEs with regularization constraints.

---

> > > ### Author Response · Authors · 2024-11-28
> > >
> > > Hello.
> > >
> > > Could you please provide an explanation of why you still consider that the "*avoidance [of posterior collapse] is the main driver of any improvement of the generative performance of the VAE*" even after we conducted the experiment you suggested (*)  in your initial review and then presented the results in A.7 that clearly contradict your initial intuition about this (there is no collapse, and yet the generation is bad)? We added several other appendices discussing collapse and relating the improvement in generation to the decrease of sparsity, always in non-collapsed models, too.
> > >
> > > As for the lower dimensional model possibility: yes, that is an excellent point, and the main motivation that we had for doing the parameter sweep we present in Fig.2 of the paper. That figure shows that the overall improvement in both reconstruction and generation being closer to reconstruction being done by our compression models cannot be reached by a standard VAE working with a different choice of parameters (dimension of latent and $\beta$, and also collapsed or non-collapsed).
> > >
> > > Finally, if you consider the insghts that we obtained regarding the training dynamics, etc., as valuable, then, also taking what we said in the paragaraphs above, invite you to reconsider the very negative score you gave initially and that is still your official score.
> > >
> > > (*) We are very confident in this because we had similar concerns when we started investigating these ideas and ran the exact experiment that the reviewer suggested a time ago, from which the results we added in A.7 come from. We didn't include this in the paper initially because we thought it was too much of a side point. But we changed our mind after this review and included it, it is indeed useful.

---

> ### Author Response · Authors · 2024-11-25
>
> The point of the paper is to improve generation while keeping a high number of latent dimensions. Regarding the avoidance of collapse, please check appendix A.7 and A.8 where we explicitly show, based on the suggested experiment, how it's not enough by itself to improve the generation even in MNIST. Then, in A.8., we show how it starts to improve only after the compression.
>
> We demonstrated that compression allows improving reconstruction and generation, unlike reduction of dimension as suggested (which affect very negatively the reconstruction, i.e., very blurry images).
>
> We do agree that the insight about training dynamics is very interesting and worth publishing!

---

### Author Response · Authors · 2024-11-20

We thank all the reviewers for their careful reading of the manuscript and raising of several interesting points, giving us the opportunity to expand and improve some aspects of our work and paper.

We will answer each reviewer one by one in the order in which their reviews appear here in the coming days. Since some of the reviewers raised similar issues, we may repeat a similar answer to each of them regarding those particular points.

Best regards.

The authors.

EDIT: the Appendix is now in the Supplementary Material.

---

### Meta-Review · Area_Chair_kEvt · 2024-12-20

**Metareview:**

This paper proposes a novel approach to improve the image generation quality of Variational Autoencoders (VAEs) by applying a mechanism akin to the quenching process used to reduce the entropy of high-dimensional systems. The authors hypothesize that the poor image generation quality of VAEs at inference time is due to the sparsity of high-dimensional spaces and propose to apply a change of coordinates from Euclidean to hyperspherical during the KL divergence computation.

The reviewers generally found the paper to be interesting and creative, with a good potential for improving the generation quality of VAEs. However, they raised several concerns regarding the clarity of the paper, the soundness of the experiments, and the interpretation of the results.

After a thorough discussion and revisions, the authors addressed many of the concerns, providing additional experimental results, clarifying the interpretation of the metrics, and improving the overall clarity of the paper. The reviewers appreciated the authors' dedication and engagement in the discussion. However, after discussion with the authors and among themselves, the paper is still very borderline, with three reviewers leaning towards acceptance and two towards rejection. We will therefore need to reject the paper in its current form. We would still like to encourage the authors to resubmit an improved version of the paper in the future.

**Additional Comments On Reviewer Discussion:**

see above

---

### Decision · Program_Chairs · 2025-01-22

Reject